# On the estimation of persistence intensity functions and linear representations of persistence diagrams

## Abstract

Persistence diagrams are one of the most popular types of data summaries used in Topological Data Analysis. The prevailing statistical approach to analyzing persistence diagrams is concerned with filtering out topological noise. In this paper, we adopt a different viewpoint and aim at estimating the actual distribution of a random persistence diagram, which captures both topological signal and noise. To that effect, [CD19] has shown that, under general conditions, the expected value of a random persistence diagram is a measure admitting a Lebesgue density, called the persistence intensity function. In this paper, we are concerned with estimating the persistence intensity function and a novel, normalized version of it – called the persistence density function. We present a class of kernel-based estimators based on an i.i.d. sample of persistence diagrams and derive estimation rates in the supremum norm. As a direct corollary, we obtain uniform consistency rates for estimating linear representations of persistence diagrams, including Betti numbers and persistence images. Interestingly, the persistence density function delivers stronger statistical guarantees.

## 1   Introduction

*Topological Data Analysis* (TDA) is a field at the interface of computational geometry, algebraic topology and data science whose primary objective is to extract topological and geometric features from possibly high-dimensional, noisy and/or incomplete data. The literature on the statistical analysis of TDA summaries has mainly focused on distinguishing topological signatures from the unavoidable topological noise resulting from the data sampling process. Toward that goal, the primary objective in designing statistical inference methods for TDA is to isolate points on the sample persistence diagrams that are sufficiently far from the diagonal to be deemed statistically significant in the sense of expressing underlying topological features instead of randomness. This paradigm is entirely natural when the target of inference is the unobservable persistence diagram arising from a filtration of interest, and the sample persistent diagrams are noisy and imprecise approximations to it. On the other hand, empirical evidence has also demonstrated that topological noise is not unstructured and, in fact, may also carry expressive and discriminative power that can be leveraged for various machine-learning tasks. In some applications, the distribution of the topological noise itself is of interest; in cosmology, see e.g., [WNv+21]. As a result, statistical summaries able to express the properties of both topological signal and topological noise in a unified manner have also been proposed and investigated: e.g., persistence images and

Submitted to 37th Conference on Neural Information Processing Systems (NeurIPS 2023). Do not distribute.

linear functional of the persistence diagrams. In a recent contribution, [CD19] has derived sufficient conditions to ensure that the expected persistent measure – the expected value of the random counting measure corresponding to a noisy persistent diagram – admits a Lebesgue density, hereafter called the persistence intensity function; see also [CWRW15]. The significance of this result is multifaceted. First, the persistent intensity function provides an explicit and highly-interpretable representation of the entire distribution of the persistence homology of random filtrations. Secondly, it allows for a straightforward calculation of the expected linear representation of a persistent diagram as a Lebesgue integral. Finally, the representation provided by the persistence intensity function is of functional, as opposed to algebraic, nature and thus analytically simpler. It is amenable to statistical analysis via well-established theories and methods from the non-parametric statistics literature.

In this paper we derive consistency rates of estimation of the persistence intensity function and of a novel variant called persistence density function in the $\ell_\infty$ norm based on a sample of i.i.d. persistent diagrams. As we argue below in Theorem 3.1, controlling the estimation error for the persistence intensity function in the $\ell_\infty$ norm is stronger than controlling the optimal transport measure $\mathsf{OT}_q$ for any $q > 0$ and, under mild assumptions, immediately implies uniform control and concentration of any bounded linear representation of the persistence diagram including (persistent) Betti numbers and persistence surfaces.

## 2   Background and definitions

In this section we introduce fundamental concepts from TDA that we will use throughout the paper. We refer the reader to [CM21, CD19] for detailed background and extensive references.

**Persistence diagrams.**   We define a persistence diagram to be a locally finite multiset of points $D = \{\boldsymbol{r}_i = (b_i, d_i) \mid 1 \le i \le N(D)\}$ belonging to the set

$$\Omega = \Omega(L) = \{(b, d) \mid 0 < b < d \le L\} \subset \mathbb{R}^2, \tag{1}$$

consisting of all the points on the plane in the positive orthant above the identity line and of coordinate values no larger than a fixed constant $L > 0$. The coordinates of each point of $D$ correspond to the birth and death times of a persistent homology feature, where time is measured with respect to the totally ordered set indexing a filtration. The restriction that the persistence diagrams be contained in a box of side length $L$ is a mild assumption that is widely used in the TDA literature; see [DL21] and the discussion therein. To simplify our notation, we will omit the dependence on $L$, but we will keep track of this parameter in our error bounds. Some related quantities used throughout are

$$\partial\Omega := \{(x, x) \mid 0 \le x \le L\}; \qquad \overline{\Omega} := \Omega \cup \partial\Omega;$$
$$\Omega_\ell := \left\{ \boldsymbol{\omega} \in \partial\Omega : \min_{x \in \Omega} \|\boldsymbol{\omega} - \boldsymbol{x}\|_2 \le \ell \right\}, \quad \ell \in (0, L/\sqrt{2}). \tag{2}$$

That is, $\partial\Omega$ is a segment on the diagonal in $\mathbb{R}^2$ and $\Omega_\ell$ consists of all the points in $\Omega$ at a Euclidean distance of $\ell$ or smaller from it.

**The expected persistent measure and its normalization.**   A persistence diagram $D = \{\boldsymbol{r}_i = (b_i, d_i) \in \Omega \mid 1 \le i \le N(D)\}$ can be equivalently represented as a counting measure $\mu$ on $\Omega$ given by

$$A \in \mathcal{B} \mapsto \mu(A) = \sum_{i=1}^{N(D)} \delta_{\boldsymbol{r}_i}(A),$$

where $\mathcal{B} = \mathcal{B}(\Omega)$ is the class of all Borel subsets of $\Omega$ and $\delta_{\boldsymbol{r}}$ denotes the Dirac point mass at $\boldsymbol{r} \in \Omega$. We will refer to $\mu$ as the *persistence measure* corresponding to $D$ and, with a slight abuse of notation, will treat persistence diagrams as counting measures. If $D$ is a random persistence diagram, then the associated persistence measure is also random. In addition to the persistence measure $\mu$ associated to a persistence diagram $D$, we will also study its *normalized measure* $\tilde{\mu}$, which is the persistence measure divided by the total number of points

$N(D)$ in the persistence diagram. In detail, $\tilde{\mu}$ is the (possibly random) probability measure on $\Omega$ given by

$$A \in \mathcal{B} \mapsto \tilde{\mu}(A) = \frac{1}{N(D)} \sum_{i=1}^{N(D)} \delta_{\boldsymbol{r}_i}(A).$$

The normalized persistence measure may be desirable when the number of points $N(D)$ in the persistence diagram is not of direct interest but their spatial distribution is. This is typically the case when the persistence diagrams at hand contain many points or are obtained from large random filtrations (e.g. the Vietoris-Rips complex built on point clouds), so that the value of $N(D)$ will mostly accounts for noisy topological fluctuations due to sampling.

We will consider the setting in which the observed persistence diagram $D$ is a random draw from an unknown distribution. Then, the (non-random) measures

$$A \in \mathcal{B} \mapsto \mathbb{E}[\mu](A) = \mathbb{E}[\mu(A)] \quad \text{and} \quad A \in \mathcal{B} \mapsto \mathbb{E}[\tilde{\mu}](A) = \mathbb{E}[\tilde{\mu}(A)]$$

are well defined. We will refer to $\mathbb{E}[\mu]$ and $\mathbb{E}[\tilde{\mu}]$ as the *expected persistence measure* and the *expected persistence probability,* respectively. Notice that typically, neither is a discrete measure, and that the expected persistence probability is a probability measure by construction.

The interpretations of the measure $\mathbb{E}[\mu]$ and the probability measure $\mathbb{E}[\tilde{\mu}]$ is straightforward: for any Borel set $A \subset \Omega$, $\mathbb{E}[\mu](A)$ is the expected number of points from the random persistence diagram falling in $A$, while $\mathbb{E}[\tilde{\mu}](A)$ is the probability that a random persistence diagram will intersect $A$. As a result, they are able to directly express the randomness of the distribution of persistence diagram including structural properties of the topological noise. Despite their interpretability, the expected persistence measure and probability are not yet standard concepts in the practice and theory of TDA. As a result, they have not been thoroughly investigated.

**The persistence intensity and density functions and linear representations.** In a recent, important contribution, [CD19] derived conditions – applicable to a wide range to problems – that ensure that the expected persistence measure $\mathbb{E}[\mu]$ and its normalization $\mathbb{E}[\tilde{\mu}]$ both admit densities with respect to the Lebesgue measure on $\Omega$. Specifically, under fairly mild and general conditions detailed in [CD19] there exist measurable functions $p : \Omega \to \mathbb{R}_{\geq 0}$ and $\tilde{p} : \Omega \to \mathbb{R}_{\geq 0}$, such that for any Borel set $A \subset \Omega$,

$$\mathbb{E}[\mu](A) = \int_A p(\boldsymbol{u})\mathrm{d}\boldsymbol{u}, \quad \text{and} \quad \mathbb{E}[\tilde{\mu}](A) = \int_A \tilde{p}(\boldsymbol{u})\mathrm{d}\boldsymbol{u}. \tag{3}$$

In fact, [CD19] provided explicit expressions for $p$ and $\tilde{p}$ (see Section D.5). Notice that, by construction, $\tilde{p}$ integrates to 1 over $\Omega$. We will refer to the functions $p$ and $\tilde{p}$ as the *persistence intensity* and the *persistence density* functions, respectively. We remark that the notion of a persistence intensity function was originally put forward by [CWRW15].

The persistence intensity and density functions "operationalize" the notions of expected persistence measure and expected persistence probability introduced above, allowing to evaluate, for any Borel set $A$, $\mathbb{E}[\mu](A)$ and $\mathbb{E}[\tilde{\mu}](A)$ in a straightforward way as Lebesgue integrals.

The main objective of the paper is to construct estimators $\widehat{p}$ and $\check{p}$ of the persistence intensity $p$ and persistence density $\tilde{p}$, respectively, and to provide high probability error bounds with respect to the $L_\infty$ norm. As we show below in Theorem 3.1, $L_\infty$-consistency for the persistence intensity function is a stronger guarantee than consistency in the $\mathsf{OT}_p$ metric, for any $p < \infty$. Interestingly, we find that estimation of the persistence probability density function is statistically easier, in the sense that uniform estimation error bounds can be obtained for all points in $\Omega$. In contrast, estimating the persistence intensity function becomes progressively more difficult for points near $\partial\Omega$. See Theorem 3.6 below.

**Linear representations of persistence diagrams.** As noted in [CD19], the persistence intensity and density functions are naturally suited to compute the expected value of linear

representations of random persistence diagrams. A linear representation $\Psi$ of the persistence diagram $D = \{\boldsymbol{r}_i = (b_i, d_i) \in \Omega \mid 1 \leq i \leq N(D)\}$ with corresponding persistence measure $\mu$ is a summary statistic of $D$ of the form

$$\Psi(D) = \sum_{i=1}^{N(D)} f(\boldsymbol{r}_i) = \int_\Omega f(\boldsymbol{u})d\mu(\boldsymbol{u}), \tag{4}$$

for a given measurable function $f$ on $\Omega$. (An analogous definition can be given for the normalized persistence measure $\tilde{\mu}$ instead). Then,

$$\mathbb{E}[\Psi(D)] = \int_\Omega f(\boldsymbol{u})d\mathbb{E}[\mu](\boldsymbol{u}) = \int_\Omega f(\boldsymbol{u})p(\boldsymbol{u})d\boldsymbol{u}, \tag{5}$$

where the second identity follows from (3). Linear representations include persistent Betti numbers, persistence surfaces, persistence silhouettes and persistence weighted Gaussian kernels.

The *persistence surface* is an especially popular linear representation introduced by [AEK$^+$17]. In detail, for a *kernel function* $K(\cdot) : \mathbb{R}^2 \to \mathbb{R}_{\geq 0}$ and any $\boldsymbol{x} \in \mathbb{R}^2$, let $K_h(\boldsymbol{x}) = \frac{1}{h^2}K(\frac{\boldsymbol{x}}{h})$, where $h > 0$ is the bandwidth parameter[1]. The persistence surface of a persistence measure $\mu$ is defined as

$$\rho_h(\boldsymbol{u}) = \int_\Omega f(\boldsymbol{\omega})K_h(\boldsymbol{u} - \boldsymbol{\omega})\mathrm{d}\mu(\boldsymbol{\omega}), \tag{6}$$

where $f(\boldsymbol{\omega}) \colon \mathbb{R}^2 \to \mathbb{R}$ is the user-defined *weighting function,* chosen to ensure stability of the representation. Our analysis allows to immediately obtain consistency rates for the expected persistence surface in $L_\infty$ norm, which, for brevity, we present in the supplementary material (see Theorem B.5). Instead we focus on the estimation error the expected Betti numbers.

**Betti and the persistent Betti numbers.** The **Betti number** at scale $x \in [0, L]$ is the number of persistent homologies that are in existence at "time" $x$. Furthermore, the **persistent Betti number** at a certain point $\boldsymbol{x} = (x_1, x_2) \in \Omega$ measures the number of persistent homologies that are born before $x_1$ and die after $x_2$. In our notation, given a persistence diagram $D$ and its associated persistence measure $\mu$, for $x \in [0, L]$ and $\boldsymbol{x} = (x_1, x_2) \in \Omega$, the corresponding Betti number and persistent Betti number are given by

$$\beta_x(D) = \mu(B_x) \quad \text{and} \quad \beta_{\boldsymbol{x}}(D) = \mu(B_{\boldsymbol{x}}),$$

respectively, where $B_x = [0, x) \times (x, L]$ and $B_{\boldsymbol{x}} = [0, x_1) \times (x_2, L]$. Though Betti numbers are among the most prominent and widely used TDA summaries, relatively little is known about the statistical hardness of estimating their expected values when the sample size is fixed and the number of persistence diagrams increases. Our results will yield error bounds of this type. We will also consider normalized versions of the Betti numbers defined using the persistence probability $\tilde{\mu}$ of the persistence diagram:

$$\tilde{\beta}_x(D) = \tilde{\mu}(B_x) \quad \text{and} \quad \tilde{\beta}_{\boldsymbol{x}}(D) = \tilde{\mu}(B_{\boldsymbol{x}}).$$

Notice that, by definition, $\tilde{\beta}_{\boldsymbol{x}}(D) \leq 1$. While their interpretation is not as direct as the Betti numbers computed using persistence diagrams, the expected normalized (persistence) Betti numbers are informative topological summaries while showing favorable statistical properties (see Theorem 3.12 below).

## 3 Main results

### 3.1 The OT distance between measures and $L_\infty$ distance between intensity functions

A popular and, arguably, natural metric for persistence diagrams – and, more generally, locally finite Radon measures such as normalized persistence measures and probabilities – is

---

[1][AEK$^+$17] showed empirically that the bandwidth does not have a major influence on the efficiency of the persistence surface.

the *optimal transport* distance; see, e.g., [DL21]. In detail, for two Radon measures $\mu$ and $\nu$ supported on $\overline{\Omega}$, an *admissible transport* from $\mu$ to $\nu$ is defined as a function $\pi : \overline{\Omega} \times \overline{\Omega} \to \mathbb{R}$, such that for any Borel sets $A, B \subset \overline{\Omega}$,

$$\pi(A \times \overline{\Omega}) = \mu(A), \quad \text{and} \quad \pi(\overline{\Omega} \times B) = \nu(B).$$

Let $\mathsf{adm}(\mu, \nu)$ denote all the admissible transports from $\mu$ to $\nu$. For any $q \in \mathbb{R}^+ \cup \{\infty\}$, the $q$-th order Optimal Transport (OT) distance between $\mu$ and $\nu$ is defined as

$$\mathsf{OT}_q(\mu, \nu) = \left( \inf_{\pi \in \mathsf{adm}(\mu, \nu)} \int_{\overline{\Omega} \times \overline{\Omega}} \|\boldsymbol{x} - \boldsymbol{y}\|_2^q \mathrm{d}\pi(\boldsymbol{x}, \boldsymbol{y}) \right)^{\frac{1}{q}}.$$

When $\mu$ and $\nu$ are persistent diagrams the choice of $q = \infty$ corresponds to the widely-used *bottleneck distance.* The OT distance is widely used for good reasons: by transporting from and to the diagonal $\partial\Omega$, it captures the distance between two measures that have potentially different total masses, taking advantage of the fact that points on the diagonal have arbitrary multiplicity in persistent diagrams. It also proves to be stable with respect to perturbations of the input to TDA algorithms. However, for expected persistent measures with intensity functions with respect to the Lebesgue measure, we will show next that the $L_\infty$ distance between intensity functions provides a tighter control on the difference between two persistent measures. Below, for a real-valued function on $\Omega$, we let $\|f\|_\infty = \sup_{\boldsymbol{x} \in \Omega} |f(\boldsymbol{x})|$ be its $L_\infty$ norm.

**Theorem 3.1** *Let $\mu$, $\nu$ be two expected persistent measures on $\Omega$ with intensity functions $p_\mu$ and $p_\nu$ respectively. Then*

$$\mathsf{OT}_q^q(\mu, \nu) \le \left( \frac{L}{2} \right)^{q+2} \left( \frac{2\sqrt{2}}{q+1} - \frac{2}{q+2} \right) \|p_\mu - p_\nu\|_\infty. \tag{7}$$

*Furthermore, there exists two sequences of expected persistence measures $\{\mu_n\}_{n \in \mathbb{N}}$ and $\{\nu_n\}_{n \in \mathbb{N}}$ with intensity functions $\{p_{\mu_n}\}_{n \in \mathbb{N}}$ and $\{p_{\nu_n}\}_{n \in \mathbb{N}}$ respectively such that, as $n \to \infty$,*

$$\mathsf{OT}_q(\mu_n, \nu_n) \to 0, \quad \text{while} \quad \|p_{\mu_n} - p_{\nu_n}\|_\infty \to \infty.$$

**The bottleneck distance**   For the case $q = \infty$, which yields the bottleneck distance when applied to persistence diagrams, there can be no meaningful upper bound in the form of (7): we show in Section D.1 of the supplementary material that there exist two sequences of measures such that their bottleneck distance converges to a finite number while the $L_\infty$ distance between their intensity functions vanishes. Existing contributions [Pey18, NGK21] also upper bound the optimal transport distance by a Sobolev-type distance between density functions. It is noteworthy that these bounds require, among other things, the measures to have common support and the same total mass, two conditions that are not assumed in Theorem 3.1.

### 3.2   Non-parametric estimation of the persistent intensity and density functions

In this section, we analyze the performance of kernel-based estimators of the persistent intensity function $p(\cdot)$ and the persistent density function $\tilde{p}(\cdot)$. We adopt the setting where we observe $n$ *i.i.d.* persistent measures $\mu_1, \mu_2, \ldots, \mu_n$. The procedures we proposed are directly inspired by kernel density estimators for probability densities traditionally used in the non-parametric statistics literature; see, e.g., [GN21]. Specifically, we consider the following estimator for $p(\cdot)$ and $\tilde{p}(\cdot)$, respectively:

$$\boldsymbol{\omega} \in \mathbb{R}^2 \mapsto \hat{p}_h(\boldsymbol{\omega}) := \frac{1}{n} \sum_{i=1}^n \int_\Omega K_h(\boldsymbol{x} - \boldsymbol{\omega}) \mathrm{d}\mu_i(\boldsymbol{x}); \tag{8a}$$

$$\boldsymbol{\omega} \in \mathbb{R}^2 \mapsto \check{p}_h(\boldsymbol{\omega}) = \frac{1}{n} \sum_{i=1}^n \int_\Omega K_h(\boldsymbol{x} - \boldsymbol{\omega}) \mathrm{d}\tilde{\mu}_i(\boldsymbol{x}), \tag{8b}$$

where $K(\cdot)$ is the *kernel function,* which we assume to satisfy a number of standard regularity conditions used in non-parametric literature, discussed in detail in Section B.2 of the supplementary material.

**Assumptions.** We will impose a number of regularity conditions on the expected persistent measures, the persistence intensity and density functions and the kernel function. Of course, we will assume throughout that both $p$ and $\tilde{p}$ (see (3)) are well-defined as densities with respect to the Lebesgue measure, though we point out that this is not strictly necessary for Theorems 3.6 and 3.9.

Our first assumption of smoothness of both $p$ and $\tilde{p}$ is needed to control the point-wise bias of our estimators and is a standard assumption in non-parametric density estimation.

**Assumption 3.2 (Smoothness)** *The persistence intensity function $p$ and persistence probability density function $\tilde{p}$ are Hölder smooth of the order of $s > 0$ with parameters $L_p$ and $L_{\tilde{p}}$ respectively*[2].

In our next assumption, we impose boundedness conditions on $p$ and $\tilde{p}$, which are needed in order to apply a key concentration inequality for empirical processes.

**Assumption 3.3 (Boundedness)** *For some $q > 0$, let $\bar{p}(\boldsymbol{\omega}) \coloneqq \|\boldsymbol{\omega} - \partial\Omega\|_2^q p(\boldsymbol{\omega})$. Then,*

$$\|\bar{p}\|_\infty = \sup_{\omega \in \Omega} \|\boldsymbol{\omega} - \partial\Omega\|_2^q p(\boldsymbol{\omega}) < \infty \quad and \quad \|\tilde{p}\|_\infty = \sup_{\omega \in \Omega} \tilde{p}(\boldsymbol{\omega}) < \infty.$$

Notice that instead of assuming a bound on the $L_\infty$ norm of the intensity function $p$, we are only requiring the weaker condition that the weighted intensity function $\bar{p}(\boldsymbol{\omega}) = \|\boldsymbol{\omega} - \partial\Omega\|_2^q p(\boldsymbol{\omega})$ has finite $L_\infty$ norm, due to the fact that the total mass of the persistence measure may not be uniformly bounded in a number of common data-generating mechanisms. Indeed, it is not a priori clear that Assumption 3.3 itself is realistic; in the supplementary material we prove that this assumption holds for the Vietoris-Rips filtration built on i.i.d. samples. On the other hand, assuming that the persistence density is uniformly bounded poses no problems. See Theorems B.1 and B.2 in the supplementary material for formal arguments. This fact is the primary reason why the persistence probability density function – unlike the persistence intensity function – can be estimated uniformly well over the entire set $\Omega$ - see (3.6) below. We refer readers to Section B.1 of the supplementary materials for details and a discussion on this subtle but consequential point.

In our last assumption, we require a uniform bound on the $q$-th order total persistence, though not on the total number of points in the persistence diagram. As elucidated in [CSEHM10] and discussed in [DP19] and [DL21], this is a relatively mild assumption, which should be expected to hold under a broad variety of data-generating mechanisms.

**Assumption 3.4 (Bounded total persistence)** *There exists a constant $M > 0$, such that, for the value of $q$ as in Assumption 3.3, it holds that, almost surely,*

$$\max_{i=1,\dots,n} \int_\Omega \|\boldsymbol{\omega} - \partial\Omega\|_2^q \mathrm{d}\mu_i(\omega) < M.$$

We will denote with $\mathcal{Z}_{L,M}^q$ the set of persistent measures on $\Omega_L$ satisfying Assumption 3.4.

We are now ready to present our first result concerning the bias of the kernel estimators, whose proof is relatively standard.

**Theorem 3.5** *Under Assumption 3.2, for any $\boldsymbol{\omega} \in \Omega$,*

$$|\mathbb{E}[\hat{p}_h(\boldsymbol{\omega})] - p(\boldsymbol{\omega})| \le L_p h^s \int_{\|\boldsymbol{v}\|_2 \le 1} K(\boldsymbol{v})\|\boldsymbol{v}\|_2^s \mathrm{d}\boldsymbol{v}, \quad and$$

$$|\mathbb{E}[\check{p}_h(\boldsymbol{\omega})] - \tilde{p}(\boldsymbol{\omega})| \le L_{\tilde{p}} h^s \int_{\|\boldsymbol{v}\|_2 \le 1} K(\boldsymbol{v})\|\boldsymbol{v}\|_2^s \mathrm{d}\boldsymbol{v}.$$

The next result provides high-probability uniform bounds on the fluctuations of the kernel estimators around their expected values.

---

[2]We refer readers to the supplementary material for definitions.

**Theorem 3.6** *Suppose that Assumptions 3.3 and 3.4 hold. Then,*

(a) *there exists a positive constant $C$ depending on $M, \|K\|_\infty, \|K\|_2, \|\bar p\|_\infty$ and $q$ such that for any $\delta \in (0,1)$, it can be guaranteed with probability at least $1 - \delta$ that*

$$\sup_{\boldsymbol{\omega} \in \Omega_{2h}} \ell_{\boldsymbol{\omega}}^q |\hat p_h(\boldsymbol{\omega}) - \mathbb{E}\hat p_h(\boldsymbol{\omega})| \le C \max \left\{ \frac{1}{nh^2} \log \frac{1}{\delta h^2}, \sqrt{\frac{1}{nh^2}} \sqrt{\log \frac{1}{\delta h^2}} \right\},$$

*where $\ell_\omega := \|\boldsymbol{\omega} - \partial\Omega\|_2 - h$;*

(b) *there exists a positive constant $C$ depending on $M, \|K\|_\infty, \|K\|_2, \|\tilde p\|_\infty$ and $q$ such that for any $\delta \in (0,1)$, it can be guaranteed with probability at least $1 - \delta$ that*

$$\sup_{\boldsymbol{\omega} \in \Omega} |\check p_h(\boldsymbol{\omega}) - \mathbb{E}\check p_h(\boldsymbol{\omega})| \le C \max \left\{ \frac{1}{nh^2} \log \frac{1}{\delta h^2}, \sqrt{\frac{1}{nh^2}} \sqrt{\log \frac{1}{\delta h^2}} \right\}.$$

**Remark.** The dependence of the constants on problem related parameters is made explicit in the proofs; see the supplementary material.

There is an important difference between the two bounds in Theorem 3.6: while the variation of $\check p_h(\boldsymbol{\omega})$ is uniformly bounded everywhere on $\Omega$, the variation of $\hat p_h(\boldsymbol{\omega})$ is uniformly bounded only when $\boldsymbol{\omega}$ is at least $2h$ away from the diagonal $\partial\Omega$, and may increase as $\boldsymbol{\omega}$ approaches the diagonal. The difficulty in controlling the variation of $\hat p_h$ near the diagonal comes from the fact that we only assume the total persistence of the persistent measures to be bounded; in other words, the number of points near the diagonal in the sample persistent diagrams can be prohibitively large, since their contribution to the total persistence is negligible. This is to be expected in noisy settings in which the sampling process will result in topological noise consisting of many points in the persistence diagram near the diagonal. The above result suggests that it is advantageous to rely on density-based, instead of intensity-based representations of the persistent measures.

**Bias-variance trade-off and minimax lower bound.** If follows from Theorems 3.5 and 3.6 that the choice $h \asymp n^{-\frac{1}{2(s+1)}}$ for the bandwidth will optimize the bias-variance trade-off, yielding high-probability estimation errors

$$\sup_{\boldsymbol{w} \in \Omega_{2h}} \ell_\omega^q |\hat p_h(\boldsymbol{\omega}) - p(\omega)| \lesssim O\left(n^{-\frac{s}{2(s+1)}}\right), \quad \text{and} \quad \sup_{\boldsymbol{w} \in \Omega} |\check p_h(\boldsymbol{\omega}) - \tilde p(\omega)| \lesssim O\left(n^{-\frac{s}{2(s+1)}}\right).$$

The following theorem shows that the above rate is minimax optimal for the persistence density function. For brevity, we here omit a similar result for the persistence intensity function (see Theorem B.4 in the supplementary material).

**Theorem 3.7** *Let $\mathscr{F}$ denote the set of functions on $\Omega$ with Besov norm bounded by $B > 0$:*
$$\mathscr{F} = \{ f : \Omega \to \mathbb{R}, \|f\|_{B^s_{\infty,\infty}} \le B \}.$$

*Then,*

$$\inf_{\check p_n} \sup_P \mathbb{E}_{\mu_1,\dots,\mu_n \overset{i.i.d.}{\sim} P} \|\check p_n - \tilde p\|_\infty \ge O(n^{-\frac{s}{2(s+1)}}),$$

*where the infimum is taken over estimator $\check p_n$ mapping $\mu_1, \dots, \mu_n$ to an intensity function in $\mathscr{F}$, the supremum is over the set of all probability distributions on $\mathcal{Z}^q_{L,M}$ and $\tilde p$ is the intensity function of $\mathbb{E}_P[\tilde\mu]$.*

### 3.3 Kernel-based estimators for linear functionals of the persistent measure

The kernel estimators (8) can serve as a basis for estimating bounded linear representations of the expected persistence measure $\mathbb{E}[\mu]$ and its normalized counterpart $\mathbb{E}[\tilde\mu]$. Specifically, for $R > 0$, let $\mathscr{F}_{2h,R}$ and $\widetilde{\mathscr{F}}_R$ denote the set of linear representations of the form

$$\mathscr{F}_{2h,R} = \left\{ \Psi = \int_{\Omega_{2h}} f \mathrm{d}\mathbb{E}[\mu] \middle| f : \Omega_{2h} \to \mathbb{R}_{\ge 0}, \int_{\Omega_{2h}} \ell_{\boldsymbol{\omega}}^{-q} f(\boldsymbol{\omega}) \mathrm{d}\boldsymbol{\omega} \le R \right\}, \quad \text{and}$$

$$\widetilde{\mathscr{F}}_R = \left\{ \widetilde\Psi = \int_\Omega f \mathrm{d}\mathbb{E}[\tilde\mu] \middle| f : \Omega \to \mathbb{R}_{\ge 0}, \int_\Omega f(\boldsymbol{\omega}) \mathrm{d}\boldsymbol{\omega} \le R \right\}.$$

Then, any linear representations $\Psi \in \mathscr{F}_{2h,R}$ and $\widetilde{\Psi} \in \widetilde{\mathscr{F}}_R$ can be estimated by

$$\hat{\Psi}_h = \int_{\Omega_{2h}} f(\boldsymbol{\omega})\hat{p}_h(\boldsymbol{\omega})\mathrm{d}\boldsymbol{\omega}, \quad \text{and} \quad \check{\Psi}_h = \int_{\Omega} f(\boldsymbol{\omega})\check{p}_h(\boldsymbol{\omega})\mathrm{d}\boldsymbol{\omega}, \tag{9}$$

respectively. The following theorems provide uniform bounds on the bias and variation of these kernel-based estimators.

**Theorem 3.8** *Under Assumption 3.2, it holds that*

$$\sup_{\Psi \in \mathscr{F}_{2h,R}} \left| \mathbb{E}[\hat{\Psi}_h] - \Psi \right| \leq L_p h^s R \int_{\|\boldsymbol{v}\|_2 \leq 1} K(\boldsymbol{v})\|\boldsymbol{v}\|_2^2 \mathrm{d}\boldsymbol{v}; \quad and$$

$$\sup_{\Psi \in \widetilde{\mathscr{F}}_R} \left| \mathbb{E}[\check{\Psi}_h] - \widetilde{\Psi} \right| \leq L_{\tilde{p}} h^s R \int_{\|\boldsymbol{v}\|_2 \leq 1} K(\boldsymbol{v})\|\boldsymbol{v}\|_2^2 \mathrm{d}\boldsymbol{v}.$$

**Theorem 3.9** *Assume that Assumptions 3.2 and 3.3 hold. Then,*

*(a) there exists a constant $C$ depending on $M, \|K\|_\infty, \|K\|_2, \|\bar{p}\|_\infty$ and $q$ such that for any $\delta \in (0,1)$, it can be guaranteed with probability at least $1 - \delta$ that*

$$\sup_{\Psi \in \mathscr{F}_{2h,R}} \left| \hat{\Psi}_h - \mathbb{E}[\hat{\Psi}_h] \right| \leq CR \cdot \max\left\{ \frac{1}{nh^2}\log\frac{1}{\delta h^2}, \sqrt{\frac{1}{nh^2}}\sqrt{\log\frac{1}{\delta h^2}} \right\};$$

*(b) there exists a constant $C$ depending on $M, \|K\|_\infty, \|K\|_2, \|\tilde{p}\|_\infty$ and $q$ such that for any $\delta \in (0,1)$, it can be guaranteed with probability at least $1 - \delta$ that*

$$\sup_{\Psi \in \widetilde{\mathscr{F}}_R} \left| \check{\Psi}_h - \mathbb{E}[\check{\Psi}_h] \right| \leq CR \cdot \max\left\{ \frac{1}{nh^2}\log\frac{1}{\delta h^2}, \sqrt{\frac{1}{nh^2}}\sqrt{\log\frac{1}{\delta h^2}} \right\}.$$

It is important to highlight the fact that the above bounds hold uniformly over the choice of linear representations under only mild integrability assumptions.

Theorems 3.8 and 3.9 are direct corollaries of Theorems 3.5 and 3.6. We again stress the difference between the two upper bounds of Theorem 3.9: part (a) shows that for a linear functional of the original persistent measure to have controlled variation, we need the field of integral to be at least $2h$ away from the diagonal $\partial\Omega$, a requirement that is not necessary for linear functionals of the normalized persistent measure, as is shown in part (b).

Next, we apply Theorems 3.9 and 3.9(a) to the persistent Betti number, which, for any $\boldsymbol{x} \in \Omega$, can be estimated by

$$\hat{\beta}_{\boldsymbol{x},h} = \int_{B_{\boldsymbol{x}}} \hat{p}_h(\boldsymbol{\omega})\mathrm{d}\boldsymbol{\omega}. \tag{10}$$

**Corollary 3.10** *Under Assumption 3.2, it holds that*

$$\sup_{\boldsymbol{x} \in \Omega} \left| \mathbb{E}[\hat{\beta}_{\boldsymbol{x},h}] - \beta_{\boldsymbol{x}} \right| \leq L_p h^s \frac{L^2}{4} \int_{\|\boldsymbol{v}\|_2 \leq 1} K(\boldsymbol{v})\|\boldsymbol{v}\|_2^2 \mathrm{d}\boldsymbol{v}.$$

**Corollary 3.11** *Under Assumptions 3.2 and 3.3(a), there exists a constant $C$ depending on $M, \|K\|_\infty, \|K\|_2, \|\bar{p}\|_\infty$ and $q > 2$ such that for any $\delta \in (0,1)$,*

$$\sup_{\boldsymbol{x} \in \Omega : \ell_{\boldsymbol{x}} > h} \ell_{\boldsymbol{x}}^{q-2} \left| \hat{\beta}_{\boldsymbol{x},h} - \mathbb{E}[\hat{\beta}_{\boldsymbol{x},h}] \right| \leq C \max\left\{ \frac{1}{nh^2}\log\frac{1}{\delta h^2}, \sqrt{\frac{1}{nh^2}}\sqrt{\log\frac{1}{\delta h^2}} \right\}$$

*holds with probability at least $1 - \delta$.*

Notice that in order for the variation of $\hat{\beta}_{\boldsymbol{x},h}$ to be bounded, we need $\boldsymbol{x}$ to be at least $2h$ away from the diagonal $\partial\Omega$, and that the upper bound for the variation increases as $\boldsymbol{x}$

approaches the diagonal. Therefore, based on our analysis, the kernel-based estimator $\hat{p}_h$ *will not be guaranteed to* yield a stable estimation of the Betti number $\beta_x$. As remarked above, this issue arises as the intensity function may not be uniformly bounded near the diagonal. Indeed, in the supplementary material, we describe an alternative proof technique based on an extension of the standard VC inequality and arrive at a very similar rate.

If instead we target the *normalized Betti numbers* $\tilde{\beta}_{\boldsymbol{x}}$, this issue disappears when we deploy the analogous estimator $\check{\beta}_{x,h} = \int_{B_x} \check{p}_h(\boldsymbol{\omega})\mathrm{d}\boldsymbol{\omega}$, constructed using $\check{p}_h$. Indeed, Theorem 3.9(b) leads to the following uniform bounds.

**Corollary 3.12** *Assume that Assumptions 3.2 and 3.3 hold true. Then there exist a constant $C > 0$ depending on $M, \|K\|_\infty, \|K\|_2, \|\tilde{p}\|_\infty$ and $q$ such that for any $\delta \in (0,1)$, it can be guaranteed with probability at least $1 - \delta$ that*

$$\sup_{\boldsymbol{x} \in \Omega} \left| \check{\beta}_{\boldsymbol{x},h} - \mathbb{E}[\check{\beta}_{\boldsymbol{x},h}] \right| \leq \frac{CL^2}{4} \max \left\{ \frac{1}{nh^2} \log \frac{1}{\delta h^2}, \sqrt{\frac{1}{nh^2}} \sqrt{\log \frac{1}{\delta h^2}} \right\}$$

As a direct consequence of the previous result, we obtain a uniform error bound for the *expected normalized Betti curve*, i.e.

$$\sup_{x \in (0,L)} \left| \check{\beta}_{x,h} - \mathbb{E}[\check{\beta}_{x,h}] \right| \leq \frac{CL^2}{4} \max \left\{ \frac{1}{nh^2} \log \frac{1}{\delta h^2}, \sqrt{\frac{1}{nh^2}} \sqrt{\log \frac{1}{\delta h^2}} \right\},$$

To the best of our knowledge this is the first result of this kind, as typically one can only establish pointwise and not uniform consistency of Betti numbers.

## 4 Numerical Illustration and discussion

To illustrate our methodology and highlight the differences between the persistence intensity and density functions, we consider the MNIST handwritten digits dataset and the ORBIT5K dataset. The ORBIT5K dataset contains independent simulations for the linked twist map, dynamical systems for fluid flow as described in [AEK+17]; see also Appendix G.2 of [KKZ+20]. In Section E of the supplementary material, we show the estimated persistence intensity and density functions computed from persistence diagrams obtained over a varying number of random samples from the ORBIT5K datasets, for different model parameters. The figures confirm our theoretical finding that the values of the persistence density function near the diagonal are not as high (on a relative scale) as those of the persistence intensity function. An analogous conclusion can be reached when inspecting the persistence intensity and density functions for different draws of the MNIST datasets for the digits 4 and 8. We further include plots of the average Betti and normalized Betti curves from the ORBIT5K dataset, along with the curves of the empirical point-wise 5% and 95% quantiles. These plots reveal the different scales of the Betti curves and normalized Betti curves, and of their uncertainty.

In this paper, we have taken the first step towards developing a new set of methods and theories for statistical inference for TDA based on samples of persistence diagrams. Our main focus is on the estimation of the persistence intensity function [CD19, CWRW15], a TDA summary of a functional type that encodes the entire distribution of a random persistence diagram and is naturally suited to handle linear representations. We have analyzed a simple kernel estimator and derived uniform consistency rates that hold under very mild assumptions. We also propose the persistence density function, a novel functional TDA summary that enjoys stronger statistical guarantees. Though our results guarantee that the proposed estimators are consistent, in order to carry out statistical inference, it is necessary to develop more sophisticated procedures that quantify the uncertainty of our estimators. Towards that goal, it would be interesting to develop bootstrap-based methods for constructing confidence bands for both the persistence intensity and density functions.

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
