# Supplementary material: On the estimation of persistence intensity functions and linear representations of persistence diagrams

# Contents

Submitted to 37th Conference on Neural Information Processing Systems (NeurIPS 2023). Do not distribute.

**Notation.** We use boldface small letters like $\boldsymbol{u}, \boldsymbol{x}, \boldsymbol{\omega}$ to denote points in $\mathbb{R}^2$ and sub-scripted letters like $x_1, x_2$ to denote their entries. Boldface capital letters like $\boldsymbol{X}, \boldsymbol{Y}$ would be used to denote points on a Riemann manifold. For any positive integer $n$, the symbol $[n]$ refers to the set of all positive integers no larger than $n$. For any set $S$, the symbol $2^S$ represents the power set of $S$, which contains all subsets of $S$ as its elements. The set of all non-negative real numbers would be denoted as $\mathbb{R}_{\geq 0}$. For any function $f$ with domain $\mathcal{A}$, the infinity norm of $f$ is denoted as $\|f\|_\infty := \sup_{x \in \mathcal{A}} |f(x)|$.

# A  Background: The persistence diagram

In this section, we give a brief introduction to the persistence diagram. We refer readers to [CD19] for a detailed description. Consider a random point cloud $\boldsymbol{X} = (\boldsymbol{X}_1, \boldsymbol{X}_2, \ldots, \boldsymbol{X}_N) \in \mathcal{M}^N$ where $\mathcal{M}$ is a Riemann manifold; and a *filtering function* $\varphi : 2^{[N]} \times \mathcal{M}^N \to \mathbb{R}$, which satisfies

$$\varphi(J, \boldsymbol{X}) \leq \varphi(J', \boldsymbol{X}), \quad \forall J \subset J' \in 2^{[N]}, \boldsymbol{X} \in \mathcal{M}^N.$$

A simplicial complex given $\boldsymbol{X}$ and $\varphi$ at level $\alpha$ is defined as

$$K_\alpha(\boldsymbol{X}, \varphi) = \{J \subset 2^{[N]} \mid \varphi(J, \boldsymbol{X}) \leq \alpha\}.$$

Two common examples are the *Cech complex*, where $\varphi(J, \boldsymbol{X})$ equals the radius of the circumscribed ball of $\boldsymbol{X}[J]$; and the *Vietoris-Rips complex*, where $\varphi[J, \boldsymbol{X}]$ is chosen as the maximum distance between points in $\boldsymbol{X}[J]$.

Throughout the paper, we assume that the filtering function $\varphi$ takes its value in $[0, L]$. For all values $\alpha \in [0, L]$, the sequence of simplicial complexes $\{K_\alpha(\boldsymbol{X}, \varphi)\}_{\alpha \in [0,L]}$ forms a *filtration* denoted as $\mathcal{F}(\boldsymbol{X}, \varphi)$, where $K_\alpha(\boldsymbol{X}, \varphi) \subseteq K_{\alpha'}(\boldsymbol{X}, \varphi)$ whenever $\alpha \leq \alpha'$.

*Persistent homology* is a method for computing topological features of a simplicial complex, and can be represented by the *persistence diagram*. In the filtration $\mathcal{F}(\boldsymbol{X}, \varphi)$, for any persistent homology that begins to appear at level $b$ and disappears at level $d$, we say that the homology is *born* at $b$ and *dies* at $d$. With $\Omega$ defined as in (1), the persistence diagram of the point cloud $\boldsymbol{X}$ is a multiset on $\Omega$ that summarizes the birth and death times of all persistent homologies in the filtration $\mathcal{F}(\boldsymbol{X}, \varphi)$:

$$\mathsf{Dgm}(\boldsymbol{X}, \varphi) = \{(b_i, d_i) : \text{the } i\text{-th persistent homology in } \mathcal{F}(\boldsymbol{X}, \varphi)$$
$$\text{that is born at } b_i \text{ and dies at } d_i\}.$$

# B  Supportive theoretical results

## B.1  Validation of Assumption 3.3

In this part, we provide some common data-generating mechanisms where Assumption 3.3 can be validated.

**Theorem B.1** *Let $q, d$ be two positive integers and $q > d$. Let $\kappa$ be a density on $[0, 1]^d$ such that $0 < \inf \kappa \leq \sup \kappa < \infty$. Suppose that $\boldsymbol{X}_N$ be either a binomial process with parameters $N$ and $\kappa$ or a Poisson process of intensity $N\kappa$ in the cube $[0, 1]^d$. Denote $p(\boldsymbol{u})$ as the intensity function for the $k$-dimensional expected persistent measure induced by the Vietoris-Rips filtration. Then when $N$ is sufficiently large, for $\boldsymbol{u} \in \Omega$, there exists a polynomial function $\mathsf{poly}(\cdot)$, such that*

$$p(\boldsymbol{u}) \leq \mathsf{poly}(N, d) \sup \kappa,$$

*and $\overline{p}(\boldsymbol{u})$ can be correspondingly bounded.*

**Theorem B.2** *Let $q, d$ be two positive integers and $q > d$. Let $\kappa$ be a density on $[0, 1]^{d \times N}$ such that $0 < \inf \kappa < \sup \kappa < \infty$. Suppose that $\boldsymbol{X}_1, \boldsymbol{X}_2, \ldots, \boldsymbol{X}_N \in [0, 1]^d$ and that $\boldsymbol{X} = (\boldsymbol{X}_1, \boldsymbol{X}_2, \ldots, \boldsymbol{X}_N) \sim \kappa$. Denote $\tilde{p}(\boldsymbol{u})$ as the persistence density induced by the Vietoris-Rips filtration of $\boldsymbol{X}$. Then there exists a polynomial function $(\cdot)$, such that*

$$\tilde{p}(\boldsymbol{u}) \leq \mathsf{poly}(N, d) \sup \kappa.$$

## B.2  Clarification of Assumptions

In this part, we provide the details in the smoothness assumption of the persistence intensity and density functions, and the regularization assumptions of the kernel function.

**Hölder smoothness.** Recall from Assumption 3.2 that we assume the persistence intensity function $p(\cdot)$ and the persistence density function $\tilde{p}(\cdot)$ are Hölder smooth. A function $f : \Omega \to \mathbb{R}_{\geq 0}$ is $s$-th order Hölder smooth with parameter $L_f$ if it is at least $(s-1)$-differentiable and that for any $\boldsymbol{x}, \boldsymbol{x}' \in \Omega$,

$$\left| f(\boldsymbol{x}') - f(\boldsymbol{x}) - \sum_{t=1}^{s-1} \frac{1}{t!} \sum_{t_1+t_2=t} \frac{\mathrm{d}^t f(\boldsymbol{x})}{\mathrm{d}x_1^{t_1}\mathrm{d}x_2^{t_2}} (x_1' - x_1)^{t_1}(x_2' - x_2)^{t_2} \right| \leq L_f \|\boldsymbol{x}' - \boldsymbol{x}\|_2^s. \quad (1)$$

**Assumptions regarding the kernel function.** Throughout the paper, we assume the kernel function $K(\cdot)$ satisfies some properties that are commonly used in non-parametric statistics [GN21]. Specifically, we make the following assumption.

**Assumption B.3** *The kernel function $K : \mathbb{R}^2 \to \mathbb{R}$ satisfies the following conditions:*

*(a) $K(\boldsymbol{x}) = 0$ for all $\boldsymbol{x}$ with $\|\boldsymbol{x}\|_2 > 1$;*

*(b) $\|K\|_\infty := \sup_{\boldsymbol{x}} |K(\boldsymbol{x})| < \infty$;*

*(c) $\int_{\mathbb{R}^2} K(\boldsymbol{x})\mathrm{d}\boldsymbol{x} = 1$;*

*(d) $\|K\|_2^2 := \int_{\mathbb{R}^2} K^2(\boldsymbol{x})\mathrm{d}\boldsymbol{x} < \infty$.*

*(e) There exists a positive integer $s$, such that for all non-negative integers $s_1, s_2$ satisfying $1 \leq s_1 + s_2 < s$,*

$$\int_{\boldsymbol{x} \in \mathbb{R}^2} x_1^{s_1} x_2^{s_2} K(\boldsymbol{x})\mathrm{d}\boldsymbol{x} = 0.$$

*(f) $K$ is $L_K$-Lipchitz with respect to the $\ell_2$ norm on $\mathbb{R}^2$.*

## B.3 Minimax lower bound for estimating the persistence intensity function

Below we provide a minimax lower bound on the $L_\infty$ estimation error of the persistence intensity function by levering well-known minimax arguments for estimating a smooth probability density function based on an i.i.d. sample; see [GN21] for details, as well for the definition of Besov norms.

**Theorem B.4** *Let $\mathscr{F}$ denote the set of functions on $\Omega$ with Besov norm bounded by $B > 0$:*

$$\mathscr{F} = \{f : \Omega \to \mathbb{R}, \|f\|_{B_{\infty,\infty}^s} \leq B\}.$$

*Then,*

$$\inf_{\hat{p}_n} \sup_{P} \mathbb{E}_{\mu_1,\ldots,\mu_n \overset{i.i.d.}{\sim} P} \sup_{\boldsymbol{\omega} \in \Omega} \|\boldsymbol{\omega} - \partial\Omega\|_2^q |\hat{p}_n(\boldsymbol{\omega}) - p(\boldsymbol{\omega})| \geq O(n^{-\frac{s}{2(s+1)}}),$$

*where the infimum is taken over estimator $\hat{p}_n$ mapping $\mu_1, \ldots, \mu_n$ to an intensity function in $\mathscr{F}$, the supremum is over the set of all probability distribution on $\mathcal{Z}_{L,M}^q$ and $p$ is the intensity function of $\mathbb{E}_P[\mu]$.*

## B.4 Estimating the persistence surface

For estimating the persistence surface in (6), we directly generate the persistence surface from the empirical averaged persistence measure $\bar{\mu}_n$ given by

$$A \in \mathcal{B} \mapsto \bar{\mu}_n(A) = \frac{1}{n} \sum_{i=1}^{n} \mu_i(A).$$

Since $\bar{\mu}_n$ is unbiased for $\mathbb{E}[\mu]$ and $\rho$ is a linear transformation, $\rho_h(\bar{\mu}_n)$ is also unbiased for $\rho_h(\mathbb{E}[\mu])$. The following theorem bounds its variation.

92 **Theorem B.5** *With the choice of the weight function*

$$f(\boldsymbol{\omega}) = \|\boldsymbol{\omega} - \partial\Omega\|_2^q,$$

93 *when Assumptions 3.3(a) and 3.4 hold true, there exists a constant $C$ depending on $L, M, L_K, \|K\|_\infty$*
94 *and $\|\bar{p}\|_\infty$, such that for any $\delta \in (0,1)$, it can be guaranteed with probability at least $1 - \delta$ that*

$$\|\rho_h(\bar{\mu}_n) - \rho_h(\mathbb{E}[\mu])\|_\infty \leq C \max\left\{ \frac{1}{nh^2}\log\frac{1}{\delta h^2}, \sqrt{\frac{1}{nh^2}}\sqrt{\log\frac{1}{\delta h^2}} \right\}.$$

95 ## B.5   Estimating the persistent betti number by the empirical averaged persistence measure

96 As an alternative to the kernel-based estimator for the persistent betti number in (10), we can directly
97 use the empirical persistent betti number as the estimator:

$$\bar{\beta}_{\boldsymbol{x}} = \bar{\mu}_n(B_{\boldsymbol{x}}).$$

98 Since $\bar{\mu}_n$ is an unbiased estimator for $\mathbb{E}[\mu]$, $\bar{\beta}_{\boldsymbol{x}}$ is an unbiased estimator for $\beta_{\boldsymbol{x}}$. As for the variation
99 of the estimator, we provide the following theorem.

100 **Theorem B.6** *Under Assumptions 3.2, 3.3(a) and 3.4, for any $\delta \in (0,1)$, there exists a universal*
101 *constant $C$ such that with probability at least $1 - \delta$, it can be guaranteed that*

$$\sup_{\boldsymbol{x}\in\Omega_\ell} |\bar{\beta}_{\boldsymbol{x}} - \beta_{\boldsymbol{x}}| \leq C \left( \frac{M\ell^{-q}}{n}\left(2\log(M\ell^{-q}n + 1) + \log\frac{1}{\delta}\right) \right.$$

$$\left. + \sqrt{\min\left\{ \frac{M^2\ell^{-2q}}{n}, \frac{\sqrt{2}ML\ell^{1-2q}\|\bar{p}\|_\infty}{(q-1)_+ n} \right\}\left( \sqrt{2\log(M\ell^{-q}n + 1)} + \sqrt{\log\frac{1}{\delta}} \right)} \right),$$

102 *where $(q-1)_+ = \max\{q-1, 0\}$.*

103 # C   Preliminary facts

104 In this section we present and prove various auxiliary results that are needed in the proofs of the main
105 theorems.

106 ## C.1   Preliminary facts for the proof of Theorem B.1

107 Bounding the weighted intensity function as in Theorem B.1 requires a detailed exploration of the
108 persistent diagram for the Vietoris-Rips filtration. Throughout this section, we will consider the
109 filtering function corresponding to the Vietoris-Rips filtration

$$\varphi[J](\boldsymbol{X}) = \min_{i,j\in J, i\neq j} \|\boldsymbol{X}_i - \boldsymbol{X}_j\|_2.$$

110 Firstly, we state a form of the **area formula** given by [Mor16], which would be useful for a change
111 of variable in deriving the intensity function for the expected persistence measure.

112 **Theorem C.1** *Denote $\mathscr{L}^M$ as the $M$-dimensional Lebesgue measure and $\mathscr{H}^M$ as the $M$-*
113 *dimensional Hausdorff measure. Consider a Lipchitz function $f : \mathbb{R}^M \to \mathbb{R}^N$ for $M \leq N$. If*
114 *$h : \mathbb{R}^M \to \mathbb{R}$ is an $\mathscr{L}^M$-integrable function, then*

$$\int_{\mathbb{R}^M} h(\boldsymbol{X}) J_{\boldsymbol{X}} f(\boldsymbol{X}) \mathrm{d}\mathscr{L}^M(\boldsymbol{X}) = \int_{\mathbb{R}^N} \sum_{\boldsymbol{X}\in f^{-1}\{\boldsymbol{Y}\}} h(\boldsymbol{X}) \mathrm{d}\mathscr{H}^M \boldsymbol{Y},$$

115 *where $J_{\boldsymbol{X}} f(\boldsymbol{X})$ is the Jacobian determinant of the function $f$:*

$$J_{\boldsymbol{X}} f(\boldsymbol{X}) = \sqrt{\det\left( \left(\frac{\mathrm{d}f}{\mathrm{d}\boldsymbol{X}}\right)^\top \left(\frac{\mathrm{d}f}{\mathrm{d}\boldsymbol{X}}\right) \right)}.$$

116   Theorem C.1 directly implies the following corollary, the proof of which would be omitted.

117   **Corollary C.2** *Let $\psi : \mathbb{R}^M \to \mathbb{R}^N$ be a Lipchitz bijection with $M \leq N$, and $\kappa : \mathbb{R}^N \to \mathbb{R}$ be a*
118   *function which satisfies that $h := \kappa \circ \psi$ is $\mathscr{L}^M$-integrable. Then*

$$\int_{\mathbb{R}^M} \kappa \circ \psi(\boldsymbol{X}) J_{\boldsymbol{X}} \psi(\boldsymbol{X}) \mathrm{d}\mathscr{L}^M(\boldsymbol{X}) = \int_{\mathbb{R}^N} \kappa(\boldsymbol{Y}) \mathrm{d}\mathscr{H}^M(\boldsymbol{Y}).$$

119   The following proposition considers two kinds of partitions of the unit cube $[0,1]^{d \times N}$, with each part
120   satisfying some desired properties.

121   **Proposition C.3** *There exists a set $S$ with cardinality $card(S) = 4d^2$, such that for any $J_1, J_2 \subset [N]$*
122   *that satisfies $J_1 \neq J_2, |J_1| = |J_2| = 2$, bearing a zero-measured set, $[0,1]^{d \times n}$ can be partitioned as*

$$[0,1]^{d \times n} = \bigcup_{s \in S} W^s_{J_1, J_2},$$

123   *such that within each part $W^s_{J_1, J_2}$, there exists a diffeomorphism $\Psi^s_{J_1, J_2} : W^s_{J_1, J_2} \to \mathbb{R}^2 \times [0,1]^{nd-2}$,*
124   *such that:*

125       *1. For every $\boldsymbol{X} \in W^s_{J_1, J_2}$, $\Psi^s_{J_1, J_2}(\boldsymbol{X})_1 = \varphi[J_1](\boldsymbol{X})$ and $\Psi^s_{J_1, J_2}(\boldsymbol{X})_2 = \varphi[J_2](\boldsymbol{X})$;*

126       *2. The Jacobian determinant $J_{\boldsymbol{X}} \Psi^s_{J_1, J_2}(\boldsymbol{X}) \geq \frac{1}{d}$.*

127   *Proof*: Let $S = [d]^2 \times \{-1, +1\}^2$, then it is easy to see that $|S| = 4d^2$. For any $J_1, J_2 \subset [n]$ with
128   $J_1 \neq J_2$ and $|J_1| = |J_2| = 2$, let denote $J_1 = \{i_1, j_1\}, J_2 = \{i_2, j_2\}$ with $j_2 = \max\{j \in J_2 : j \notin$
129   $J_1\}$. For any $s = (k_1, k_2, s_1, s_2) \in S$, let

$$W^s_{J_1, J_2} = \{X : \{k_1\} = \operatorname{argmax}_k |X^k_{i_1} - X^k_{j_1}|, s_1(X^k_{j_1} - X^k_{i_1}) > 0,$$
$$\{k_2\} = \operatorname{argmax}_k |X^k_{i_2} - X^k_{j_2}|, s_2(X^k_{j_2} - X^k_{i_2}) > 0.\}$$

130   Notice here that $\{k_1\} = \operatorname{argmax}_k |X^k_{i_1} - X^k_{j_1}|$ means $k_1$ is the *only* index for $|X^k_{i_1} - X^k_{j_1}|$ to reach its
131   maximum.

132   We begin by proving that $\{W^s_{J_1, J_2}\}_{s \in S}$ forms a partition of $[0,1]^{d \times n}$ bearing a zero-measured set.
133   Firstly, for $s, s' \in S$ with $s \neq s'$, it is easy to see that $W^s_{J_1, J_2}$ and $W^{s'}_{J_1, J_2}$ are disjoint. Secondly, if

$$\boldsymbol{X} \in [0,1]^{d \times n} - \bigcup_{s \in S} W^s_{J_1, J_2},$$

134   then by definition, there exists $k, k' \in [d]$, such that $k \neq k'$ and that either

$$|X^k_{j_1} - X^k_{i_1}| = |X^{k'}_{j_1} - X^{k'}_{i_1}|$$

135   or

$$|X^k_{j_2} - X^k_{i_2}| = |X^{k'}_{j_2} - X^{k'}_{i_2}|.$$

136   Notice that for any $k, k' \in [d]$ with $k \neq k'$, the set

$$\left\{ \boldsymbol{X} : |X^k_{j_1} - X^k_{i_1}| = |X^{k'}_{j_1} - X^{k'}_{i_1}| \right\}$$
$$= \left\{ \boldsymbol{X} : X^k_{j_1} - X^k_{i_1} = |X^{k'}_{j_1} - X^{k'}_{i_1}| \right\} \cup \left\{ \boldsymbol{X} : X^k_{j_1} - X^k_{i_1} = -|X^{k'}_{j_1} - X^{k'}_{i_1}| \right\},$$

137   where the sets

$$\left\{ \boldsymbol{X} \in [0,1]^{d \times n} : X^k_{j_1} - X^k_{i_1} = |X^{k'}_{j_1} - X^{k'}_{i_1}| \right\} \quad \text{and}$$
$$\left\{ \boldsymbol{X} \in [0,1]^{d \times n} : X^k_{j_1} - X^k_{i_1} = -|X^{k'}_{j_1} - X^{k'}_{i_1}| \right\}$$

138   are a subsets of $(nd - 1)$ dimensional linear manifolds in $[0,1]^{d \times n}$, and are therefore zero-measured
139   in $\mathscr{L}^{nd}$. Similarly, we can prove that the set $[0,1]^{d \times n} - \bigcup_{s \in S} W^s_{J_1, J_2}$ is the union of a finite number
140   of subsets of $(nd - 1)$ dimensional linear manifolds in $[0,1]^{d \times n}$. Consequently,

$$\bigcup_{s \in S} W^s_{J_1, J_2}$$

141    is a partition of $[0, 1]^{d \times n}$ bearing a zero-measured set.

142    Furthermore, define $\Psi^s_{J_1, J_2}$ as

$$\Psi^s_{J_1, J_2}(\boldsymbol{X}) = \left( \varphi[J_1](\boldsymbol{X}), \varphi[J_2](\boldsymbol{X}), \{X^k_j\}_{\substack{1 \leq j \leq n \\ 1 \leq k \leq d \\ (j,k) \neq (j_1, k_1) \\ (j,k) \neq (j_2, k_2)}} \right), \quad \forall \boldsymbol{X} \in W^s_{J_1, J_2}.$$

143    Then we can firstly notice that

$$X^{k_1}_{j_1} = s_1 \sqrt{u^2_1 - \sum_{k \neq k_1} \left( X^k_{j_1} \right)^2} + X^{k_1}_{i_1} \quad \text{and}$$

$$X^{k_2}_{j_2} = s_2 \sqrt{u^2_2 - \sum_{k \neq k_2} \left( X^k_{j_2} \right)^2} + X^{k_2}_{i_2},$$

144    for $u_1 = \varphi[J_1](X)$ and $u_2 = \varphi[J_2](X)$. This validates $\Psi^s_{J_1, J_2}$ as a diffeomorphism. The proof now
145    boils down to bounding the Jacobian of $\Psi^s_{J_1, J_2}$. Towards this end, notice that the partial derivative of
146    $\varphi$ is bounded by

$$\left| \frac{\partial \varphi[J_1](\boldsymbol{X})}{\partial X^{k_1}_{j_1}} \right| = \left| \frac{\partial}{\partial X^{k_1}_{j_1}} \sqrt{\sum_{k=1}^{d} (X^k_{i_1} - X^k_{j_1})^2} \right|$$

$$= \left| \frac{X^{k_1}_{j_1} - X^{k_1}_{i_1}}{\sqrt{\sum_{k=1}^{d} (X^k_{i_1} - X^k_{j_1})^2}} \right|$$

$$\geq \frac{1}{\sqrt{d}},$$

147    where in the last line we applied the fact that

$$\left| X^{k_1}_{j_1} - X^{k_1}_{i_1} \right| = \max_{1 \leq k \leq d} \left| X^k_{j_1} - X^k_{i_1} \right| \geq \sqrt{\frac{1}{d} \sum_{k=1}^{d} (X^k_{i_1} - X^k_{j_1})^2}.$$

148    Similarly,

$$\left| \frac{\partial \varphi[J_2](\boldsymbol{X})}{\partial X^{k_2}_{j_2}} \right| = \left| \frac{\partial}{\partial X^{k_2}_{j_2}} \sqrt{\sum_{k=1}^{d} (X^k_{i_2} - X^k_{j_2})^2} \right| \geq \frac{1}{\sqrt{d}}.$$

149    Furthermore, since $j_2 \notin J_1$, it is easy to see that

$$\frac{\partial \varphi[J_1](\boldsymbol{X})}{\partial X^{k_2}_{j_2}} = 0.$$

150    Therefore, the Jacobian determinant of $\Psi^s_{J_1, J_2}$ is bounded by

$$J_{\boldsymbol{X}} \Psi^s_{J_1, J_2}(\boldsymbol{X}) = \left| \det \left( \frac{\mathrm{d}\Psi^s_{J_1, J_2}(\boldsymbol{X})}{\mathrm{d}\boldsymbol{X}} \right) \right|$$

$$= \left| \det \left( \begin{pmatrix} \mathbf{I}_{nd-2} & \mathbf{0}_{(nd-2) \times 1} & \mathbf{0}_{(nd-2) \times 1} \\ \mathbf{0}_{1 \times (nd-2)} & \frac{\partial \varphi[J_1](\boldsymbol{X})}{\partial X^{k_1}_{j_1}} & \frac{\partial \varphi[J_1](\boldsymbol{X})}{\partial X^{k_2}_{j_2}} \\ \mathbf{0}_{1 \times (nd-2)} & \frac{\partial \varphi[J_2](\boldsymbol{X})}{\partial X^{k_1}_{j_1}} & \frac{\partial \varphi[J_2](\boldsymbol{X})}{\partial X^{k_2}_{j_2}} \end{pmatrix} \right) \right|$$

$$= \left| \frac{\partial \varphi[J_1](\boldsymbol{X})}{\partial X^{k_1}_{j_1}} \cdot \frac{\partial \varphi[J_2](\boldsymbol{X})}{\partial X^{k_2}_{j_2}} \right| \geq \frac{1}{d}.$$

151    This completes the proof.                               ■

152    The following is important for representing of the persistence intensity function $p$ and the persistence
153    density function $\tilde{p}$.

**154** **Proposition C.4** *Bearing a zero-measured set, $[0,1]^{d \times n}$ can be partitioned as*

$$[0,1]^{d \times n} = \bigcup_{r=1}^{R} V_r,$$

**155** *such that*

**156**      *1. For every $\boldsymbol{X}, \boldsymbol{X}' \in V_r$, $J_1, J_2 \subset [n]$ with $|J_1| = |J_2| = 2$, it is guaranteed that $\varphi[J_1](\boldsymbol{X}) \neq$*
**157**      *$\varphi[J_2](\boldsymbol{X})$; furthermore, if $\varphi[J_1](\boldsymbol{X}) < \varphi[J_2](\boldsymbol{X})$, then $\varphi[J_1](\boldsymbol{X}') < \varphi[J_2](\boldsymbol{X}')$;*

**158**      *2. For every $\boldsymbol{X}, \boldsymbol{X} \in V_r$, $J_1, J_2, J_3, J_4 \subset [n]$ with $|J_1| = |J_2| = |J_3| = |J_4| = 2$,*
**159**      *it is guaranteed that $\varphi[J_1](\boldsymbol{X}) - \varphi[J_2](\boldsymbol{X}) \neq \varphi[J_3](\boldsymbol{X}) - \varphi[J_4](\boldsymbol{X})$; furthermore, if*
**160**      *$\varphi[J_1](\boldsymbol{X}) - \varphi[J_2](\boldsymbol{X}) > \varphi[J_3](\boldsymbol{X}) - \varphi[J_4](\boldsymbol{X}) > 0$, then $\varphi[J_1](\boldsymbol{X}') - \varphi[J_2](\boldsymbol{X}') >$*
**161**      *$\varphi[J_3](\boldsymbol{X}') - \varphi[J_4](\boldsymbol{X}) > 0$.*

**162**      *3. For every $r \in [R]$ and $\boldsymbol{X} \in V_r$, there are $N_r$ points in $\mathrm{Dgm}(\boldsymbol{X}, \varphi)$; furthermore, all these*
**163**      *points can be ordered by their orthogonal distance to the diagonal, and the order is fixed for*
**164**      *all $\boldsymbol{X} \in V_r$.*

**165** *Furthermore, the expected persistence measure $\mathbb{E}[\mu]$ and its normalized counterpart $\mathbb{E}[\tilde{\mu}]$ can be*
**166** *characterized such that for any Borel set $B \subset \Omega$,*

$$\mathbb{E}[\mu](B) = \sum_{r=1}^{R} \sum_{i=1}^{N_r} \int_{x \in \Phi^{-1}[J_{ir}^1, J_{ir}^2](B) \cap V_r} \kappa(\boldsymbol{X}) \mathrm{d}\boldsymbol{X} \quad and$$

$$\mathbb{E}[\tilde{\mu}](B) = \sum_{r=1}^{R} \frac{1}{N_r} \sum_{i=1}^{N_r} \int_{x \in \Phi^{-1}[J_{ir}^1, J_{ir}^2](B) \cap V_r} \kappa(\boldsymbol{X}) \mathrm{d}\boldsymbol{X}$$

**167** *, in which*

$$\Phi[J_1, J_2](\boldsymbol{X}) = (\varphi[J_1](\boldsymbol{X}), \varphi[J_2](\boldsymbol{X})),$$

**168** *and $J_{ir}^1, J_{ir}^2$ are the simplicial complexes corresponding to the birth and death of the $i$-th persistence*
**169** *homology for all $\boldsymbol{X} \in V_r$.*

**170** *Proof:* For simplicity, we only give a sketch of the proof for this proposition. A weaker version
**171** of this proposition is proved in [CD19], where the second property of the partition is not required.
**172** Therefore, the partition we aim to construct here is a refinement of the partition given in [CD19]. In
**173** order to see that the second condition can be reached, we firstly prove that the set

$$A = \big\{ \boldsymbol{X} \in [0,1]^{d \times n} : \exists J_1, J_2, J_3, J_4 \subset [n], \text{ s.t.}$$
$$|J_1| = |J_2| = |J_3| = |J_4| = 2,$$
$$J_1 \neq J_2, J_3 \neq J_4, (J_1, J_2) \neq (J_3, J_4),$$
$$\varphi[J_1](\boldsymbol{X}) - \varphi[J_2](\boldsymbol{X}) = \varphi[J_3](\boldsymbol{X}) - \varphi[J_4](\boldsymbol{X}) \big\}$$

**174** is zero-measured. For this step, the technique in proving Lemma 4.1 in [CD19] can be applied to
**175** prove that $A$ does not contain any open set, and all its points are singular.

**176** We can further define

$$\mathcal{F}_n^2 = \{ (J_1, J_2) : J_1, J_2 \subset [n], |J_1| = |J_2| = 2, J_1 \neq J_2 \}.$$

**177** Since $A$ is zero-measured, we can only consider the set $[0,1]^{d \times n} - A$, on which

$$\{ \Delta\varphi[J_1, J_2](\boldsymbol{X}) := \varphi[J_1](\boldsymbol{X}) - \varphi[J_2](\boldsymbol{X}) \}_{(J_1, J_2) \in \mathcal{F}_n^2}$$

**178** must take different values for different $(J_1, J_2) \in \mathcal{F}_n^2$. Denote these values as $r_1 < r_2 < ... <$
**179** $r_L$, and let $E_\ell(\boldsymbol{X})$ denote the element $(J_1, J_2) \subset \mathcal{F}_n^2$ such that $\Delta\varphi[J_1, J_2](\boldsymbol{X}) = r_\ell$. The sets
**180** $E_1(\boldsymbol{X}), E_2(\boldsymbol{X}), ..., E_L(\boldsymbol{X})$ then form a partition of $\mathcal{F}_n^2$. With similar techniques as Lemma 4.2 in
**181** [CD19], we can prove that the map $\boldsymbol{X} \mapsto \mathcal{A}^2(\boldsymbol{X})$ is locally constant almost surely everywhere. This
**182** essentially completes the proof.

■

184 The following lemma is a direct application of Proposition 4.6 in [DP19], and guarantees that the
185 number of points in the persistence diagram $\mathsf{Dgm}(\boldsymbol{X}, \varphi)$ that are far enough from the diagonal is
186 upper bounded in terms of the expectation.

187 **Lemma C.5** *Let $\kappa$ be a probability density function on $[0, 1]^d$ that satisfies $0 < \inf \kappa < \sup \kappa < \infty$.*
188 *Denote $\mathbb{X}_n$ as a binomial process with parameters $n$ and $\kappa$ or a Poisson process with parameter $n\kappa$*
189 *on $[0, 1]^d$. In the kth dimensional persistence diagram of the Vietoris-Rips filtration of $\mathbb{X}_n$, let $N_\ell$ be*
190 *the number of points with persistence of at least $\ell$. Then there are some universal constant $C$ that the*
191 *expectation of $N_\ell$ is upper bounded as*

$$\mathbb{E}\left[N_\ell\right] \leq Cn \exp\left(-Cn\ell^d\right),$$

192 *where $C$ is a constant depends only on $k$.*

193 *Proof:* Let $\mu$ be the persistence measure corresponding to the $k$-th dimensional persistence diagram
194 of the Vietoris-Rips filtration of $\mathbb{X}_n$. From Proposition 4.6 in [DP19],

$$P\left(\mu(\mathbb{R} \times [\ell, \infty)) > t\right) \leq c_1 \exp\left(-c_2\left(n\ell^d + (\frac{t}{n})^{1/(k+1)}\right)\right).$$

195 And hence the expectation of $\mu(\mathbb{R} \times [\ell, \infty))$ is bounded as

$$
\begin{aligned}
\mathbb{E}\left[\mu(\mathbb{R} \times [\ell, \infty))\right] &\leq \int_0^\infty c_1 \exp\left(-c_2\left(n\ell^d + (\frac{t}{n})^{1/(k+1)}\right)\right) dt \\
&= c_1 \exp\left(-c_2(n\ell^d)\right) \int_0^\infty \exp\left(-c_2(\frac{t}{n})^{1/(k+1)}\right) dt \\
&= c_1 \exp\left(-c_2(n\ell^d)\right) \int_0^\infty (k+1)nu^k \exp\left(-c_2 u\right) du \\
&= Cn \exp\left(-Cn\ell^d\right),
\end{aligned}
$$

196 for some constant $C$ that depends on $k$. Now, $\mathbb{R} \times [\ell, \infty)$ contains all the homological features whose
197 persistence is at least $\ell$, so

$$N_\ell \leq \mu(\mathbb{R} \times [\ell, \infty)).$$

198 And hence

$$\mathbb{E}\left[N_\ell\right] \leq Cn \exp\left(-Cn\ell^d\right).$$

199

■

## C.2 Uniform tail bounds

201 In this section, we provide some uniform tail bound theorems that are important for bounding the
202 variation of estimators. We will omit the proofs of these theorems in the paper.

203 **The Talagrand's inequality.** The following form of the Talagrand's inequality was shown in
204 [SC08].

205 **Theorem C.6** *Let $(\mathcal{Z}, \mathscr{F}, P)$ be a probability space and $(T, d)$ be a separable metric space. Con-*
206 *sider a function class $\mathcal{G} = \{g_t : t \in T\} \in L_0(\mathcal{Z})$, such that the function $t \mapsto g_t(z)$ is continuous in*
207 *t for all $z \in \mathcal{Z}$. Furthermore, suppose that there exists a constant $B > 0, \sigma^2 > 0$ such that for all*
208 *$g \in \mathcal{G}$, $\mathbb{E}[g] = 0, \mathbb{E}[g^2] \leq \sigma^2, ||g||_\infty \leq B$. Let $Z_1, Z_2, ..., Z_n \sim$ i.i.d. P, and define*

$$G = \sup_{g \in \mathcal{G}} \left| \frac{1}{n} \sum_{i=1}^n g(Z_i) \right|.$$

209 *Then for any $\delta \in (0, 1)$, with probability of at least $1 - \delta$,*

$$G \leq 4\mathbb{E}[G] + \sqrt{\frac{2\sigma^2}{n} \log \frac{1}{\delta}} + \frac{B}{n} \log \frac{1}{\delta}. \tag{2}$$

210 Theorem C.6 implies that the expectation of $G$ is an important factor in bounding $G$. The following
211 theorem gives and upper bound of $\mathbb{E}[G]$ by the covering number of $\mathcal{G}$.

**Theorem C.7** *Under the same conditions as in Theorem C.6, if for any $\eta \in (0, B)$, there exists*
213 $A > 0$, $\nu > 0$ *such that for any probability measure $Q$ on $\mathcal{Z}$, the covering number*

$$\mathcal{N}(\mathcal{G}, L_2(Q), \eta) \leq \left(\frac{AB}{\eta}\right)^{\nu},$$

214 *then there exists a constant $C$ such that*

$$\mathbb{E}[G] \leq C \left(\frac{\nu B}{n} \log\left(\frac{AB}{\sigma}\right) + \sqrt{\frac{\nu \sigma^2}{n} \log\left(\frac{AB}{\sigma}\right)}\right).$$

**Tail bound by polynomial discrimination.** As an alternative to the Talagrand's inequality, the
216 following theorem bounds $G$ with high probability when the function class $\mathcal{G}$ has *polynomial*
217 *discrimination*. The proof applies the Bernstein's inequality and a straightforward union bound
218 argument.

**Theorem C.8** *Under the same conditions as in Theorem C.6, define*

$$\mathcal{G}(\boldsymbol{Z}_1^n) = \{(g(Z_1), g(Z_2), ..., g(Z_n)) : g \in \mathcal{G}\}. \tag{3}$$

220 *If the cardinality of the set $\mathcal{G}(\boldsymbol{Z}_1^n)$ is bounded by*

$$Card(\mathcal{G}(\boldsymbol{Z}_1^n)) \leq (An + 1)^{\nu} \tag{4}$$

221 *for some $\nu > 0$, then there exists a universal constant $C$ such that with probability at least $1 - \delta$,*

$$G \leq C \left(\sqrt{\frac{\sigma^2}{n}} \left(\sqrt{\nu \log(An + 1)} + \sqrt{\log\frac{1}{\delta}}\right) + \frac{B}{n}\left(\nu \log(An + 1) + \log\frac{1}{\delta}\right)\right) \tag{5}$$

222 The following lemma shows that for persistent measures with bounded total persistence, the total
223 mass of the set away from the diagonal $\partial\Omega$ is upper bounded.

**Lemma C.9** *Let $\Omega_\ell$ denote the set of points in $\Omega$ that are at least $\ell$ away from the diagonal:*

$$\Omega_\ell = \{\boldsymbol{\omega} \in \Omega : \|\boldsymbol{\omega} - \partial\Omega\|_2 \geq \ell\}.$$

225 *Then for a persistent measure $\mu$, if $Pers_q(\mu) \leq M$, then $\mu(\Omega_\ell) \leq M\ell^{-q}$.*

226 The following theorem shown in [DL21] provides a standard lower bound for the minimax rate of
227 estimating a probability density function using independent samples. This is useful for deducting the
228 minimax rate for estimating the (weighted) intensity functions.

**Theorem C.10** *Let $\mathscr{F}$ denote the set of probability density functions on $[0, 1]^2$ with Bounded Besov*
230 *norm:*

$$\mathscr{F} = \{f : [0, 1]^2 \to \mathbb{R}, \int_{[0,1]^2} f(x)\mathrm{d}x = 1, \|f\|_{\infty,\infty}^r \leq B\}.$$

231 *Then for any estimator (measurable function)*

$$\hat{f}_n : ([0, 1]^2)^n \to \mathscr{F},$$

232 *there exists $f \in \mathscr{F}$, such that if $X_1, X_2, ..., X_n \sim$ i.i.d. $f$, then*

$$\mathbb{E}\|\hat{f}_n(X_1, X_2, ..., X_n) - f\|_\infty \geq O\left(n^{-\frac{r}{2r+2}}\right).$$

# D   Proof of theorems and supportive propositions

## D.1   Proof of Theorem 3.1

235 In order to prove Theorem 3.1, we firstly show the following supportive lemma.

**Lemma D.1** *Let $\Omega$ and $\partial\Omega$ be defined as in (1) and (2). Then for any $q > 0$,*

$$\int_\Omega \|\boldsymbol{x} - \partial\Omega\|_2^q \mathrm{d}\boldsymbol{x} = \frac{2}{(q+1)(q+2)}\left(\frac{L}{\sqrt{2}}\right)^{q+2}.$$

*Proof of Lemma D.1*: Take the coordinate transformation

$$\begin{cases} y_1 = \frac{x_2 - x_1}{\sqrt{2}} = \|\boldsymbol{x} - \partial\Omega\|_2; \\ y_2 = \frac{x_2 + x_1}{\sqrt{2}}. \end{cases}$$

Then it can be easily verified that the determinant of the Jacobian matrix between $\boldsymbol{x}$ and $\boldsymbol{y}$ coordinates is 1, and that the $\ell_1$ ball $\Omega$ can be represented using $\boldsymbol{y}$ coordinates by

$$\Omega = \{(y_1, y_2) : 0 < y_1 \leq \frac{L}{\sqrt{2}}, y_1 \leq y_2 \leq \sqrt{2}L - y_1\}.$$

Therefore,

$$\int_\Omega \|\boldsymbol{x} - \partial\Omega\|_2^q \mathrm{d}\boldsymbol{x} = \int_0^{\frac{L}{\sqrt{2}}} \left(\int_{y_1}^{\sqrt{2}L - y_1} \mathrm{d}y_2\right) y_1^q \mathrm{d}y_1$$

$$= \int_0^{\frac{L}{\sqrt{2}}} (\sqrt{2}L - 2y_1) y_1^q \mathrm{d}y_1$$

$$= \frac{2}{(q+1)(q+2)}\left(\frac{L}{\sqrt{2}}\right)^{q+2}.$$

With this lemma, we can now prove Theorem 3.1.

*Proof of Theorem 3.1*: The main idea of bounding the OT distance is to construct an admissible transport between $\mu$ and $\nu$, and then control the cost of this transport. We will separate the proof into three steps accordingly.

**Step 1: Construct an admissible transport from $\mu$ to $\nu$.** Define $\hat{\pi}$ as a measure on $\overline{\Omega} \times \overline{\Omega}$ such that for any Borel sets $A, B \subset \overline{\Omega}$,

$$\hat{\pi}(A \times B) = \int_{A \cap B \cap \Omega} \min\{p_\mu(\boldsymbol{x}), p_\nu(\boldsymbol{x})\}\mathrm{d}\boldsymbol{x} + $$
$$\int_{A \cap \mathsf{Proj}_{\partial\Omega}^{-1}(B) \cap \Omega} [p_\mu(\boldsymbol{x}) - p_\nu(\boldsymbol{x})]^+ \mathrm{d}\boldsymbol{x} + \int_{B \cap \mathsf{Proj}_{\partial\Omega}^{-1}(A) \cap \Omega} [p_\nu(\boldsymbol{x}) - p_\mu(\boldsymbol{x})]^+ \mathrm{d}\boldsymbol{x}. \tag{6}$$

Here, for any set $A \subset \overline{\Omega}$,

$$\mathsf{Proj}_{\partial\Omega}^{-1}(A) = \{\boldsymbol{\omega} \in \Omega : \mathsf{Proj}_{\partial\Omega}(\omega) \in A\}.$$

Intuitively, $\hat{\pi}$ represents such a transport: at each point $\boldsymbol{x} \in \Omega$, if $p_\mu(\boldsymbol{x}) > p_\nu(\boldsymbol{x})$, then we transport the mass of $p_\nu$ from $\boldsymbol{x}$ to $\boldsymbol{x}$, and the remaining mass from $\boldsymbol{x}$ to its projection onto $\partial\Omega$; if $p_\nu(\boldsymbol{x}) > p_\mu(\boldsymbol{x})$, then the opposite is done.

Firstly, we prove that this is an admissible transport between $\mu$ and $\nu$. Notice that for any Borel set $A \subset \Omega$, $A \cap \overline{\Omega} \cap \Omega = A$, $A \cap \mathsf{Proj}_{\partial\Omega}^{-1}(\overline{\Omega}) \cap \Omega = A$ and $\mathsf{Proj}_{\partial\Omega}^{-1}(A) = \emptyset$. Therefore, by taking $B = \overline{\Omega}$ in (6), we get

$$\hat{\pi}(A \times \overline{\Omega}) = \int_A \min\{p_\mu(\boldsymbol{x}), p_\nu(\boldsymbol{x})\}\mathrm{d}\boldsymbol{x} + \int_A [p_\mu(\boldsymbol{x}) - p_\nu(\boldsymbol{x})]^+ \mathrm{d}\boldsymbol{x} + 0$$

$$= \int_A \left\{\min\{p_\mu(\boldsymbol{x}), p_\nu(\boldsymbol{x})\} + [p_\mu(\boldsymbol{x}) - p_\nu(\boldsymbol{x})]^+\right\} \mathrm{d}\boldsymbol{x}$$

$$= \int_A p_\mu(\boldsymbol{x})\mathrm{d}\boldsymbol{x} = \mu(A).$$

Similarly, we can prove that $\hat{\pi}(\overline{\Omega} \times B) = \nu(B)$ for any Borel set $B \subset \Omega$. Therefore, $\hat{\pi}$ is an admissible transport between $\mu$ and $\nu$.

**Step 2: Present** $\mathrm{d}\hat{\pi}$. In order to calculate the transport cost of $\hat{\pi}$, we firstly need to present $\mathrm{d}\hat{\pi}$. For this, we would make use of *pushforward measures*. Define $\imath : \bar{\Omega} \to \bar{\Omega} \times \bar{\Omega}$ by $\imath(\boldsymbol{x}) = (\boldsymbol{x}, \boldsymbol{x})$, and let $\jmath : \bar{\Omega} \times \bar{\Omega} \to \bar{\Omega}$ be satisfying $\jmath \circ \imath = id$. Furthermore, let $\imath_*(\lambda_\Omega)$ be the pushforward measure on $\bar{\Omega} \times \bar{\Omega}$ generated by $\imath$. Then for any Borel sets $A, B \subset \overline{\Omega}$, one has $\imath^{-1}(A \times B) = A \cap B$, and the first term in (6) can be presented as

$$
\int_{A \cap B \cap \Omega} \min\{p_\mu(\boldsymbol{x}), p_\nu(\boldsymbol{x})\}\, d\boldsymbol{x}
$$
$$
= \int_{\imath^{-1}(A \times B)} \min\{(p_\mu \circ \jmath)(\imath(\boldsymbol{x})), (p_\nu \circ \jmath)(\imath(\boldsymbol{x}))\}\, d\lambda_\Omega(\boldsymbol{x})
$$
$$
= \int_{A \times B} \min\{(p_\mu \circ \jmath)(\boldsymbol{x}, \boldsymbol{y}), (p_\nu \circ \jmath)(\boldsymbol{x}, \boldsymbol{y})\}\, d\imath_*(\lambda_\Omega)(\boldsymbol{x}, \boldsymbol{y}).
$$

For the second term in (6), we can similarly, define $\imath^{(1)} : \bar{\Omega} \to \bar{\Omega} \times \bar{\Omega}$ by $\imath^{(1)}(\boldsymbol{x}) = (\boldsymbol{x}, \mathrm{Proj}_{\partial\Omega}(\boldsymbol{x}))$, let $\jmath^{(1)} : \bar{\Omega} \times \bar{\Omega} \to \bar{\Omega}$ be satisfying $\jmath^{(1)} \circ \imath^{(1)} = id$, and consider the pushforward measure $\imath^{(1)}_*(\lambda_\Omega)$. Then $(\imath^{(1)})^{-1}(A \times B) = A \cap \mathrm{Proj}^{-1}_{\partial\Omega}(B)$, and

$$
\int_{A \cap \mathrm{Proj}^{-1}_{\partial\Omega}(B) \cap \Omega} [p_\mu(\boldsymbol{x}) - p_\nu(\boldsymbol{x})]^+\, d\boldsymbol{x}
$$
$$
= \int_{(\imath^{(1)})^{-1}(A \times B)} \left[(p_\mu \circ \jmath^{(1)})(\imath^{(1)}(\boldsymbol{x})) - (p_\nu \circ \jmath^{(1)})(\imath^{(1)}(\boldsymbol{x}))\right]^+\, d\lambda_\Omega(\boldsymbol{x})
$$
$$
= \int_{A \times B} \left[(p_\mu \circ \jmath^{(1)})(\boldsymbol{x}, \boldsymbol{y}) - (p_\nu \circ \jmath^{(1)})(\boldsymbol{x}, \boldsymbol{y})\right]^+\, d\imath^{(1)}_*(\lambda_\Omega)(\boldsymbol{x}, \boldsymbol{y}).
$$

For the third term in (6), we can similarly define $\imath^{(2)} : \bar{\Omega} \to \bar{\Omega} \times \bar{\Omega}$ by $\imath^{(2)}(\boldsymbol{x}) = (\mathrm{Proj}_{\partial\Omega}(\boldsymbol{x}), \boldsymbol{x})$, let $\jmath^{(2)} : \bar{\Omega} \times \bar{\Omega} \to \bar{\Omega}$ be satisfying $\jmath^{(2)} \circ \imath^{(2)} = id$, and consider a pushforward measure $\imath^{(2)}_*(\lambda_\Omega)$. Then $(\imath^{(2)})^{-1}(A \times B) = \mathrm{Proj}^{-1}_{\partial\Omega}(A) \cap B$, and

$$
\int_{\mathrm{Proj}^{-1}_{\partial\Omega}(A) \cap B \cap \Omega} [p_\mu(\boldsymbol{x}) - p_\nu(\boldsymbol{x})]^+\, d\boldsymbol{x}
$$
$$
= \int_{(\imath^{(2)})^{-1}(A \times B)} \left[(p_\mu \circ \jmath^{(2)})(\imath^{(2)}(\boldsymbol{x})) - (p_\nu \circ \jmath^{(2)})(\imath^{(2)}(\boldsymbol{x}))\right]^+\, d\lambda_\Omega(\boldsymbol{x})
$$
$$
= \int_{A \times B} \left[(p_\mu \circ \jmath^{(2)})(\boldsymbol{x}, \boldsymbol{y}) - (p_\nu \circ \jmath^{(2)})(\boldsymbol{x}, \boldsymbol{y})\right]^+\, d\imath^{(1)}_*(\lambda_\Omega)(\boldsymbol{x}, \boldsymbol{y}).
$$

Combining these results, we can obtain the following presentation of $\mathrm{d}\hat{\pi}$:

$$
\mathrm{d}\hat{\pi} = \min\{(p_\mu \circ \jmath)(\boldsymbol{x}, \boldsymbol{y}), (p_\nu \circ \jmath)(\boldsymbol{x}, \boldsymbol{y})\}\, d\imath_*(\lambda_\Omega)
$$
$$
+ \left[(p_\mu \circ \jmath^{(1)})(\boldsymbol{x}, \boldsymbol{y}) - (p_\nu \circ \jmath^{(1)})(\boldsymbol{x}, \boldsymbol{y})\right]^+\, d\imath^{(1)}_*(\lambda_\Omega)
$$
$$
+ \left[(p_\mu \circ \jmath^{(2)})(\boldsymbol{x}, \boldsymbol{y}) - (p_\nu \circ \jmath^{(2)})(\boldsymbol{x}, \boldsymbol{y})\right]^+\, d\imath^{(2)}_*(\lambda_\Omega).
$$

**Step 3: Calculate the transportation cost of** $\hat{\pi}$. Based on our presentation of $\mathrm{d}\hat{\pi}$, the $q$-th order transportation cost of $\hat{\pi}$ is, by definition:

$$
C^q_q(\hat{\pi}) = \int_{\overline{\Omega} \times \overline{\Omega}} \|\boldsymbol{x} - \boldsymbol{y}\|^q_2 \mathrm{d}\hat{\pi}(\boldsymbol{x}, \boldsymbol{y})
$$
$$
= \int_{\overline{\Omega} \times \overline{\Omega}} \|\boldsymbol{x} - \boldsymbol{y}\|^q_2 \min\{(p_\nu \circ \jmath)(\boldsymbol{x}, \boldsymbol{y}), (p_\mu \circ \jmath)(\boldsymbol{x}, \boldsymbol{y})\}\, d\imath_*(\lambda_\Omega)
$$
$$
+ \int_{\overline{\Omega} \times \overline{\Omega}} \|\boldsymbol{x} - \boldsymbol{y}\|^q_2 \left[(p_\mu \circ \jmath^{(1)})(\boldsymbol{x}, \boldsymbol{y}) - (p_\nu \circ \jmath^{(1)})(\boldsymbol{x}, \boldsymbol{y})\right]^+\, d\imath^{(1)}_*(\lambda_\Omega)
$$
$$
+ \int_{\overline{\Omega} \times \overline{\Omega}} \|\boldsymbol{x} - \boldsymbol{y}\|^q_2 \left[(p_\mu \circ \jmath^{(2)})(\boldsymbol{x}, \boldsymbol{y}) - (p_\nu \circ \jmath^{(2)})(\boldsymbol{x}, \boldsymbol{y})\right]^+\, d\imath^{(2)}_*(\lambda_\Omega). \tag{7}
$$

We now explore the three terms in (7). First of all, since $\imath_*(\lambda_\Omega)$ is a pushforward measure generated by the function $\imath(\boldsymbol{x}) = (\boldsymbol{x}, \boldsymbol{x})$, it is easy to see that

$$\imath_*(\lambda_\Omega)(\{(\boldsymbol{x}, \boldsymbol{y}) \in \Omega \times \Omega : \boldsymbol{x} \neq \boldsymbol{y}\}) = 0.$$

Therefore, the first term in (7) is simply

$$\int_{\overline{\Omega} \times \overline{\Omega}} \|\boldsymbol{x} - \boldsymbol{y}\|_2^q \min \{(p_\nu \circ \jmath)(\boldsymbol{x}, \boldsymbol{y}), (p_\mu \circ \jmath)(\boldsymbol{x}, \boldsymbol{y})\} \, d\imath_*(\lambda_\Omega)$$

$$= \int_{(\boldsymbol{x}, \boldsymbol{y}) \in \overline{\Omega} \times \overline{\Omega}, \boldsymbol{x} = \boldsymbol{y}} \|\boldsymbol{x} - \boldsymbol{y}\|_2^q \min \{(p_\nu \circ \jmath)(\boldsymbol{x}, \boldsymbol{y}), (p_\mu \circ \jmath)(\boldsymbol{x}, \boldsymbol{y})\} \, d\imath_*(\lambda_\Omega)$$

$$= \int_{\boldsymbol{x} \in \overline{\Omega}} \|\boldsymbol{x} - \boldsymbol{x}\|_2^q \min\{p_\mu(\boldsymbol{x}), p_\nu(\boldsymbol{x})\} d\boldsymbol{x} = 0.$$

As for the second term, notice that $\imath_*^{(1)}(\lambda_\Omega)$ is a pushforward measure generated by the function $\imath^{(1)}(\boldsymbol{x}) = (\boldsymbol{x}, \mathsf{Proj}_{\partial\Omega}(\boldsymbol{x}))$. Therefore by definition,

$$\imath_*^{(1)}(\lambda_\Omega)(\{(\boldsymbol{x}, \boldsymbol{y}) \in \Omega \times \Omega : \boldsymbol{y} \neq \mathsf{Proj}_{\partial\Omega}(\boldsymbol{x})\}) = 0.$$

Hence, the second term in (7) is equal to

$$\int_{\overline{\Omega} \times \overline{\Omega}} \|\boldsymbol{x} - \boldsymbol{y}\|_2^q \left[(p_\mu \circ \jmath^{(1)})(\boldsymbol{x}, \boldsymbol{y}) - (p_\nu \circ \jmath^{(1)})(\boldsymbol{x}, \boldsymbol{y})\right]^+ d\imath_*^{(1)}(\lambda_\Omega)$$

$$= \int_{(\boldsymbol{x}, \boldsymbol{y}) \in \overline{\Omega} \times \overline{\Omega}, \boldsymbol{y} = \mathsf{Proj}_{\partial\Omega}(\boldsymbol{x})} \|\boldsymbol{x} - \boldsymbol{y}\|_2^q$$

$$\times \left[(p_\mu \circ \jmath^{(1)})(\boldsymbol{x}, \mathsf{Proj}_{\partial\Omega}(\boldsymbol{x})) - (p_\nu \circ \jmath^{(1)})(\boldsymbol{x}, \mathsf{Proj}_{\partial\Omega}(\boldsymbol{x}))\right]^+ d\imath_*^{(1)}(\lambda_\Omega)$$

$$= \int_{\boldsymbol{x} \in \overline{\Omega}} \|\boldsymbol{x} - \mathsf{Proj}_{\partial\Omega}(\boldsymbol{x})\|_2^q \left[(p_\mu \circ \jmath^{(1)} \circ \imath^{(1)})(\boldsymbol{x}) - (p_\nu \circ \jmath^{(1)} \circ \imath^{(1)})(\boldsymbol{x})\right] d\boldsymbol{x}$$

$$= \int_\Omega \|\boldsymbol{x} - \partial\Omega\|_2^q \left[p_\mu(\boldsymbol{x}) - p_\nu(\boldsymbol{x})\right]^+ d\boldsymbol{x}.$$

Similarly, we can obtain that the third term of (7) is equal to

$$\int_{\overline{\Omega} \times \overline{\Omega}} \|\boldsymbol{x} - \boldsymbol{y}\|_2^q \left[(p_\mu \circ \jmath^{(2)})(\boldsymbol{x}, \boldsymbol{y}) - (p_\nu \circ \jmath^{(2)})(\boldsymbol{x}, \boldsymbol{y})\right]^+ d\imath_*^{(2)}(\lambda_\Omega)$$

$$= \int_\Omega [p_\nu(\boldsymbol{x}) - p_\mu(\boldsymbol{x})]^+ \|\boldsymbol{x} - \partial\Omega\|_2^q d\boldsymbol{x}.$$

Combining these results, we obtain

$$C_q^q(\hat{\pi}) = \int_\Omega [p_\mu(\boldsymbol{x}) - p_\nu(\boldsymbol{x})]^+ \|\boldsymbol{x} - \partial\Omega\|_2^q d\boldsymbol{x} + \int_\Omega [p_\nu(\boldsymbol{x}) - p_\mu(\boldsymbol{x})]^+ \|\boldsymbol{x} - \partial\Omega\|_2^q d\boldsymbol{x}$$

$$= \int_\Omega |p_\mu(\boldsymbol{x}) - p_\nu(\boldsymbol{x})| \|\boldsymbol{x} - \partial\Omega\|_2^q d\boldsymbol{x}$$

$$\leq \|p_\mu - p_\nu\|_\infty \int_\Omega \|\boldsymbol{x} - \partial\Omega\|_2^q d\boldsymbol{x} = \frac{2}{(q+1)(q+2)} \left(\frac{L}{\sqrt{2}}\right)^{q+2} \|p_\mu - p_\nu\|_\infty.$$

Notice that the last equality uses Lemma D.1.

Finally, since $\hat{\pi}$ is an admissible transport from $\mu$ to $\nu$, the optimal transport distance between $\mu$ and $\nu$, $\mathsf{OT}_q(\mu, \nu)$, should be at most $C_q(\hat{\pi})$. The bound (7) follows naturally.

**Example of converging OT distance while intensity functions diverge.** Consider the following sequences of intensity functions

$$p_{\mu_n} = \frac{4^n}{L^2} \mathbb{1} \left\{ \|\boldsymbol{x} - \boldsymbol{u}_n\|_1 < \frac{\sqrt{2}L}{2^{n+1}} \right\}$$

$$p_{\nu_n} = \frac{4^n}{L^2} \mathbb{1} \left\{ \|\boldsymbol{x} - \boldsymbol{d}_n\|_1 < \frac{\sqrt{2}L}{2^{n+1}} \right\},$$

in which

$$\boldsymbol{u}_n = \left( \frac{\sqrt{2}L}{4}, \frac{\sqrt{2}L}{4} + \frac{\sqrt{2}L}{2^{n+1}} \right)$$

$$\boldsymbol{d}_n = \left( \frac{\sqrt{2}L}{4} - \frac{\sqrt{2}L}{2^{n+1}}, \frac{\sqrt{2}L}{4} \right).$$

Essentially, $\mu_n$ and $\nu_n$ are uniform distributions on two adjacent $\ell_1$ balls. It is easy to verify that the total mass of both $\mu_n$ and $\nu_n$ is 1, and the optimal transport distance between $\mu_n$ and $\nu_n$ is upper bounded by

$$\mathsf{OT}_q(\mu_n, \nu_n) \leq \frac{L}{2^n} \to 0;$$

on the other hand, the $\ell_\infty$ distance between the intensity functions clearly diverges as $n \to \infty$:

$$\|p_{\mu_n} - p_{\nu_n}\|_\infty \geq |p_{\mu_n}(\boldsymbol{u}_n) - p_{\nu_n}(\boldsymbol{u}_n)| = \frac{4^n}{L^2} \to \infty.$$

∎

*A remark on the bottleneck distance.* We argue that there can be no meaningful upper bound for the bottleneck distance $\mathsf{OT}_\infty$ by the $\ell_\infty$ distance between the intensity or density functions. Consider the following example: define $T_h$ as an upper-left triangle in $\Omega$:

$$T_h := \{\boldsymbol{\omega} \in \Omega \mid \|\boldsymbol{\omega} - \partial\Omega\|_2 \geq \frac{L - h}{\sqrt{2}}\},$$

and $T'_h$ as a triangle tangent to the diagonal:

$$T'_h := \left\{ \boldsymbol{\omega} \in \Omega \mid \left\| \boldsymbol{\omega} - \left( \frac{L}{2}, \frac{L}{2} \right) \right\|_\infty \leq \frac{h}{2} \right\}.$$

We define $\mu_h$ as the uniform distribution on $T_h$, so that

$$p_{\mu_h}(\boldsymbol{\omega}) = \frac{2}{h^2} \mathbb{1}\{\boldsymbol{\omega} \in T_h\};$$

on the other hand $\nu$ is very similar to $\mu$ but has a small part of its mass on $T'_h$:

$$p_{\nu_h}(\boldsymbol{\omega}) = \left( \frac{2}{h^2} - h \right) \mathbb{1}\{\boldsymbol{\omega} \in T_h\} + h\mathbb{1}\{\boldsymbol{\omega} \in T'_h\}.$$

As $h \to 0$, it is easy to verify that $\|p_{\mu_h} - p_{\nu_h}\|_\infty = h \to 0$, while $\mathsf{OT}(\mu_h, \nu_h) \to L/\sqrt{2}$. This is because although the densities for $\mu$ and $\nu$ becomes very close, there is always a small part of the mass of $\mu$ in $T_h$ that has to be transported to $T'_h$; since the bottleneck distance only considers the *maximum* transport cost, it would converge to the limiting distance between $T_h$ and $T'_h$, which is $L/\sqrt{2}$. It is easy to generalize this example to the case where $p_{\mu_h}$ and $p_{\nu_h}$ are smooth.

## D.2    Proof of Theorem 3.5

Both theorems are classic results on the bias of kernel estimators and are proved by the smoothness of the target functions as supposed by Assumption 3.2. We here provides the proof of Theorem 3.5 (a), and part (b) can be proved in a completely similar fashion.

We firstly clarify the specific smoothness condition proposed by Assumption 3.2. It guarantees Hence, we can represent the bias of $\mathbb{E}[\hat{p}_h(\boldsymbol{\omega})]$ as an integral. Since $\bar{\mu}_n$ is an unbiased estimator for $\mathbb{E}[\mu]$,

$$\begin{aligned}
\mathbb{E}[\hat{p}_h(\boldsymbol{\omega})] - p(\boldsymbol{\omega}) &= \mathbb{E}\left[ \int_{\boldsymbol{x}} \frac{1}{h^2} K\left( \frac{\boldsymbol{x} - \boldsymbol{\omega}}{h} \right) \mathrm{d}\bar{\mu}_n \right] - p(\boldsymbol{\omega}) \\
&= \int_{\boldsymbol{x}} \frac{1}{h^2} K\left( \frac{\boldsymbol{x} - \boldsymbol{\omega}}{h} \right) \mathrm{d}\mathbb{E}[\bar{\mu}_n] - p(\boldsymbol{\omega}) \\
&= \int_{\boldsymbol{x}} \frac{1}{h^2} K\left( \frac{\boldsymbol{x} - \boldsymbol{\omega}}{h} \right) p(\boldsymbol{x})\mathrm{d}\boldsymbol{x} - p(\boldsymbol{\omega}) \\
&= \int_{\boldsymbol{x}} \frac{1}{h^2} K\left( \frac{\boldsymbol{x} - \boldsymbol{\omega}}{h} \right) [p(\boldsymbol{x}) - p(\boldsymbol{\omega})]\mathrm{d}\boldsymbol{x},
\end{aligned}$$

where in the last line we applied the property that the kernel function $K(\cdot)$ integrals to 1. We can then apply the smoothness of $p(\cdot)$ as in (1) and obtain that

$$|\mathbb{E}[\hat{p}_h(\boldsymbol{\omega})] - p(\boldsymbol{\omega})|$$

$$\leq \left| \int_{\boldsymbol{x}} \frac{1}{h^2} K\left(\frac{\boldsymbol{x}-\boldsymbol{\omega}}{h}\right) \sum_{t=1}^{s-1} \frac{1}{t!} \sum_{t_1+t_2=t} \frac{\mathrm{d}^t p(\boldsymbol{\omega})}{\mathrm{d}\omega_1^{t_1}\mathrm{d}\omega_2^{t_2}} (x_1-\omega_1)^{t_1}(x_2-\omega_2)^{t_2}\mathrm{d}\boldsymbol{x} \right|$$

$$+ \int_{\boldsymbol{x}} \frac{1}{h^2} \left| K\left(\frac{\boldsymbol{x}-\boldsymbol{\omega}}{h}\right) \right| L_p \|\boldsymbol{x}-\boldsymbol{\omega}\|_2^s \mathrm{d}\boldsymbol{x}$$

By taking a change of variable $\boldsymbol{v} = \frac{\boldsymbol{x}-\boldsymbol{\omega}}{h}$ , the first term can be represented as

$$\sum_{t=1}^{s-1} \frac{1}{t!} \sum_{t_1+t_2=t} \frac{\mathrm{d}^t p(\boldsymbol{\omega})}{\mathrm{d}\omega_1^{t_1}\mathrm{d}\omega_2^{t_2}} \int_{\|\boldsymbol{v}\|_2 \leq 1} K(\boldsymbol{v}) h^t v_1^{t_1} v_2^{t_2}\mathrm{d}\boldsymbol{v}.$$

The zero-moment condition of the kernel function in Assumption B.3 guarantees that this term equals to 0. Hence,

$$|\mathbb{E}[\hat{p}_h(\boldsymbol{\omega})] - p(\boldsymbol{\omega})| \leq \int_{\boldsymbol{x}} \frac{1}{h^2} \left| K\left(\frac{\boldsymbol{x}-\boldsymbol{\omega}}{h}\right) \right| L_p \|\boldsymbol{x}-\boldsymbol{\omega}\|_2^s \mathrm{d}\boldsymbol{x}$$

$$\xrightarrow{\boldsymbol{v}=(\boldsymbol{x}-\boldsymbol{\omega})/h} L_p h^s \int_{\|\boldsymbol{v}\|_2 \leq 1} |K(\boldsymbol{v})| \|\boldsymbol{v}\|_2^s \mathrm{d}\boldsymbol{v}.$$

### D.3 Proof of Theorem 3.6 (a)

**A useful claim.** The following claim can be applied for easing calculation in Theorem 3.6.

**Claim D.2** *For $q \in \mathbb{R}$ and $x \in [0,1]$,*

$$1 - x^q \leq (q \vee 1)(1-x),$$

*where $q \vee 1 = \max\{q, 1\}$.*

**Proof of Claim D.2.** If $q \geq 1$ or $q \leq 0$, let $f(x) = 1 - x^q$. Then $f'(x) = -qx^{q-1}$ and $f''(x) = -q(q-1)x^{q-2}$, so $f''(x) \leq 0$ for $x \in [0,1]$ and $f$ is concave on $[0,1]$. Then by Jensen's inequality,

$$1 - x^q = f(x) \leq f(1) + f'(1)(x-1) = q(1-x).$$

If $q \in [0,1]$, then $x^q \geq x$ implies

$$1 - x^q \leq 1 - x.$$

Hence combining these gives

$$1 - x^q \leq (q \vee 1)(1-x).$$

∎

This proof applies the Talagrand's inequality. For this purpose, we firstly define an auxiliary family of functions, and then verify the conditions in Theorems C.6 and C.7 .

**Defining an auxiliary function class.** Let $\mu_1, \mu_2, ...., \mu_n$ be i.i.d. random measures in $\mathcal{Z}_{L,M}^q$, $\ell_{\boldsymbol{\omega}} = \|\boldsymbol{\omega} - \partial\Omega\|_2 - h$ and $g_{\boldsymbol{\omega}}$ be defined as

$$g_{\boldsymbol{\omega}}(\mu) = \ell_{\boldsymbol{\omega}}^q \left( \int_{\Omega} \frac{1}{h^2} K\left(\frac{\boldsymbol{x}-\boldsymbol{\omega}}{h}\right) \mathrm{d}\mu - \int_{\Omega} \frac{1}{h^2} K\left(\frac{\boldsymbol{x}-\boldsymbol{\omega}}{h}\right) \mathrm{d}\mathbb{E}[\mu] \right), \tag{8}$$

and $K$ satisfy Assumption B.3. Take $\mathcal{Z} = \mathcal{Z}_{L,M}^q$, $(T,d) = (\Omega_{2h}, \|\cdot\|_2)$, and for all $\mu \in \mathcal{Z}_{L,M}^q$, define $\mathcal{G} = \{g_{\boldsymbol{\omega}} : \boldsymbol{\omega} \in \Omega_{2h}\}$. By definition, $g_{\boldsymbol{\omega}}(\mu)$ has zero mean and the variation of the kernel estimator $\hat{p}_h(\cdot)$ can be represented by

$$\sup_{\boldsymbol{\omega}\in\Omega_{2h}} \ell_{\boldsymbol{\omega}}^q |\hat{p}_h(\boldsymbol{\omega}) - \mathbb{E}[\hat{p}_h(\boldsymbol{\omega})]| = \sup_{\boldsymbol{\omega}\in\Omega_{2h}} \left| \frac{1}{n}\sum_{i=1}^{n} g_{\boldsymbol{\omega}}(\mu) \right|.$$

Hence, in order to apply the Talagrand's inequality, we need to bound $\|g_{\boldsymbol{\omega}}(\mu)\|_\infty$, $\mathbb{E}[g_{\boldsymbol{\omega}}(\mu)^2]$ and the covering number of $\mathcal{G}$. We provide these upper bound accordingly in the following paragraphs.

**Bounding** $\|g_{\boldsymbol{\omega}}(\mu)\|_\infty$ **and** $\mathbb{E}[g_{\boldsymbol{\omega}}(\mu)^2]$. Notice that since $K$ vanishes outside the unit circle of $\mathbb{R}^2$, for any $\boldsymbol{x} \notin \Omega_{\ell_{\boldsymbol{\omega}}}$, we have $\left\|\frac{\boldsymbol{x}-\boldsymbol{\omega}}{h}\right\|_2 > 1$ and therefore $K\left(\frac{\boldsymbol{x}-\boldsymbol{\omega}}{h}\right) = 0$. Hence, for all $\boldsymbol{\omega} \in \Omega_{2h}$,

$$
\begin{aligned}
|g_{\boldsymbol{\omega}}(\mu)| &= \ell_{\boldsymbol{\omega}}^q \left| \int_\Omega \frac{1}{h^2} K\left(\frac{\boldsymbol{x}-\boldsymbol{\omega}}{h}\right) \mathrm{d}\mu - \int_\Omega \frac{1}{h^2} K\left(\frac{\boldsymbol{x}-\boldsymbol{\omega}}{h}\right) \mathrm{d}\mathbb{E}[\mu] \right| \\
&\leq \ell_{\boldsymbol{\omega}}^q \max\left\{ \left| \int_\Omega \frac{1}{h^2} K\left(\frac{\boldsymbol{x}-\boldsymbol{\omega}}{h}\right) \mathrm{d}\mu \right|, \left| \int_\Omega \frac{1}{h^2} K\left(\frac{\boldsymbol{x}-\boldsymbol{\omega}}{h}\right) \mathrm{d}\mathbb{E}[\mu] \right| \right\} \\
&= \ell_{\boldsymbol{\omega}}^q \max\left\{ \left| \int_{\Omega_{\ell_{\boldsymbol{\omega}}}} \frac{1}{h^2} K\left(\frac{\boldsymbol{x}-\boldsymbol{\omega}}{h}\right) \mathrm{d}\mu \right|, \left| \int_{\Omega_{\ell_{\boldsymbol{\omega}}}} \frac{1}{h^2} K\left(\frac{\boldsymbol{x}-\boldsymbol{\omega}}{h}\right) \mathrm{d}\mathbb{E}[\mu] \right| \right\} \\
&\leq \ell_{\boldsymbol{\omega}}^q \frac{\|K\|_\infty}{h^2} \max\left\{ (\mu(\Omega_{\ell_{\boldsymbol{\omega}}}), \mathbb{E}[\mu](\Omega_{\ell_{\boldsymbol{\omega}}})) \right\} \\
&\leq \ell_{\boldsymbol{\omega}}^q \frac{\|K\|_\infty M}{h^2 \ell_{\boldsymbol{\omega}}^q} = \frac{\|K\|_\infty M}{h^2}
\end{aligned}
\tag{9}
$$

where in the last inequality we used Lemma C.9. On the other hand, the variance of $g_{\boldsymbol{\omega}}$ is bounded by

$$
\begin{aligned}
\mathbb{E}[g_{\boldsymbol{\omega}}(\mu)^2] &= \ell_{\boldsymbol{\omega}}^{2q} \mathbb{E}\left| \int \frac{1}{h^2} K\left(\frac{\boldsymbol{x}-\boldsymbol{\omega}}{h}\right) \mathrm{d}\mu - \int \frac{1}{h^2} K\left(\frac{\boldsymbol{x}-\boldsymbol{\omega}}{h}\right) \mathrm{d}\mathbb{E}[\mu] \right|^2 \\
&\leq \ell_{\boldsymbol{\omega}}^{2q} \mathbb{E}\left| \int_{\Omega_{\ell_{\boldsymbol{\omega}}}} \frac{1}{h^2} K\left(\frac{\boldsymbol{x}-\boldsymbol{\omega}}{h}\right) \mathrm{d}\mu \right|^2 \\
&\leq \ell_{\boldsymbol{\omega}}^{2q} \mathbb{E}\left\{ \mu(\Omega_{\ell_{\boldsymbol{\omega}}}) \cdot \int_{\Omega_\ell} \frac{1}{h^4} K^2\left(\frac{\boldsymbol{x}-\boldsymbol{\omega}}{h}\right) \mathrm{d}\mu \right\} \\
&= \ell_{\boldsymbol{\omega}}^{2q} \mu(\Omega_\ell) \int_{\Omega_{\ell_{\boldsymbol{\omega}}}} \frac{1}{h^4} K^2\left(\frac{\boldsymbol{x}-\boldsymbol{\omega}}{h}\right) \mathrm{d}\mathbb{E}[\mu] \\
&\leq \ell_{\boldsymbol{\omega}}^{2q} \cdot \frac{M}{\ell_{\boldsymbol{\omega}}^q} \int_{\|\boldsymbol{x}-\boldsymbol{\omega}\|_2 \leq h} \frac{1}{h^4} K^2\left(\frac{\boldsymbol{x}-\boldsymbol{\omega}}{h}\right) p(\boldsymbol{x}) \mathrm{d}\boldsymbol{x} \\
&\xlongequal{\boldsymbol{v}=(\boldsymbol{x}-\boldsymbol{\omega})/h} \ell_{\boldsymbol{\omega}}^q M \int_{\|\boldsymbol{v}\|_2 \leq 1} \frac{1}{h^2} K^2(\boldsymbol{v}) p(\boldsymbol{\omega}+\boldsymbol{v}h) \mathrm{d}\boldsymbol{v} \\
&\leq \ell_{\boldsymbol{\omega}}^q M \frac{1}{h^2} \frac{\|\bar{p}\|_\infty}{\ell_{\boldsymbol{\omega}}^q} \int_{\|\boldsymbol{v}\|_2 \leq 1} K^2(\boldsymbol{v}) \mathrm{d}\boldsymbol{v} = \frac{M\|\bar{p}\|_\infty \|K\|_2^2}{h^2}.
\end{aligned}
\tag{10}
$$
$$
\tag{11}
$$

**Bounding the covering number of** $\mathcal{G}$. For any probability measure $Q$ on $\mathcal{Z}_{L,M}^q$ and any $\eta \in (0, \frac{\|K\|_\infty M}{h^2})$, we aim to bound the covering number of $\mathcal{G}$ with respect to $L_2(Q)$ distance. This requires relating the $L_2(Q)$ distance in $\mathcal{G}$ and the $\ell_2$ distance in $\mathbb{R}^2$. Specifically, for any $\boldsymbol{\omega}, \boldsymbol{\omega}' \in \Omega_{2h}$ and $\mu \in \mathcal{Z}_{L,M}^q$, we can assume without loss of generality that $\ell_{\boldsymbol{\omega}} \leq \ell_{\boldsymbol{\omega}'}$. In this case, we firstly observe that

$$
\begin{aligned}
&\left| \ell_{\boldsymbol{\omega}}^q \int K\left(\frac{\boldsymbol{x}-\boldsymbol{\omega}}{h}\right) \mathrm{d}\mu - \ell_{\boldsymbol{\omega}'}^q \int K\left(\frac{\boldsymbol{x}-\boldsymbol{\omega}'}{h}\right) \mathrm{d}\mu \right| \\
&\leq \left| \int \ell_{\boldsymbol{\omega}}^q \left[ K\left(\frac{\boldsymbol{x}-\boldsymbol{\omega}}{h}\right) - K\left(\frac{\boldsymbol{x}-\boldsymbol{\omega}'}{h}\right) \right] \mathrm{d}\mu \right| + \left| \int (\ell_{\boldsymbol{\omega}}^q - \ell_{\boldsymbol{\omega}'}^q) K\left(\frac{\boldsymbol{x}-\boldsymbol{\omega}'}{h}\right) \mathrm{d}\mu \right| \\
&\leq \ell_{\boldsymbol{\omega}}^q \int_{\Omega_{\ell_{\boldsymbol{\omega}}}} \frac{L_k}{h} \|\boldsymbol{\omega}-\boldsymbol{\omega}'\|_2 \mathrm{d}\mu + \int_{\Omega_{\ell_{\boldsymbol{\omega}'}}} (\ell_{\boldsymbol{\omega}'}^q - \ell_{\boldsymbol{\omega}}^q) \|K\|_\infty \mathrm{d}\mu \\
&\leq \ell_{\boldsymbol{\omega}}^q \frac{L_k}{h} \|\boldsymbol{\omega}-\boldsymbol{\omega}'\|_2 \mu(\Omega_{\ell_{\boldsymbol{\omega}}}) + \|K\|_\infty (\ell_{\boldsymbol{\omega}'}^q - \ell_{\boldsymbol{\omega}}^q) \mu(\Omega_{\ell_{\boldsymbol{\omega}'}}) \\
&\leq \frac{ML_k}{h} \|\boldsymbol{\omega}-\boldsymbol{\omega}'\|_2 + M\|K\|_\infty \left[ 1 - \left(\frac{\ell_{\boldsymbol{\omega}}}{\ell_{\boldsymbol{\omega}'}}\right)^q \right].
\end{aligned}
\tag{12}
$$

Since $\ell_{\boldsymbol{\omega}} \geq \ell_{\boldsymbol{\omega}'} - \|\boldsymbol{\omega} - \boldsymbol{\omega}'\|_2$, the last term of (12) can be bounded by using Claim D.2 and $\ell_{\boldsymbol{\omega}} \geq \ell_{\boldsymbol{\omega}'} - \|\boldsymbol{\omega} - \boldsymbol{\omega}'\|_2$ as

$$
\begin{aligned}
1 - \left(\frac{\ell_{\boldsymbol{\omega}}}{\ell_{\boldsymbol{\omega}}'}\right)^q &\leq (q \vee 1)\left(1 - \frac{\ell_{\boldsymbol{\omega}}}{\ell_{\boldsymbol{\omega}}'}\right) \\
&\leq \frac{q \vee 1}{\ell_{\boldsymbol{\omega}}'}\|\boldsymbol{\omega} - \boldsymbol{\omega}'\|_2 \\
&\leq \frac{q \vee 1}{h}\|\boldsymbol{\omega} - \boldsymbol{\omega}'\|_2.
\end{aligned}
\tag{13}
$$

Notice that in the last line, we applied the fact that since $\boldsymbol{\omega}' \in \Omega_{2h}$, $\ell_{\boldsymbol{\omega}'} = \|\boldsymbol{\omega} - \partial\Omega\|_2 - h \geq h$.

From now on, we use $q'$ to denote $q \vee 1$ for simplicity. Equations (12) and (13) imply that

$$
\left|\ell_{\boldsymbol{\omega}}^q \int K\left(\frac{\boldsymbol{x} - \boldsymbol{\omega}}{h}\right)\mathrm{d}\mu - \ell_{\boldsymbol{\omega}'}^q \int K\left(\frac{\boldsymbol{x} - \boldsymbol{\omega}'}{h}\right)\mathrm{d}\mu\right| \leq \frac{M(L_k + q'\|K\|_\infty)}{h}\|\boldsymbol{\omega} - \boldsymbol{\omega}'\|_2.
$$

Therefore, the difference between $g_{\boldsymbol{\omega}}(\mu)$ and $g_{\boldsymbol{\omega}'}(\mu)$ can be bounded by

$$
\begin{aligned}
|g_{\boldsymbol{\omega}}(\mu) - g_{\boldsymbol{\omega}'}(\mu)| &\leq \left|\ell_{\boldsymbol{\omega}}^q \int \frac{1}{h^2}K\left(\frac{\boldsymbol{x} - \boldsymbol{\omega}}{h}\right)\mathrm{d}\mu - \ell_{\boldsymbol{\omega}'}^q \int \frac{1}{h^2}K\left(\frac{\boldsymbol{x} - \boldsymbol{\omega}'}{h}\right)\mathrm{d}\mu\right| \\
&\quad + \left|\ell_{\boldsymbol{\omega}}^q \int \frac{1}{h^2}K\left(\frac{\boldsymbol{x} - \boldsymbol{\omega}}{h}\right)\mathrm{d}\mathbb{E}[\mu] - \ell_{\boldsymbol{\omega}'}^q \int \frac{1}{h^2}K\left(\frac{\boldsymbol{x} - \boldsymbol{\omega}'}{h}\right)\mathrm{d}\mathbb{E}[\mu]\right| \\
&\leq \frac{2M(L_k + q'\|K\|_\infty)}{h^3}\|\boldsymbol{\omega} - \boldsymbol{\omega}'\|_2.
\end{aligned}
$$

In this way, we have related the distance between $g_{\boldsymbol{\omega}}$ and $g_{\boldsymbol{\omega}'}$ to the distance between $\boldsymbol{\omega}$ and $\boldsymbol{\omega}'$. Now, for any $\eta \in (0, \frac{\|K\|_\infty M}{h^2})$, we can set $\epsilon = \frac{\eta h^3}{2M(L_K + q'\|K\|_\infty)}$. It is easy to verify that

$$
\epsilon < \frac{h^3}{2M(L_K + q'\|K\|_\infty)}\frac{\|K\|_\infty M}{h^2} = \frac{\|K\|_\infty}{2(L_K + q'\|K\|_\infty)}h < h.
$$

Hence, we can construct a $\epsilon$-covering of $\Omega_{2h}$ in the $\ell_2$ distance, denoted as $S$. It is easy to show that the covering number

$$
\mathscr{N}(\Omega_{2h}, \|\cdot\|_2, \epsilon) \leq \frac{2L^2}{\epsilon^2}.
$$

By definition, for any $\boldsymbol{\omega} \in \Omega_{2h}$, there exists $\boldsymbol{\omega}' \in S$, such that $\|\boldsymbol{\omega} - \boldsymbol{\omega}'\|_2 \leq \epsilon < h < \ell_{\boldsymbol{\omega}'}$. Therefore, for any measure $Q$ on $\mathcal{Z}_{L,M}^q$,

$$
\begin{aligned}
\|g_{\boldsymbol{\omega}}(\mu) - g_{\boldsymbol{\omega}'}(\mu)\|_{L_2(Q)} &\leq \sup_{\mu \in \mathcal{Z}_{L,M}^q}|g_{\boldsymbol{\omega}}(\mu) - g_{\boldsymbol{\omega}'}(\mu)| \\
&\leq \frac{2M(L_K + q'\|K\|_\infty)}{h^3}\|\boldsymbol{\omega} - \boldsymbol{\omega}'\|_2 \leq \frac{2M(L_K + q'\|K\|_\infty)}{h^3}\epsilon = \eta.
\end{aligned}
$$

In conclusion,

$$
\begin{aligned}
\mathscr{N}(\mathcal{G}, L_2(Q), \eta) &\leq \mathscr{N}\left(\Omega_{2h}, \|\cdot\|_2, \frac{\eta h^3}{2M(L_K + q'\|K\|_\infty)}\right) \\
&< \left(\frac{4LM(L_K + q'\|K\|_\infty)}{\eta h^3}\right)^2.
\end{aligned}
\tag{14}
$$

**Completing the proof.** With $\|g_{\boldsymbol{\omega}}(\mu)\|_\infty$, $\mathbb{E}[g_{\boldsymbol{\omega}}(\mu)^2]$ and the covering number of $\mathcal{G}$ bounded as in (9), (10) and (14), we can apply Theorems C.6 and C.7 with

$$
\begin{cases}
AB = \frac{4LM(L_K + q'\|K\|_\infty)}{h^3}; \\
B = \frac{\|K\|_\infty M}{h^2}; \\
\sigma^2 = \frac{M\|\bar{p}\|_\infty}{h^2}\|K\|_2^2; \\
\nu = 2.
\end{cases}
$$

This gives us the conclusion that with probability at least $1 - \delta$,

$$\sup_{\boldsymbol{\omega} \in \Omega_{2h}} \left| \frac{1}{n} \sum_{i=1}^{n} g_{\boldsymbol{\omega}}(\mu) \right| \lesssim \frac{2\|K\|_{\infty} M}{nh^2} \log \left( \frac{4L(L_K + q'\|K\|_{\infty})}{\delta h^2 \|K\|_2} \sqrt{\frac{M}{\|\bar{p}\|_{\infty}}} \right) +$$

$$\sqrt{\frac{2M\|\bar{p}\|_{\infty}}{n}} \frac{\|K\|_2}{h} \sqrt{\log \left( \frac{4L(L_K + q'\|K\|_{\infty})}{\delta h^2 \|K\|_2} \sqrt{\frac{M}{\|\bar{p}\|_{\infty}}} \right)}.$$

∎

## D.4    Proof of Theorem 3.6(b)

Part (b) of Theorem 3.6 can be proved in a similar, though slightly easier, fashion to part (a). We therefore provide a sketch of the proof and omit the details.

**Defining an auxiliary function class.**    For every $\tilde{\mu}$ and $\boldsymbol{\omega} \in \Omega$, define

$$g_{\boldsymbol{\omega}}(\tilde{\mu}) = \int_{\Omega} \frac{1}{h^2} K \left( \frac{\boldsymbol{x} - \boldsymbol{\omega}}{h} \right) \mathrm{d}\tilde{\mu} - \int_{\Omega} \frac{1}{h^2} K \left( \frac{\boldsymbol{x} - \boldsymbol{\omega}}{h} \right) \mathrm{d}\mathbb{E}[\tilde{\mu}],$$

and let $\mathcal{G} = \{g_{\boldsymbol{\omega}} : \boldsymbol{\omega} \in \Omega\}$. It is easy to verify that $\mathbb{E}[g] \equiv 0$ for all $\boldsymbol{\omega} \in \Omega$, and that

$$\|\breve{p}_h(\boldsymbol{\omega}) - \tilde{p}(\boldsymbol{\omega})\| = \sup_{g \in \mathcal{G}} \left| \frac{1}{n} \sum_{i=1}^{n} g(\mu_i) \right|.$$

**Bounding $\|g\|_{\infty}$ and $\mathbb{E}[g^2]$.**    Since $\tilde{\mu}$ and $\mathbb{E}[\tilde{\mu}]$ are normalized measures with a total mass of 1, $\|g\|_{\infty}$ can be bounded by

$$\|g\|_{\infty} \leq \frac{\|K\|_{\infty}}{h^2};$$

in the mean time, Assumption 3.3 (b) guarantees that $\mathbb{E}[g_{\boldsymbol{\omega}}(\tilde{\mu})^2]$ can be bounded by

$$\mathbb{E}[g_{\boldsymbol{\omega}}(\tilde{\mu})^2] \leq \frac{\|\tilde{p}\|_{\infty}\|K\|_2^2}{h^2}.$$

**Bounding the covering number of $\mathcal{G}$.**    We again apply the Lipchitz property of the kernel function $K(\cdot)$ to conclude that for any $\boldsymbol{\omega}, \boldsymbol{\omega}' \in \Omega$,

$$|g_{\boldsymbol{\omega}}(\tilde{\mu}) - g_{\boldsymbol{\omega}'}(\tilde{\mu})| \leq \frac{2L_K}{h^3} \|\boldsymbol{\omega} - \boldsymbol{\omega}'\|_2.$$

Hence, using a similar reasoning to the proof of part (a), we can bound the covering number of $\mathcal{G}$ by

$$\mathcal{N}(\mathcal{G}, L^2(Q), \eta) < \left( \frac{4LL_K}{\eta h^3} \right)^2.$$

**Completing the proof.**    Theorem 3.6 (b) is a direct corollary of Theorems C.6 and C.7 with the following choice of parameters:

$$\begin{cases} AB = \frac{4LL_K}{h^3}; \\ B = \frac{\|K\|_{\infty}}{h^2}; \\ \sigma^2 = \frac{\|\tilde{p}\|_{\infty}}{h^2}\|K\|_2^2; \\ \nu = 2. \end{cases}$$

## D.5    Proof of Theorems 3.7 and B.4

In this section, we provide the proof of Theorem B.4, which gives a minimax lower bound for estimating the weighted persistence intensity function. Theorem 3.7, which gives the minimax lower

370 bound for estimating the persistence density function, can be proved in a similar while simpler
371 fashion, so we omit its proof for brevity.

372 The main idea of this proof is to build a connection of weighted intensity function $\bar{p}(\cdot)$ and a
373 probability density function. First of all, we can observe the conclusion of Theorem C.10 holds true
374 also when the support for the density function is $\Omega$ instead of $[0, 1]^2$. Now, notice that for any $\boldsymbol{x} \in \Omega$,
375 we can define the following measure:

$$\mu_{\boldsymbol{x}} = M\delta_{\boldsymbol{x}}||\boldsymbol{x} - \partial\Omega||_2^{-q}. \tag{15}$$

376 It is easy to verify that $\mathsf{Pers}_q(\mu_{\boldsymbol{x}}) = M$, so $\mu_{\boldsymbol{x}} \in \mathcal{Z}_{L,M}^q$ . Therefore, for any estimator $\hat{p}_n$ :
377 $(\mathcal{Z}_{L,M}^q)^n \to \mathcal{F}$, we can construct the following estimator $\hat{f}_n$:

$$\hat{f}_n(\boldsymbol{x}_1, \boldsymbol{x}_2, ..., \boldsymbol{x}_n) = \hat{p}_n(\mu_{\boldsymbol{x}_1}, \mu_{\boldsymbol{x}_2}, ..., \mu_{\boldsymbol{x}_n}).$$

378 Theorem C.10 states that there exists a probability density function $f : \Omega \to \mathbb{R}$ with $||f||_{\infty,\infty}^r \leq B$
379 such that when $X_1, X_2, ..., X_n \sim$ i.i.d. $f$,

$$\mathbb{E}||\hat{f}_n(X_1, X_2, ..., X_n) - f||_\infty \geq O\left(n^{-\frac{r}{2r+2}}\right).$$

380 We can apply the probability density function $f$ to construct a probability measure on $\mathcal{Z}_{L,M}^q$. First,
381 define a map $\Phi : \Omega \to \mathcal{Z}_{L,M}^q$ by $\Phi(\boldsymbol{x}) = \mu_{\boldsymbol{x}}$ in (15). Impose a measure structure on $\mathcal{Z}_{L,M}^q$ by
382 pushforwarding the measure structure on $\Omega$, i.e. $\mathcal{Y} \subset \mathcal{Z}_{L,M}^q$ is measurable if and only if $\Phi^{-1}(\mathcal{Y})$ is
383 measurable in $\Omega$. Define a probability measure $P$ on $\mathcal{Z}_{L,M}^q$ as a pushforward measure, i.e., for any
384 measurable set $\mathcal{Y} \subset \mathcal{Z}_{L,M}^q$,

$$P(\mathcal{Y}) = \int_{\Phi^{-1}(\mathcal{Y})} f(\boldsymbol{x})\mathrm{d}\boldsymbol{x}.$$

385 Then from the change of variables,

$$\int_{\mathcal{Y}} g(\mu)dP(\mu) = \int_{\Phi^{-1}(\mathcal{Y})} g(\Phi(\boldsymbol{x}))f(\boldsymbol{x})\mathrm{d}\boldsymbol{x}.$$

386 Now, the intensity for $P$ can be represented as follows: let $p(\cdot)$ be the intensity function for $\mathbb{E}[\mu]$
387 when $\mu \sim P$, then for all $u \in \Omega$,

$$\bar{p}(\boldsymbol{u}) := \|\boldsymbol{u} - \partial\Omega\|_2^q p(\boldsymbol{u}) = Mf(\boldsymbol{u}). \tag{16}$$

388 To see this fact, consider any Borel set $\mathcal{A} \subset \Omega$. By definition, the expected measure $\mathbb{E}[\mu]$ satisfies

$$\mathbb{E}[\mu](\mathcal{A}) = \mathbb{E}[\mu(\mathcal{A})] = \int_{\mathcal{Z}_{L,M}^q} \mu(\mathcal{A})dP(\mu)$$

$$= \int_{\Phi^{-1}(\mathcal{Z}_{L,M}^q)} \Phi(\boldsymbol{x})(\mathcal{A})f(\boldsymbol{x})\mathrm{d}\boldsymbol{x}$$

$$= \int_{\Omega} \mu_{\boldsymbol{x}}(\mathcal{A})f(\boldsymbol{x})\mathrm{d}\boldsymbol{x}$$

$$= \int_{\Omega} M||\boldsymbol{x} - \partial\Omega||_2^{-q}\mathbf{1}\{\boldsymbol{x} \in \mathcal{A}\}f(\boldsymbol{x})\mathrm{d}\boldsymbol{x}$$

$$= \int_{\mathcal{A}} M||\boldsymbol{x} - \partial\Omega||_2^{-q}f(\boldsymbol{x})\mathrm{d}\boldsymbol{x}.$$

389 Since $\mathcal{A}$ can be any Borel set, we get $p(\boldsymbol{u}) = M||\boldsymbol{u} - \partial\Omega||_2^{-q}$ by definition, and Equation (16)
390 follows naturally. Since the $\ell_\infty$ difference between $\hat{f}_n$ and $f$ is lower bounded, we can obtain

$$\mathbb{E}_P \sup_{\boldsymbol{\omega}\in\Omega} \|\boldsymbol{\omega} - \partial\Omega\|_2^q|\hat{p}_n(\boldsymbol{\omega}) - p(\boldsymbol{\omega})| = M\mathbb{E}_f\|\hat{f}_n - f\|_\infty \geq O\left(n^{-\frac{r}{2r+2}}\right).$$

391 ∎

## D.6 Proof of Theorems and Corollaries regarding linear representations of the persistence measure

The theoretical results regarding linear representations of the persistence measure in Section 3.3 are rather direct applications of the theoretical results on estimating the persistence intensity and density functions. We therefore combine their proofs in this section.

**Proof of Theorem 3.8.** Theorem 3.5 directly implies that under Assumption 3.2, for any $\Psi \in \mathscr{F}_{2h,R}$, the bias of $\hat{\Psi}$ is bounded by

$$
\begin{aligned}
\left| \mathbb{E}[\hat{\Psi}] - \Psi \right| &= \left| \int_{\boldsymbol{\omega} \in \Omega} f(\boldsymbol{\omega})(\mathbb{E}[\hat{p}_h(\boldsymbol{\omega})] - p(\boldsymbol{\omega})) \mathrm{d}\boldsymbol{\omega} \right| \\
&\leq \int_{\boldsymbol{\omega} \in \Omega} f(\boldsymbol{\omega})|\mathbb{E}[\hat{p}_h(\boldsymbol{\omega})] - p(\boldsymbol{\omega})| \mathrm{d}\boldsymbol{\omega} \\
&\leq \sup_{\boldsymbol{\omega} \in \Omega} |\mathbb{E}[\hat{p}(\boldsymbol{\omega})] - p(\boldsymbol{\omega})| \int_{\boldsymbol{\omega} \in \Omega} f(\boldsymbol{\omega}) \mathrm{d}\omega \\
&\leq L_p h^s R \int_{\|\boldsymbol{v}\|_2 \leq 1} |K(\boldsymbol{v})| \|\boldsymbol{v}\|_2^2 \mathrm{d}v,
\end{aligned}
$$

where in the last line we applied Theorem 3.5 and the definition of $\mathscr{F}_{2h,R}$. The upper bound for the bias of $\check{\Psi}$ follows similarly.

**Proof of Theorem 3.9.** The upper bound for the variation of $\hat{\Psi}$ is a direct corollary of Theorem 3.6 (a) and the fact that

$$
\begin{aligned}
\sup_{\Psi \in \mathscr{F}_{2h,R}} \left| \hat{\Psi} - \mathbb{E}[\hat{\Psi}] \right| &= \sup_{\Psi \in \mathscr{F}_{2h,R}} \left| \int_{\boldsymbol{\omega} \in \Omega} f(\boldsymbol{\omega})[\hat{p}_h(\boldsymbol{\omega}) - \mathbb{E}[\hat{p}_h](\boldsymbol{\omega})] \mathrm{d}\boldsymbol{\omega} \right| \\
&\leq \int_{\boldsymbol{\omega} \in \Omega} \ell_{\boldsymbol{\omega}}^{-q} f(\boldsymbol{\omega}) \mathrm{d}\boldsymbol{\omega} \cdot \sup_{\boldsymbol{\omega} \in \Omega} \ell_{\boldsymbol{\omega}}^q |\hat{p}_h(\boldsymbol{\omega}) - \mathbb{E}[\hat{p}_h](\boldsymbol{\omega})| \\
&\leq R \cdot \sup_{\boldsymbol{\omega} \in \Omega} \ell_{\boldsymbol{\omega}}^q |\hat{p}_h(\boldsymbol{\omega}) - \mathbb{E}[\hat{p}_h](\boldsymbol{\omega})| \, ;
\end{aligned}
$$

The upper bound for the variation of $\check{\Psi}$ follows from Theorem 3.6 (b) and a similar relation:

$$
\sup_{\check{\Psi} \in \mathscr{F}_R} \left| \check{\Psi} - \mathbb{E}[\check{\Psi}] \right| \leq R \cdot \sup_{\boldsymbol{\omega} \in \Omega} |\check{p}_h(\boldsymbol{\omega}) - \mathbb{E}[\check{p}_h(\boldsymbol{\omega})]|.
$$

**Proof of Corollaries 3.10 and 3.11.** For every $\boldsymbol{x} \in \Omega_{2h}$, define

$$
f_{\boldsymbol{x}}(\boldsymbol{\omega}) = \mathbb{1}\left\{ \boldsymbol{\omega} \in B_{\boldsymbol{x}} \right\},
$$

and let

$$
\mathscr{F}_{2h,R} = \left\{ \Psi = \int_{\Omega_{2h}} f_{\boldsymbol{x}}(\boldsymbol{\omega}) \mathrm{d}\mathbb{E}[\mu] \middle| \boldsymbol{x} \in \Omega_{2h} \right\}.
$$

Corollary 3.10 follows from Theorem 3.8 and the fact that

$$
\int_{\boldsymbol{\omega} \in \Omega_{2h}} f_{\boldsymbol{x}}(\boldsymbol{\omega}) \mathrm{d}\boldsymbol{\omega} \leq \frac{L^2}{4}
$$

for every $\boldsymbol{x} \in \Omega_{2h}$. Similarly, Corollary 3.11 follows from Theorem 3.9 and the fact that

$$
\int_{\boldsymbol{\omega} \in \Omega_{2h}} \ell_{\boldsymbol{\omega}}^{-q} f_{\boldsymbol{x}}(\boldsymbol{\omega}) \mathrm{d}\boldsymbol{\omega} \leq C \ell_{\boldsymbol{x}}^{2-q},
$$

for a constant $C$.

409 **Proof of Corollary 3.12.** For every $\boldsymbol{x} \in \Omega$, we define

$$f_{\boldsymbol{x}}(\boldsymbol{\omega}) = \mathbb{1}\left\{\boldsymbol{\omega} \in B_{\boldsymbol{x}}\right\},$$

410 and let

$$\widetilde{\mathscr{F}}_R = \left\{\widetilde{\Psi} = \int_{\Omega} f_{\boldsymbol{x}} \boldsymbol{\omega} \mathrm{d}\mathbb{E}[\tilde{\mu}] \,\middle|\, \boldsymbol{x} \in \Omega\right\}.$$

411 Corollary 3.12 follows directly from Theorem 3.9 and the fact that for every $\boldsymbol{x} \in \Omega$,

$$\int_{\boldsymbol{\omega} \in \Omega} f_{\boldsymbol{x}}(\boldsymbol{\omega}) \mathrm{d}\boldsymbol{\omega} \le \frac{L^2}{4}.$$

412 ## D.7   Proof of Theorem B.5

413 This proof again involves the Talagrand's inequality, and therefore takes a similar shape to the proof
414 of Theorem 3.6. We begin by defining an auxiliary function class.

415 **Defining the auxiliary function class $\mathcal{G}$.**   Recall that we choose the weight function as $f(\boldsymbol{\omega}) =$
416 $\|\boldsymbol{\omega} - \partial\Omega\|_2^q$. Therefore, for any persistence measure $\mu \in \mathcal{Z}_{L,M}^q$, its corresponding persistence surface
417 is characterized by

$$\rho_h(\mu)(\boldsymbol{u}) = \int_{\Omega} \|\boldsymbol{\omega} - \partial\Omega\|_2^q \frac{1}{h^2} K\left(\frac{\boldsymbol{u} - \boldsymbol{\omega}}{h}\right) \mathrm{d}\mu(\boldsymbol{\omega});$$

418 hence, by defining

$$g_{\boldsymbol{u}}(\mu) = \int_{\Omega} \|\boldsymbol{\omega} - \partial\Omega\|_2^q \frac{1}{h^2} K\left(\frac{\boldsymbol{u} - \boldsymbol{\omega}}{h}\right) \mathrm{d}\left(\mu - \mathbb{E}[\mu]\right)(\boldsymbol{\omega})$$

419 and letting $\mathcal{G} = \{g_{\boldsymbol{u}}(\boldsymbol{\mu}) : \boldsymbol{u} \in \Omega\}$, we observe that $\mathbb{E}[g] = 0$ for all $g \in \mathcal{G}$ and

$$\|\rho_h(\boldsymbol{\mu}_n) - \mathbb{E}[\rho_h(\boldsymbol{\mu})]\|_\infty = \sup_{g \in \mathcal{G}} \left\|\frac{1}{n}\sum_{i=1}^{n} g(\mu_i)\right\|.$$

420 **Bounding $\|g\|_\infty$ and $\mathbb{E}[g^2]$.**   Assumptions 3.4 and B.3 directly implies that for any $g \in \mathcal{G}$ and any
421 $\boldsymbol{u} \in \Omega$,

$$|g_{\boldsymbol{u}}(\mu)| \le \frac{\|K\|_\infty}{h^2} \max\left\{\int_{\Omega} \|\boldsymbol{\omega} - \partial\Omega\|_2^q \mathrm{d}\mu, \int_{\Omega} \|\boldsymbol{\omega} - \partial\Omega\|_2^q \mathrm{d}\mathbb{E}[\mu]\right\}$$

$$= \frac{\|K\|_\infty}{h^2} \max\left\{\mathsf{Pers}_q(\mu), \mathsf{Pers}_q(\mathbb{E}[\mu])\right\} \le \frac{M\|K\|_\infty}{h^2}.$$

422 Regarding the variance of $g$, Assumption 3.3 implies that

$$\mathbb{E}[g_{\boldsymbol{u}}(\mu)^2] \le \|g\|_\infty \cdot \int_{\Omega} \|\boldsymbol{\omega} - \partial\Omega\|_2^q \frac{1}{h^2} \left|K\left(\frac{\boldsymbol{u} - \boldsymbol{\omega}}{h}\right)\right| \mathrm{d}\mathbb{E}[\mu]$$

$$\le \frac{M\|K\|_\infty}{h^2} \int_{\Omega} \frac{1}{h^2} \left|K\left(\frac{\boldsymbol{u} - \boldsymbol{\omega}}{h}\right)\right| \|\boldsymbol{\omega} - \partial\Omega\|_2^q p(\boldsymbol{\omega}) \mathrm{d}\boldsymbol{\omega}$$

$$\le \frac{M\|K\|_\infty}{h^2} \int_{\|\boldsymbol{v}\|_2 \le 1} |K(\boldsymbol{v})| \,\mathrm{d}\boldsymbol{v} \cdot \sup_{\boldsymbol{\omega} \in \Omega} \|\boldsymbol{\omega} - \partial\Omega\|_2^q p(\boldsymbol{\omega})$$

$$\le \frac{M\|K\|_1 \|K\|_\infty \|\bar{p}\|_\infty}{h^2},$$

423 where in the third line we applied the change of variable $\boldsymbol{v} = (\boldsymbol{u} - \boldsymbol{\omega})/h$, and let

$$\|K\|_1 := \int_{\|\boldsymbol{v}\|_2 \le 1} |K(\boldsymbol{v})| \,\mathrm{d}\boldsymbol{v}.$$

**Covering number of $\mathcal{G}$.** Similar to the proof of Theorem 3.6, we bound the covering number of $\mathcal{G}$ by the Lipchitz property of the kernel function $K$. For any two points $\boldsymbol{u}, \boldsymbol{u}' \in \Omega$, Assumption B.3 guarantees that

$$\left| K\left(\frac{\boldsymbol{u} - \boldsymbol{\omega}}{h}\right) - K\left(\frac{\boldsymbol{u}' - \boldsymbol{\omega}}{h}\right) \right| \leq \frac{L_K \|\boldsymbol{u} - \boldsymbol{u}'\|_2}{h}.$$

Therefore, it is easy to verify that

$$|g_{\boldsymbol{u}}(\mu) - g_{\boldsymbol{u}'}(\mu)| \leq \frac{ML_K \|\boldsymbol{u} - \boldsymbol{u}'\|_2}{h^3}.$$

A similar reasoning to the proof of Theorem 3.6 yields that the covering number of $\mathcal{G}$ is upper bounded by

$$\mathcal{N}(\mathcal{G}, L^2(Q), \eta) \leq \mathcal{N}\left(\Omega, \|\cdot\|_2, \frac{\eta h^3}{ML_K}\right) \leq 2\left(\frac{LML_K}{\eta h^3}\right)^2.$$

**Completing the proof.** Theorem B.5 is a direct application of Theorems C.6 and C.7 with the following choice of parameters:

$$\begin{cases} AB = \frac{2LML_k}{h^3}; \\ B = \frac{M\|K\|_\infty}{h^2}; \\ \sigma^2 = \frac{M\|K\|_1 \|K\|_\infty \|\bar{p}\|_\infty}{h^2}; \\ \nu = 2. \end{cases}$$

## D.8 Proof of Theorems B.1 and B.2

Observe that the persistence diagram of the Vietoris-Rips filtration of $\boldsymbol{X} = (\boldsymbol{X}_1, \boldsymbol{X}_2, ..., \boldsymbol{X}_N)$ is decided purely by $\{\varphi[J](\boldsymbol{X})\}_{J \subset [N], |J|=2}$, in which

$$\varphi[J](\boldsymbol{X}) = \|\boldsymbol{X}_i - \boldsymbol{X}_j\|_2,$$

for $J = \{i, j\}$. In what follows, we firstly focus on the proof of Theorem B.1, and apply the techniques to that of Theorem B.2 in a similar manner.

**Proof of Theorem B.1.** Propositions C.4 and C.3 imply that for any Borel set $B \subseteq \Omega$,

$$
\begin{aligned}
\mathbb{E}[\mu](B) &= \sum_{r=1}^{R} \sum_{i=1}^{N_r} \sum_{s \in S} \int_{V_r \cap W^s_{J^1_{ir}, J^2_{ir}} \cap \Phi[J^1_{ir}, J^2_{ir}]^{-1}(B)} \kappa(\boldsymbol{X}) \mathrm{d}\boldsymbol{X} \\
&= \sum_{r=1}^{R} \sum_{i=1}^{N_r} \sum_{s \in S} \\
&\int_{\Psi^s_{J^1_{ir}, J^2_{ir}}(V_r \cap W^s_{J^1_{ir}, J^2_{ir}} \cap \Phi[J^1_{ir}, J^2_{ir}]^{-1}(B))} \kappa((\Psi^s_{J^1_{ir}, J^2_{ir}})^{-1}(u, y)) J[\Psi^s_{J^1_{ir}, J^2_{ir}}]^{-1}(\boldsymbol{u}, \boldsymbol{Y}) \mathrm{d}\boldsymbol{Y} \mathrm{d}u,
\end{aligned}
$$

438 where in the second line we change the variable from $\boldsymbol{X} \in [0,1]^{d \times n}$ to $(\boldsymbol{Y}, \boldsymbol{u})$ with $\boldsymbol{Y} \in [0,1]^{nd-2}$
439 and $\boldsymbol{u} \in \Omega$. Now, a change of order of summation gives

$$
\mathbb{E}[\mu](B) = \sum_{s \in S} \sum_{\substack{J_1, J_2 \subset [N] \\ |J_1|=|J_2|=2 \\ J_1 \neq J_2}} \sum_{r=1}^{R} \sum_{i=1}^{N_r} I(J_{ir}^1 = J_1, J_{ir}^2 = J_2)
$$
$$
\times \int_{\Psi_{J_1,J_2}^s(V_r \cap W_{J_1,J_2}^s \cap \Phi[J_1,J_2]^{-1}(B))} \kappa((\Psi_{J_1,J_2}^s)^{-1}(\boldsymbol{u}, \boldsymbol{Y})) J[\Psi_{J_{ir}^1, J_{ir}^2}^s]^{-1}(u,y) \mathrm{d}\boldsymbol{Y} \, \mathrm{d}\boldsymbol{u}
$$
$$
\leq \sum_{s \in S} \sum_{\substack{J_1, J_2 \subset [N] \\ |J_1|=|J_2|=2 \\ J_1 \neq J_2}} \sum_{r=1}^{R} \sum_{i=1}^{N_r} I(J_{ir}^1 = J_1, J_{ir}^2 = J_2)
$$
$$
\times \int_{\Psi_{J_1,J_2}^s(V_r \cap W_{J_1,J_2}^s \cap \Phi[J_1,J_2]^{-1}(B))} d \sup \kappa \, \mathrm{d}\boldsymbol{Y} \, \mathrm{d}\boldsymbol{u}
$$
$$
\leq \sum_{s \in S} \sum_{\substack{J_1, J_2 \subset [N] \\ |J_1|=|J_2|=2 \\ J_1 \neq J_2}} N(B) \int_{\Psi_{J_1,J_2}^s(W_{J_1,J_2}^s \cap \Phi[J_1,J_2]^{-1}(B))} d \sup \kappa \, \mathrm{d}\boldsymbol{Y} \, \mathrm{d}\boldsymbol{u}, \tag{17}
$$

440 where $N(B)$ is the number of persistent homology points in $B$, and in the second line we use the
441 facts that $\{V_r\}_{r=1}^{R}$ are disjoint, $\kappa \leq \sup \kappa$ and $J[\Psi_{J_{ir}^1, J_{ir}^2}^s]^{-1} \leq d$. Hence, bounding $\mathbb{E}[\mu](B)$ boils
442 down to characterizing the domain of integration on the right hand side of (17). For this, notice that
443 by definition,

$$
(\boldsymbol{Y}, \boldsymbol{u}) \in \Psi_{J_1,J_2}^s(W_{J_1,J_2}^s \cap \Phi[J_1,J_2]^{-1}(B))
$$
$$
\leftrightarrow \exists \boldsymbol{X} \in W_{J_1,J_2}^s, \text{ such that } \Phi[J_1,J_2](\boldsymbol{X}) \in B, \Psi_{J_1,J_2}^s(\boldsymbol{X}) = (\boldsymbol{Y}, \boldsymbol{u})
$$
$$
\rightarrow \exists \boldsymbol{X} \in W_{J_1,J_2}^s, \text{ such that } \Phi[J_1,J_2](\boldsymbol{X}) \in B, \Phi[J_1,J_2](\boldsymbol{X}) = \boldsymbol{u}, \text{ and } \boldsymbol{Y} \in [0,1]^{Nd-2}
$$
$$
\rightarrow \boldsymbol{u} \in B, \text{ and } \boldsymbol{Y} \in [0,1]^{Nd-2}.
$$

444 Hence, $\mathbb{E}[\mu](B)$ is upper bounded by

$$
\mathbb{E}[\mu](B) \leq N(B) \sum_{s \in S} \sum_{\substack{J_1, J_2 \subset [N] \\ |J_1|=|J_2|=2 \\ J_1 \neq J_2}} \int_{\boldsymbol{u} \in B, \boldsymbol{Y} \in [0,1]^{Nd-2}} d \sup \kappa \, \mathrm{d}\boldsymbol{Y} \, \mathrm{d}\boldsymbol{u}
$$
$$
= d \sup \kappa N(B) \sum_{s \in S} \sum_{\substack{J_1, J_2 \subset [N] \\ |J_1|=|J_2|=2 \\ J_1 \neq J_2}} \int_{[0,1]^{Nd-2}} \mathrm{d}\boldsymbol{Y} \int_B \mathrm{d}\boldsymbol{u}
$$
$$
= d \sup \kappa N(B) \sum_{s \in S} \sum_{\substack{J_1, J_2 \subset [N] \\ |J_1|=|J_2|=2 \\ J_1 \neq J_2}} \int_B \mathrm{d}\boldsymbol{u}.
$$

445 This effectively means that the intensity function $p(\boldsymbol{u})$ is upper bounded by

$$
p(\boldsymbol{u}) \leq \mathbb{E}[N(\{\boldsymbol{u}\})] \, d \sup \kappa \sum_{s \in S} \sum_{\substack{J_1, J_2 \subset [N] \\ |J_1|=|J_2|=2 \\ J_1 \neq J_2}} 1
$$
$$
< \mathbb{E}[N(\{\boldsymbol{u}\})] \operatorname{card}(S) |\{(J_1, J_2) : |J_1| = |J_2| = 2, J_1 \neq J_2, J_1 \subset [N], J_2 \subset [N]\}| d \sup \kappa.
$$

446 Now, $N(\{\boldsymbol{u}\}) \leq N_\ell$, so Lemma C.5 implies $\mathbb{E}[N(\{\boldsymbol{u}\})] \leq CN$. And $\operatorname{card}(S) \leq 4d^2$ and
447 $|\{(J_1, J_2) : |J_1| = |J_2| = 2, J_1 \neq J_2, J_1 \subset [N], J_2 \subset [N]\}| \leq \frac{N^4}{4}$, so

$$
p(\boldsymbol{u}) \leq (CN) \cdot (4d^2) \cdot \left(\frac{N^4}{4}\right) \cdot d \sup \kappa
$$
$$
= C' N^5 d^3 \sup \kappa.
$$

448 Theorem B.1 follows with the choice of

$$\mathsf{poly}(N, d) = N^5 d^3.$$

449 **Proof of Theorem B.2.** Propositions C.4 and C.3 imply that for any Borel set $B \subseteq \Omega$, the
450 normalized persistence measure of $B$ is expressed by

$$\mathbb{E}[\tilde{\mu}](B) = \sum_{r=1}^{R} \frac{1}{N_r} \sum_{i=1}^{N_r} \sum_{s \in S} \int_{V_r \cap W^s_{J^1_{ir}, J^2_{ir}} \cap \Phi[J^1_{ir}, J^2_{ir}]^{-1}(B)} \kappa(\boldsymbol{X}) \mathrm{d}\boldsymbol{X}$$

$$\leq \sum_{r=1}^{R} \max_{1 \leq i \leq N_r} \sum_{s \in S} \int_{V_r \cap W^s_{J^1_{ir}, J^2_{ir}} \cap \Phi[J^1_{ir}, J^2_{ir}]^{-1}(B)} \kappa(\boldsymbol{X}) \mathrm{d}\boldsymbol{X}.$$

451 Hence, same techniques can be applied to show that the persistence density function is upper bounded
452 by

$$\tilde{p}(\boldsymbol{u}) \leq d \sup \kappa \mathbb{E}\left[ \frac{N(\{\boldsymbol{u}\})}{N(\{\boldsymbol{u}\})} \right] \sum_{s \in S} \sum_{\substack{J_1, J_2 \subset [N] \\ |J_1| = |J_2| = 2 \\ J_1 \neq J_2}} 1$$

$$\leq d \sup \kappa \max_{1 \leq i \leq N(\boldsymbol{u})} \sum_{s \in S} \sum_{\substack{J_1, J_2 \subset [N] \\ |J_1| = |J_2| = 2 \\ J_1 \neq J_2}} 1$$

$$\leq \mathrm{card}(S) |\{(J_1, J_2) : |J_1| = |J_2| = 2, J_1 \neq J_2, J_1 \subset [N], J_2 \subset [N]\}| d \sup \kappa$$

$$\leq (4d^2) \cdot \left( \frac{N^4}{4} \right) \cdot d \sup \kappa.$$

453 Theorem B.2 follows from choosing

$$\mathsf{poly}(N, d) = N^4 d^3.$$

### D.9 Proof of Theorem B.6

455 In this proof, we firstly define an auxiliary family of functions, and then verify the conditions in
456 Theorem C.8.

457 **Defining the auxiliary function class.** For every $\boldsymbol{x} \in \Omega_\ell$ and $\mu \in \mathcal{Z}^q_{L,M}$, define

$$g_{\boldsymbol{x}}(\mu) = \mu(B_{\boldsymbol{x}}) - \mathbb{E}[\mu](B_{\boldsymbol{x}}), \tag{18}$$

458 and let $\mathcal{G} = \{g_{\boldsymbol{x}} : \boldsymbol{x} \in \Omega_\ell\}$. It is easy to verify that $\mathbb{E}[g_{\boldsymbol{x}}(\mu)] = 0$ for all $\boldsymbol{x} \in \Omega_\ell$, and that

$$\sup_{\boldsymbol{x} \in \Omega_\ell} \left| \hat{\beta}_{\boldsymbol{x}} - \mathbb{E}[\hat{\beta}_{\boldsymbol{x}}] \right| = \left| \sup_{g \in \mathcal{G}} \frac{1}{n} \sum_{i=1}^{n} g(\mu_i) \right|.$$

459 **Bounding $||g_x||_\infty$ and $\mathbb{E}[g_x(\mu)^2]$.** For any $\boldsymbol{x} \in \Omega_\ell$, the set $B_{\boldsymbol{x}}$ is contained in $\Omega_\ell$. Hence for any
460 $\mu \in \mathcal{Z}^q_{L,M}$, $\mu(B_{\boldsymbol{x}})$ and $\mathbb{E}[\mu](B_{\boldsymbol{x}})$ can be bounded as

$$\mu(B_{\boldsymbol{x}}) \leq \mu(\Omega_\ell) \leq \ell^{-q} \mathsf{Pers}_q(\mu) \leq M\ell^{-q},$$
$$\mathbb{E}[\mu](B_{\boldsymbol{x}}) \leq \mathbb{E}[\mu](\Omega_\ell) \leq \ell^{-q} \mathsf{Pers}_q(\mathbb{E}[\mu]) \leq M\ell^{-q}. \tag{19}$$

461 Hence $\|g_{\boldsymbol{x}}\|_\infty$ can be bounded as

$$\|g_{\boldsymbol{x}}\|_\infty \leq \sup_{\mu \in \mathcal{Z}^q_{L,M}} \max\{\mu(B_{\boldsymbol{x}}), \mathbb{E}[\mu](B_{\boldsymbol{x}})\} \leq M\ell^{-q}. \tag{20}$$

462 As for the variance of $g_x(\mu)$, we firstly observe that

$$\mathbb{E}[g_{\boldsymbol{x}}(\mu)^2] \leq \|g_{\boldsymbol{x}}\|_\infty \mathbb{E}[\mu](B_{\boldsymbol{x}}) \tag{21}$$

Now, apart from using the bound $\mathbb{E}[\mu](B_{\boldsymbol{x}}) \leq M\ell^{-q}$ from (19), we can also have tighter bound with respect to $\ell$ when $q > 1$. To do this, we again take the coordinate transformation

$$\begin{cases} y_1 = \frac{x_2 - x_1}{\sqrt{2}} = \|\boldsymbol{x} - \partial\Omega\|_2, \\ y_2 = \frac{x_2 + x_1}{\sqrt{2}}. \end{cases}$$

It can be easily verified that the determinant of the Jacobian matrix between $\boldsymbol{x}$ and $\boldsymbol{y}$ coordinates is 1, and that the $\Omega_\ell$ can be represented using $\boldsymbol{y}$ coordinates by

$$\Omega_\ell = \left\{ (y_1, y_2) : \ell < y_1 \leq \frac{L}{\sqrt{2}}, y_1 \leq y_2 \leq \sqrt{2}L - y_1 \right\}.$$

Then, we have a tighter bound with respect to $\ell$ of $\mathbb{E}[\mu](B_{\boldsymbol{x}})$ when $q > 1$ as

$$\mathbb{E}[\mu](B_{\boldsymbol{x}}) \leq \mathbb{E}[\mu](\Omega_\ell) = \int_{\Omega_\ell} p(\boldsymbol{u}) \mathrm{d}\boldsymbol{u}$$

$$= \int_{\Omega_\ell} \|\boldsymbol{u} - \partial\Omega\|_2^{-q} \bar{p}(\boldsymbol{u}) \mathrm{d}\boldsymbol{u}$$

$$\leq \|\bar{p}\|_\infty \int_\ell^{\frac{L}{\sqrt{2}}} \left( \int_{y_1}^{\sqrt{2}L - y_1} \mathrm{d}y_2 \right) y_1^{-q} \mathrm{d}y_1$$

$$\leq \|\bar{p}\|_\infty \int_\ell^{\frac{L}{\sqrt{2}}} \sqrt{2} L y_1^{-q} \mathrm{d}y_1$$

$$\leq \frac{\sqrt{2} L \ell^{1-q} \|\bar{p}\|_\infty}{q - 1}.$$

Hence when we let $(q - 1)_+ = \max\{q - 1, 0\}$,

$$\mathbb{E}[\mu](B_{\boldsymbol{x}}) \leq \min \left\{ M\ell^{-q}, \frac{\sqrt{2} L \ell^{1-q} \|\bar{p}\|_\infty}{(q - 1)_+} \right\}. \tag{22}$$

And hence by applying (22) to (21), the variance of $g_x(\mu)$ can be upper bounded as

$$\mathbb{E}[g_{\boldsymbol{x}}(\mu)^2] \leq \|g_{\boldsymbol{x}}\|_\infty \mathbb{E}[\mu](B_{\boldsymbol{x}})$$

$$\leq \min \left\{ M^2 \ell^{-2q}, \frac{\sqrt{2} M L \ell^{1-2q} \|\bar{p}\|_\infty}{(q - 1)_+} \right\} \tag{23}$$

**Polynomial discrimination of $\mathcal{G}$.** By definition, the empirical persistent measure $\mu_i$ can be represented as

$$\mu_i = \sum_j \delta_{\boldsymbol{r}_{ij}},$$

in which $\boldsymbol{r}_{ij} = (b_{ij}, d_{ij})$ represents the $j$-th point in the corresponding persistent diagram, with $b_{ij}$ and $d_{ij}$ being its birth and death weight respectively . Without loss of generality, we can sort the points in descending order of their distance to the diagonal $\partial\Omega$. Let $N_i = \mu_i(\Omega_\ell)$, then we have $N_i \leq M\ell^{-q}$. Hence, for every $\boldsymbol{x}$ with $\|\boldsymbol{x} - \partial\Omega\|_2 = \ell$, $\mu_i(B_{\boldsymbol{x}})$ can be represented as

$$\mu_i(B_{\boldsymbol{x}}) = \sum_{j=1}^{N_i} \mathbb{1}(b_{ij} < x_1) \mathbb{1}(d_{ij} > x_2). \tag{24}$$

With this expression, we are ready to bound the cardinality of $\mathcal{G}(\boldsymbol{\mu}_1^n)$. Notice that for any fixed $\boldsymbol{x}$, the value of the tuple $(g_x(\mu_1), ..., g_x(\mu_n))$ is completely decided by the Cartesian product of indicator functions

$$\{\mathbb{1}(b_{ij} < x_1)\}_{i \in [n], j \in [N_i]} \times \{\mathbb{1}(d_{ij} > x_2)\}_{i \in [n], j \in [N_i]} := S_b \times S_d.$$

It is easy to see that with the variation of $\boldsymbol{x} = (x_1, x_2)$, the number of different values taken by $S_b$ and $S_d$ can be bounded by

$$1 + \sum_{i=1}^n N_i \leq 1 + n \cdot M\ell^{-q}.$$

Hence, the cardinality of $\mathcal{G}(\boldsymbol{\mu}_1^n)$ is bounded by

$$\mathrm{Card}(\mathcal{G}(\boldsymbol{\mu})) \leq \left( M\ell^{-q} n + 1 \right)^2. \tag{25}$$

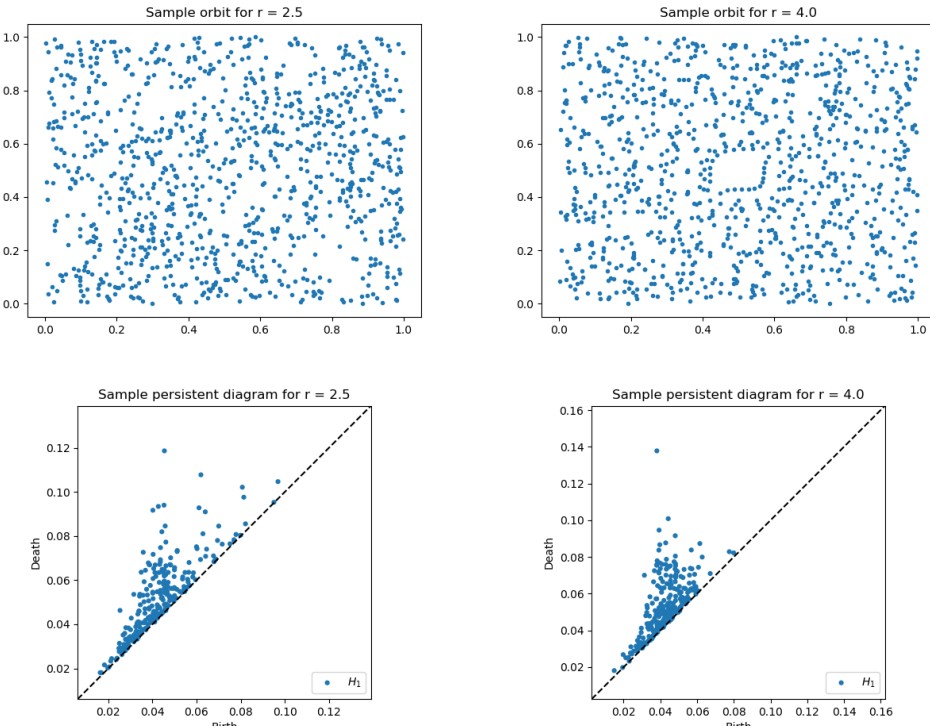

Figure 1: Top row: sample orbits from the ORBIT5K data set with $r = 2.5$ (left) and $r = 4.0$ (right). Bottom row: sample persistent diagrams.

**Completing the proof.** The theorem is a direct result for applying Theorem C.8 with the following parameters:

$$\begin{cases} A = M\ell^{-q}; \\ B = M; \\ \sigma^2 = \min\left\{M^2\ell^{-2q}, \frac{\sqrt{2}ML\ell^{1-2q}\|\bar{p}\|_\infty}{(q-1)_+}\right\}; \\ \nu = 2. \end{cases}$$

# E Experimental details

Figure 1 shows two ORBIT5K simulations with different values of $r$ (2.5 and 4) and the corresponding persistent diagrams. Figure 2 displays the kernel intensity functions for the ORBIT5K simulations set with $r = 2.5$ and $r = 4$ for varying sample sizes, while Figure 3 shows persistence density functions. Figures 4 and 5 show the Betti curves and estimated Betti curves using the kernel density function for the ORBIT5K simulations for $r = 2.5$ and $r = 4$.

Finally, Figure 6 displays the estimated persistence density functions computed over random draws of varying size of the digits "4" and "8" from the MNIST dataset.

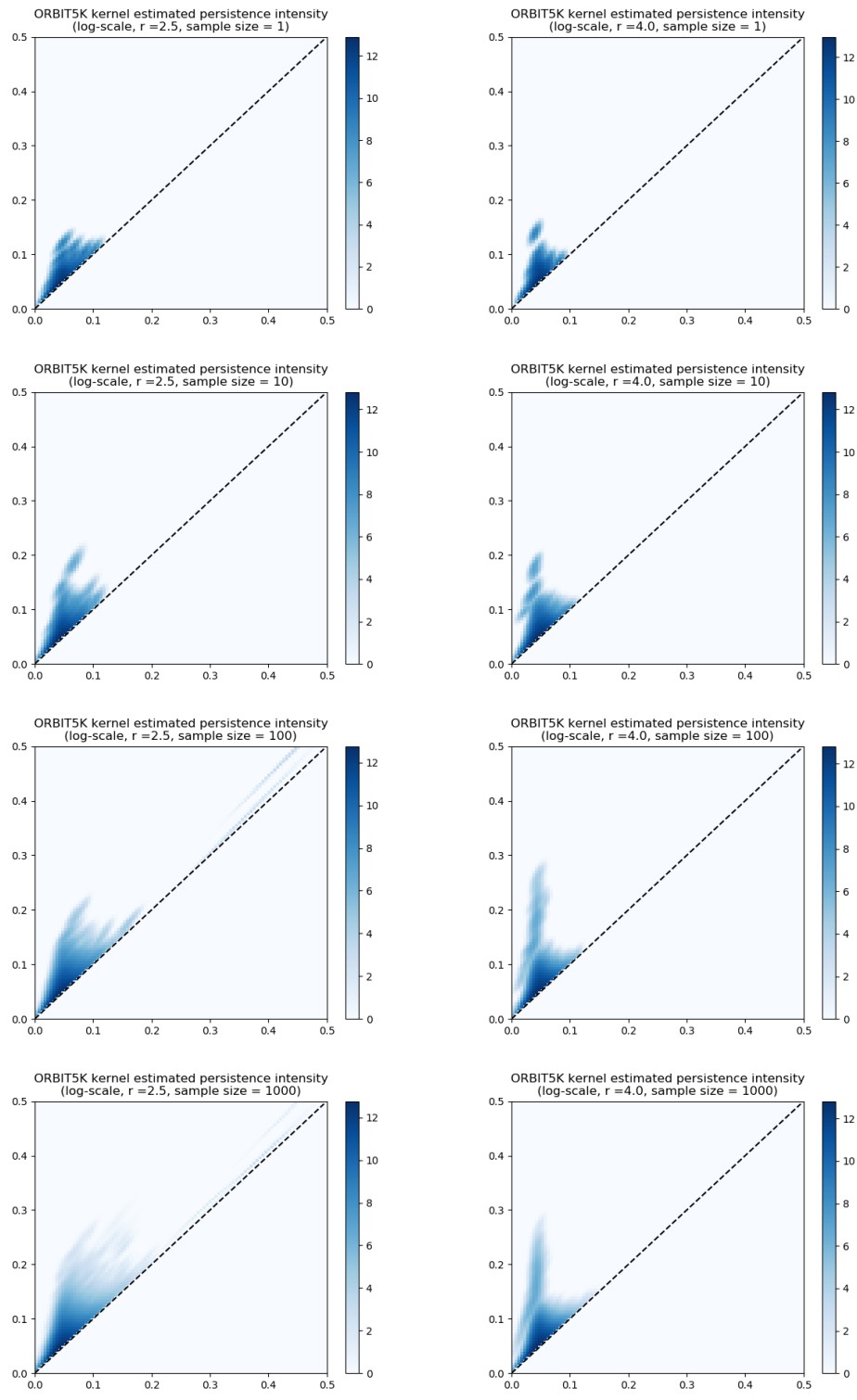

Figure 2: Kernel estimators for the persistence intensity function from the ORBIT5K data set with $r = 2.5$ (left) and $r = 4.0$ (right) and sample sizes 1, 10, 100 and 1000 (top to bottom).

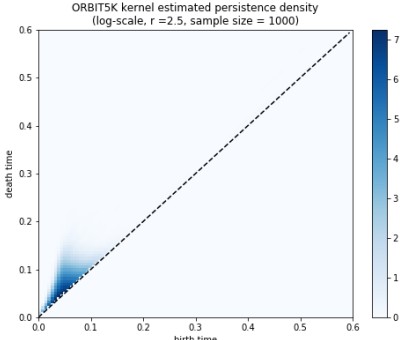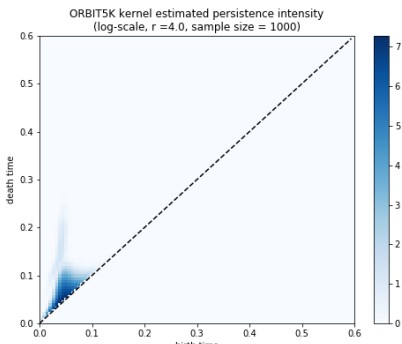

Figure 3: Kernel estimators for the persistence density function from the ORBIT5K data set with $r = 2.5$ (left) and $r = 4.0$ (right) and sample size $n = 1000$.

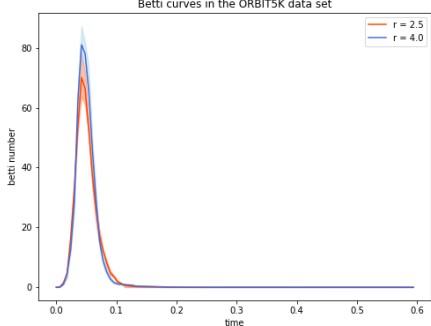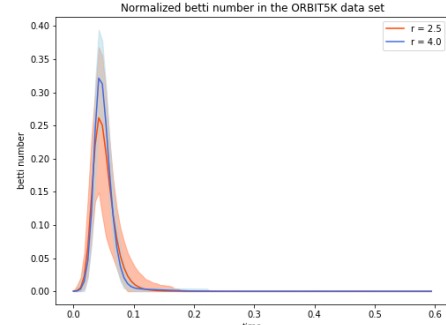

Figure 4: Empirical betti curves (left) and normalized betti curves (right) from the ORBIT5K data set with $r = 2.5$ and $r = 4.0$. Solid lines show sample average and the shades depict the lower and upper 2.5 percentiles.

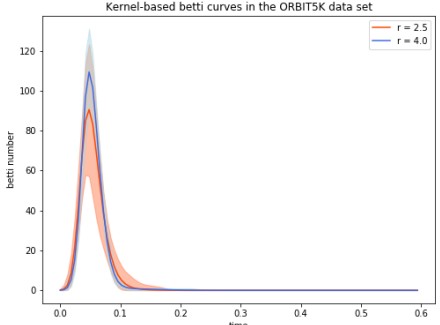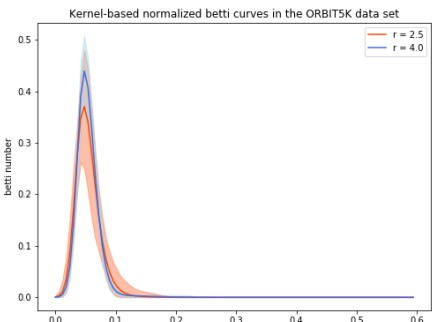

Figure 5: Kernel-based betti curves (left) and normalized betti curves (right) from the ORBIT5K data set with $r = 2.5$ and $r = 4.0$. Solid lines show sample average and the shades depict the lower and upper 2.5 percentiles.

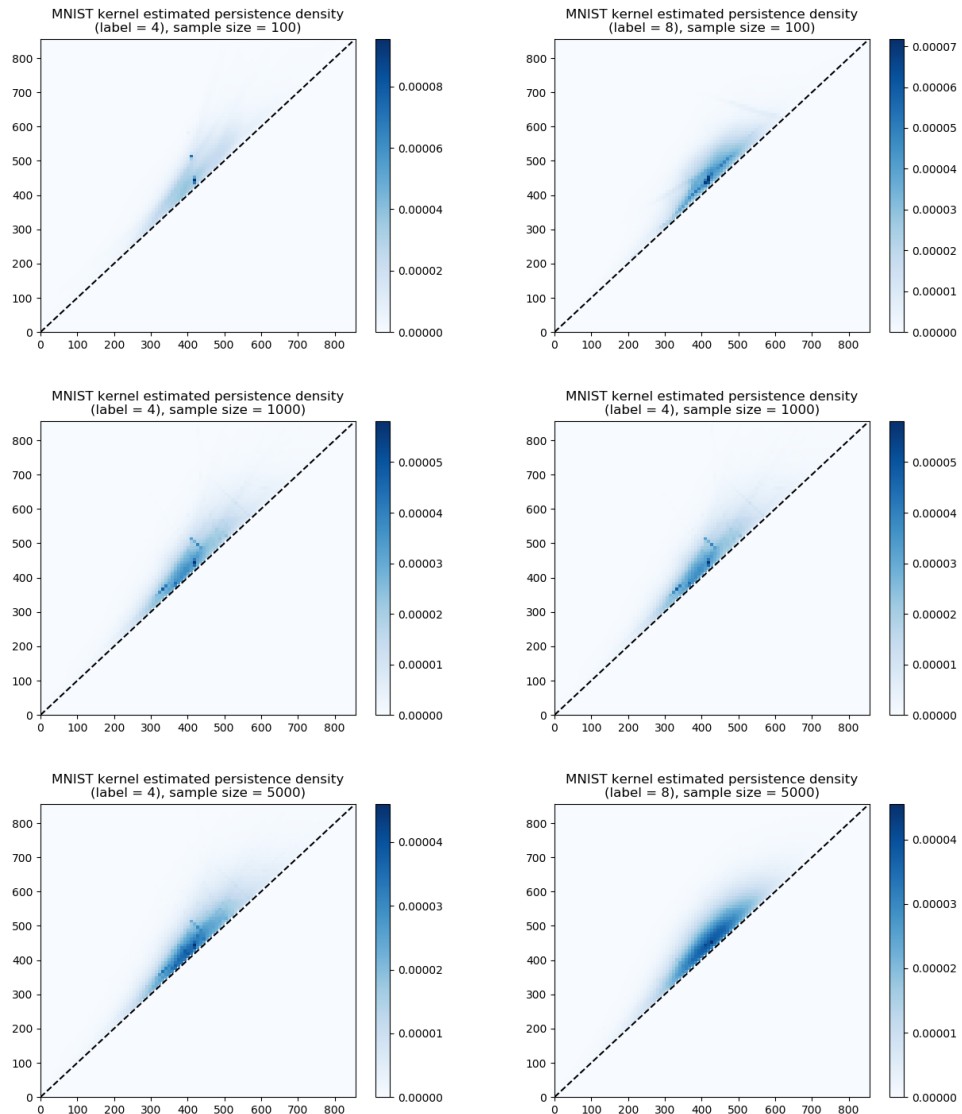

Figure 6: Kernel estimators for the persistence density function from the MNIST data set for the digits 4 (left column) and 8 (right column) based on random draws of sample sizes 100, 1000 and 5000 (top to bottom).