# OpenReview forum: "On the estimation of  persistence intensity functions and linear representations of persistence diagrams"
_NeurIPS.cc/2023/Conference — Submitted to NeurIPS 2023_

### Official Review · Reviewer_1g88 · 2023-07-03

**Soundness:** 2 fair
**Presentation:** 2 fair
**Contribution:** 1 poor
**Rating:** 3
**Confidence:** 5

**Summary:**

This paper considers the problem of estimating _persistence intensity (resp. density) functions_, which are topological summaries arising when considering multiples realizations $\mu_1,\dots,\mu_n$ of persistence diagrams---which are counting measures supported on an open-half-plane $\Omega$. Namely, (Chazal and Divol, 2019) proved that $\frac{1}{n} \sum_{i=1}^n \mu_i$ converges toward a limit object $E[\mu]$ called an _expected persistence diagram_ which, under mild assumptions, admits a density $p$ wrt the Lebesgue measure on $\Omega$, called a persistence _intensity_ function. Note that $p$ may not integrate to $1$ (the $\mu_i$s are *not* probability measures, only Radon measures).

It is natural to wonder how fast one can estimate $E[\mu]$ given an i.i.d. sample $\mu_1,\dots,\mu_n$. Given that $E[\mu]$ has a smooth density $p$, while the $(\mu_i)$ are discrete, it is tempting to adopt a kernel based approach (i.e. considering the convolution of the $\mu_i$ by a kernel $K_h$, with $h>0$ being the bandwidth), yielding an estimator $\hat{p} = \hat{p}_{h,n}$ of $p$. This is the purpose of the current paper which shows that, in brief, assuming $p$ is $s$-Hölder,

$$\| E[\hat{p}] - p \|_\infty \leq O(h^s), \qquad \text{(bias)},$$

and (with high probability)

$$ \| \hat{p} - E[\hat{p}] \|_{\infty , h} \leq O(n^{-\frac{1}{2}}h^{-1}), \qquad \text{(variance)},$$

where $\| \cdot \|_{\infty,h}$ denote the sup-norm but only accounting for points in $\Omega$ that are at distance $> 2h$ from the diagonal $\partial \Omega = \{ (t,t), t \in \mathbb{R} \}$.

As an alternative, they also study the persistence _density_ function, which is substantially similar to the above, but considering the quantity $\frac{1}{n} \sum_{i=1}^n \frac{\mu_i}{\mu_i(\Omega)}$ as a starting point, hence the limit object is a _probability_ distribution. This first normalisation step allows the authors to obtain similar results as those mentioned above, but stronger in the sense that they do not need to ignore points close to $\partial \Omega$.

**Strengths:**

## Clarity

The paper is fairly well written, though it may be hard to understand for the reader that is not familiar with statistical topological data analysis problems. Theorems are precisely stated, and, with few exceptions, mathematical quantities are well-defined.

## Originality and Significance

The authors provides quantitative results for the convergence of kernel-based density estimator for expected PD, which is new to the best of my knowledge. The introduction of the persistence _density_ may also be worth of interest, if further motivated (see below).

## Quality

This is a competent paper in terms of the results provided (which seem correct as far as I can tell). However, the motivation behind the type of results themselves is arguably questionable.

**Weaknesses:**

There are several points which fail to convince me and prevent me from supporting this paper yet.

## 1. The use of the sup-norm.

The paper provides results in the sup-norm on $\Omega$. It is important to stress that this norm does not account for the peculiar role played by the diagonal in the geometry of persistence diagrams, in contrast with the standard $\mathrm{OT}_p$ metric. The authors justify this by an inequality of the form $\mathrm{OT}_p^p(\cdot, \cdot) \leq C | \cdot - \cdot|$,
meaning that being small in sup-norm is (strictly, as proved by the authors) more demanding that being close in $\mathrm{OT}_p$ metric. [Edit: fix the rendering by removing the infty symbol; the norm in the rhs is the sup-norm.]

Sure, it means that the rate obtained for the sup-norm implies the same rate for the $\mathrm{OT}_p$ metric (up to the role played by the exponent), but it also means that the task is _harder_, and that this norm does not induce the same topology as the natural metrics on persistence diagrams. Recall (as suggested in $\ell$167-171) that accounting for the diagonal is not simply a trick to compare measures with different total masses, but also has an algebraic meaning (from which we get the mentioned stability results). The sup-norm induces a topology that fails to capture the fact that one can compare diagrams "downweighting points close to the diagonal".

To me, this is what prevents, for the persistence _intensity_ function, to obtained "global" sup bound, and get only bounds valid $2h$-away from the diagonal. The sup-norm cannot handle the noise properly.

In addition, because performing estimation in sup-norm is _harder_ than estimation in $\mathrm{OT}_p$-metric, one only get a convergence rate of $\frac{s}{2(s+1)}$, which is natural for the sup-norm, but quite _slow_ for the $\mathrm{OT}_p$ metric. Indeed, (DIvol and Lacombe, 2021) prove that the empirical expected persistence diagram (i.e. without any convolution by a kernel involved) converges toward the persistence _intensity_ function at (the faster) parametric rate $O(n^{-\frac{1}{2}})$. Of course, the latter result considers the weaker (but more natural) metric $\mathrm{OT}_p$, but as long as there is no very strong motivation to compare persistence intensity functions using a sup-norm, it is reasonable to wonder why should we struggle to obtain slower convergence rates.

## 2. Motivations behind the persistence _density_ function

As (interestingly) observed by the authors, statistical estimation improves when considering the normalized persistence _density_ function. However, here as well, I fail to be fully convinced by the proposed motivation, namely "the normalized persistence measure may be desirable when the number of points (...) is not of direct interest but their spatial distribution is" ($\ell$80-81).

I do not agree with this claim because this normalization typically discard points away from the diagonal, asymptotically. For, consider a $N$-sample on a sphere + some tubular noise, and the Vietoris-Rips filtration on it. Then (with high probability), the corresponding (random) persistence diagram in $H_2$ (degree-$2$ homology) will have one point away from the diagonal, and a bunch of points close to the diagonal accounting for the noise---so does the corresponding persistence _intensity_ function. As $N$ increases, the points accounting for the noise get closer and closer to the diagonal, but also more abundant. As such, if one normalize the persistence measure by its mass/number of points, the bump/point accounting for the "robust" topological information will asymptotically be erased. In particular, this normalized persistence measure is not continuous (for, say, the vague topology) with respect to the Hausdorff distance, a central property satisfied by the Vietoris-Rips filtration. (Note : this is what we can observe in Figure 3 vs Figure 2).

Therefore, (i) it is not surprising that one obtains stronger (this time) results with this weaker representation and (ii) it is not clear to me why would one actually consider this representation at is losses its topological interpretation, as far as I can tell.

## 3. About the experiments

The numerical illustrations have all been deferred to the supplementary material.

To me, the (main body of the) paper should be self-contained, in sense that one should not _have to_ look at the supplementary material to understand it at high level. Proofs, complementary results, _complementary_ experimental report can be deferred to the supplementary material, but having a **Numerical illustration** section without any numerical illustration, mostly saying "look at the supplementary material", is not serving the paper. Right now, the paper can be considered as experiment-less, and while numerical illustrations are not mandatory, this clearly does not support the paper.

Note that I looked at the experiments nonetheless. While they are fairly interesting, they do not bring more motivation to support the paper (with, e.g., a ML experiment where using the persistence _density_ function is much better than using the more standard persistence intensity function).

## 4. Complementary minor comments
- I think that there is a typo in the definition of $\Omega_\ell$, which is (I think) inconsistent with the description made below ($\ell$67) and its use in section 3.
- More references could have been cited through the paper, e.g., when listing different linear representations ($\ell$129-131), it may be nice to credit their respective authors.  Similarly, a more precise comparison with related work (mostly Divol's line of work with Polonik, Chazal and then Lacombe), would be helpful to understand what is the paper contribution and how it differs from these works.


**Questions:**

My main interrogation is about the motivation of the proposed objects (sup-norm, normalisation wrt the mass), as detailed in points 1. and 2. above.

The motivation can come from theoretical considerations, but also from numerical ones; e.g. an experiment showing that considering the sup-norm is more informative that the $\mathrm{OT}_p$ metric (I mean, it theoretically is---as discussed in section D1, but I do not see a _practical_ situation where it is important to observe that the sup-norm diverges while the $\mathrm{OT}_p$ converges).

**Limitations:**

I do not see any negative societal impact _specific_ to this work.

---

> ### Author Rebuttal · Authors · 2023-08-09
>
> **Motivation and rationale for the intensity function.**
>
> We thank the reviewer for the constructive and expertly crafted comments. We agree with all of them! We regard the issues raised by the reviewer as features and not drawbacks of our approach.  Please see our general comments for a clarification about the goals of our work and their relationship with current TDA methods.
>
> **The use of the sup-norm.**
>
> 1. An upper bound in $\ell_{\infty}$ error does indeed deliver a strong statistical guarantee that immediately applies to arbitrary (bounded) linear functionals. It is indeed stronger than the one afforded by the optimal transport distance, which, as the reviewer mentions, down-weights the points near the diagonal, likely to represent topological noise. Since our aim is to instead capture topological noise and its distribution, we regard this as a feature, not a shortcoming.
>
> 2. If we still wish to down-weight the points near the diagonal and emphasize the topological signature, our framework can be easily adapted, by considering a weighted intensity function (with weights proportional to any power of the distance to the diagonal). We explore this extension in Section B.4 in the supplementary material, where we derive statistical guarantees for estimating the persistence surface. We will comment on this extension in the main body of the text.
>
> 3. We also would like to emphasize that our approach is computationally appealing. Both the intensity and density functions are straightforward to compute. Furthermore, comparing different intensity/density functions is also straightforward. In contrast, computing and evaluating OT distances (even with perfect knowledge of the distributions) is computationally challenging, in general.
>
> 4. The reviewer is correct that with a bias-variance trade-off, the rage of convergence in $\ell_{\infty}$ norm is slower, of order $O\left(n^{-\frac{s}{2(s+1)}}\right)$, if we let the bandwidth vanish with the sample size. (This is an unavoidable, ubiquitous fact in non-parametric functional estimation). However, when the bandwidth $h$ is kept fixed, as is common in linear representations of the expected persistence measure like the persistence surface [DP19], the convergence rate is of order $O(n^{-\frac{1}{2}})$, as is shown in Section B.4 in the supplementary material. We will add more commentary to the paper in order and emphasize this fact.
>
>
> 5. Setting aside any issue or relevance and motivation, we would like to remark that our results are novel and original.  For instance, the result in Thm 3.1 that the optimal transport distance is bounded above by the $\ell_{\infty}$ distance between the intensity functions is a new contribution, and so is the observation (in section D.1) that, when $q=\infty$ the topology induced by the $\ell_{\infty}$ distance is strictly stronger.
>
>
> **Motivations behind the persistence density function.**
>
> The reviewer is correct that asymptotically, the points away from the diagonal would vanish in the persistence density function. With the caveat that this type of asymptotic behavior (whereby the number of points used to compute the persistence diagram increases without bounds) is *outside* the framework of our results, we still have that the primary distributional features of the topological noise, represented by points close to the diagonal in the persistent diagrams, are well-preserved. Since this a main focus of this work, we find that the persistence intensity function, despite the loss of information due to the normalization, remains a valid and potentially useful tool, which in addition enjoys better convergence guarantees.
> We agree with the reviewer that points in $\Omega$ that express topological features may not even have a positive persistence density, but, once again, we do not regard this necessarily as a limitation, but rather a feature.
>  See the example above about the difference in the distributions of the topological noise for persistence diagrams built from a uniform and non-uniform distribution over the sphere, as illustrated in Figure 1.
> We also agree that the normalized persistence measure is not continuous for the vague topology w.r.t Hausdorff distance (a fact that is not surprising, given our other results). We believe that a refined analysis of the topological properties of the normalized persistence measure in order would be an interesting (and probably subtle) problem to explore.
> We will provide better and clearer language to motivate the use, utility and limitations of the density function.
>
> **About the experiments.**
>
> Though, as pointed out by the reviewer,  our paper is primarily theoretical, we believe that the experiments do offer some insights on the properties of the intensity and density functions. We are not sure how to design experiments that demonstrate the utility of using persistence density/intensity functions over more traditional tools because we do not see them as competing and mutually exclusive - they are just different. We are happy to include the simulations depicted in Figure 1, which show two sample persistence diagrams from a uniform and non-uniform distribution over the sphere, along with the associated persistence intensity and density functions. These plots illustrate two settings in which the topological signals are nearly identical but the distribution of the topological noise is markedly different.
>
> **Complementary minor comments**
>
> 1. The reviewer is correct that there is a typo with regard to the definition of $\Omega_{\ell}$. The correct definition should be
> \begin{equation}
> \Omega_{\ell} = \\{\mathbf{x} \in \Omega: \min_{\mathbf{\omega} \in \partial \Omega} \left\Vert\mathbf{x} - \mathbf{\omega}\right\Vert_{2} \geq \ell \\}.
> \end{equation}
>
> 2. We will include more references and details about how our contribution differs from existing ones.

---

> > ### Comment · Reviewer_1g88 · 2023-08-12
> > **Thanks**
> >
> > Thank you for taking time addressing my comments.
> >
> > To be honest, I am not truly convinced by the _it's a feature not a bug_ approach, but I may consider engaging discussion with other reviewers to see how do they feel about this.

---

> > > ### Author Response · Authors · 2023-08-15
> > >
> > > Thank you for keeping an open mind. We do not intend to oversell our results: what we mean by "features" is just the properties of the intensity function and of our method (the good, the bad and the ugly). Our main point is that this perspective (which is not just ours, as it has been considered by others) is different than the prevailing TDA paradigm and worth investigating, not only mathematically (as it was done by [CD19]) but also statistically.

---

### Official Review · Reviewer_LEMN · 2023-07-06

**Soundness:** 3 good
**Presentation:** 2 fair
**Contribution:** 2 fair
**Rating:** 4
**Confidence:** 4

**Summary:**

The paper proves several theoretical inequalities involving the optimal transport distance, intensity, and density functions on a plane triangle with a non-standard boundary motivated by persistent homology.

**Strengths:**

The paper rigorously proves in appendix C six theorems and three corollaries from section 3. All definitions, statements, and proofs are written in great detail. Also, the paper is well-written overall.

**Weaknesses:**

Starting already from section 2 about the background, it seems that the term "persistence" is not really needed because there is no connection with real data.

Borel sets, measures, densities, and other concepts of classical probability theory can be considered on a plane triangle without the "persistent" adjective.

Hence it is strange to read lines 97-99 saying that "measure and probability are not yet standard concepts in the practice and theory of TDA. As a result, they have not been thoroughly investigated" because measure and probability are standard concepts in probability theory for nearly 100 years.

The conclusions reveal the main theoretical weaknesses in lines 328 and 332: "Our main focus is on the estimation of the persistence intensity function [CD19, CWRW15]." More explicitly, line 106-109 accepts that "[CD19] provided explicit expressions for p and p˜  ... We will refer to the functions p and p˜ as the persistence intensity and the persistence density functions, respectively. We remark that the notion of a persistence intensity function was originally put forward by [CWRW15]."

**Questions:**

Taking into account lines 106-109, what is the theoretical advance in the paper over the past work [CD19, CWRW15]?

Are Assumptions 3.2, 3.3, and 3.4 essential for the proven results? Do you have counter-examples to the theorems and corollaries when one of these assumptions fails?

Even if we accept Assumptions 3.2, 3.3, and 3.4, can the word "persistence" be removed from sections 2-3 so that all results are proved for measures on any plane triangle? The words "Betti numbers" can be easily defined for any triangle bounded by the diagonal x=y. Then will the paper become much more suitable for a more theoretical venue in statistics?

Do Assumptions 3.2, 3.3, and 3.4 hold for persistence diagrams obtained from the experiments in section 4?

The main practical weakness is the lack of a problem statement for the data mentioned in section 4. Is this data real or simulated? Some pictures would be helpful.

**Limitations:**

Though the paper doesn't include the required keyword "limitation", the limitations appear in Assumptions 3.2, 3.3, 3.4. For instance, Assumption 3.4 essentially requires that there is not too much "little noise".

In a simple case of the sublevel persistence of a scalar function, this function can be perturbed only by introducing a "bounded amount" of pairs of adjacent local maxima and minima. More exactly, the persistence diagram allows only a bounded sum of "noisy artefacts" near the diagonal.

---

> ### Author Rebuttal · Authors · 2023-08-09
>
> **Comparison with previous literature.**
>
> The notion of persistence intensity function is neither due to us nor new, and has been considered and used before: it has been suggested by [CWRW15], used in practical applications, e.g. by [WNv+ 21], and recently formalized and studied in great detail by [CD19] in the TDA literature. Indeed our results complement the contributions of [CD19], who posed the problem of statistical estimation of persistence intensity functions but did not tackle it (so our results leverage theirs, but are different n. In this regard, our results fit well within a recent line of work in the TDA/statistical literature and offer new results. We will provide more detailed commentary and comparisons with existing papers.
>
> **Probability theory and the use of the word persistence.**
>
> Our analyses rely critically on the fact that the points in the persistence diagram belong to a plane triangle and that the total persistence is bounded. Furthermore, our results are specifically tailored to TDA settings, so we prefer to keep using the word ``persistence" to make this explicit, even if in principle our analysis could be made more general. We will rephrase the sentence in lines 97-99. We did not mean to imply that measure theory and probability are not standard concepts in TDA!
>
> **Assumptions.**
>
> While we agree with the reviewer that it would be interesting to determine how critical  Assumptions 3.2, 3.3 and 3.4 are,
> we will abstain from pursuing this line of research for the following reasons: (i) those assumptions are standard and widely used in the TDA and non-parametric statistical literature; and (ii) weakening these assumptions may be feasible but it will produce significant technical challenges, and will likely lead to new and weaker assumptions that are also much more technical.
>
> **Experiments.**
>
> The original ORBIT5K dataset consists of simulations. We will include additional figures in the appendix to illustrate the data.

---

> > ### Comment · Reviewer_LEMN · 2023-08-11
> > **remaining questions**
> >
> > Thank you for the reply.
> >
> > >our results complement the contributions of [CD19], who posed the problem of statistical estimation of persistence intensity functions
> >
> > Could the authors please specify the exact place (problem number or at least a page) in [CD19], where this problem was posed?
> >
> > The following two questions seem unanswered.
> >
> > Taking into account lines 106-109, what is the theoretical advance in the paper over the past work [CD19, CWRW15]?
> >
> > Do Assumptions 3.2, 3.3, and 3.4 hold for persistence diagrams obtained from the experiments in section 4?
> >
> > Could the author comment on their results in the context of the work by Bobrowski et al (https://www.nature.com/articles/s41598-023-37842-2, available at https://arxiv.org/abs/2207.03926 since July 2022) claiming "a surprising discovery: normalized properly, persistence diagrams arising from random point-clouds obey a universal probability law" (quoted from the abstract)?

---

> > > ### Author Response · Authors · 2023-08-15
> > >
> > > Apologies for the terseness in our response.
> > >
> > > In [CD19] the authors formalize in rigorous mathematical ways the notion of the persistence intensity function, showing that, under mild conditions, this function is not only well-defined but in fact corresponds to the Radon-Nykodin derivative of the expected persistence measure w.r.t. Lebesgue measure. Their core contributions are mathematical, but in (the brief) Section 8, they suggest using a kernel-density-based method to estimate the persistence intensity function and, using results from non-parametric statistics, prove that a data-driven method for selecting the bandwidth is asymptotically optimal (with respect to the mean square error loss). The authors focus on the mean integrated squared error for the problem of bandwidth selection, and suggest the consistency for the mean squared error but do not investigate into detailed conditions and proofs. In contrast, in our paper, we conduct a detailed finite sample statistical analysis of the same estimator, and provide rates of consistency under the sup-norm metric, as opposed to the mean squared error, so our statistical contributions are stronger. Our results rely on the mathematical formalism and tools of [CD19] but tackle material that was only alluded to and not analyzed in [CD19]. In this sense, our results complement theirs.
> > >
> > > The manuscript [CWRW15] is arguably the first contribution to propose estimating the persistence intensity functions using a sample of many persistence diagrams. The authors provide a bound on the MISE (mean integrated squared error), while in our paper we focus on the stronger and more challenging sup-norm guarantees. Furthermore, our analysis is non-asymptotic and arguably more sophisticated and leverages the mathematical results from [CD19], which was not available to [CWRW15].
> > >
> > > About assumptions 3.2, 3.3 and 3.4: yes, they are satisfied in our experiments. Thank you for pointing this out. We will clarify it in our revision.
> > >
> > > Thank you so much for pointing out the reference by Bobrowski and Skraba about the fascinating conjecture of the universality of the noise distribution. We will include that reference in our revision. Their conjecture is that an appropriately standardized aggregate statistic of the points in the persistence diagram will have a universal limiting distribution. There are two key differences from our approach: (i) the asymptotic are different: Bobrowski and Skraba are concerned with the limiting behavior arising from one persistence diagram computed using an increasing number of sample points while we instead consider an increasing number of persistent diagrams, each evaluated with a fixed number of sample points and (ii) the conjectured universality is about the limiting behavior of an aggregate statistic, while we are focused on the entire distribution of the topological noise (as captured by the persistence measure).

---

### Official Review · Reviewer_LDNi · 2023-07-07

**Soundness:** 3 good
**Presentation:** 3 good
**Contribution:** 3 good
**Rating:** 7
**Confidence:** 3

**Summary:**

This work develops a set of methods and theories for statistical inference for TDA based on samples of persistence diagrams:
a. The work focuses on the estimation of the persistence intensity function. The work also proposes the novel persistence density function, which is the normalized version of the persistence intensity function.
b. The work present a class of kernel-based estimators based on an iid sample of persistence diagrams and derive estimation rates in the supremum norm, which is stronger than the optimal transport distance norm.
c. The work obtains uniform consistency rates of estimating linear representations of persistence diagrams, including Betti numbers, persistent surfaces, persistent silhouttes and weighted Gaussian kernels.
d. The work presents several theorems, theorem 3.1 compares the L^\infty norm and the optimal transport distance in terms of controlling the estimation error; theorem 3.5, 3.6 show the kernel estimation error bound for the persistent density function and the persistent intensity function; theorem 3.8, 3.9 show the estimation error bounds for the linear representations.

These theoretic results are fundamental and important, they lead to novel direction for statistical inference for TDA based on random persistent diagrams.


**Strengths:**

This work is very solid, and gives rigorous mathematical proofs. The formulations of key concepts, main theorems are clear and rigorous, the mathematical deductions for lemmas, theorems, corollaries are thorough and in detail. The theoretic results are convincing and impressive.

**Weaknesses:**

The work is highly theoretical, the heavy mathematical deductions are abstract. It will be more helpful for readers to digest if the authors further explain the motivations, the main proof approaches, the interpretations of the theorem, the potential direct applications of the results. More specifically,

1. It will be helpful for readers to better understand the article to give a table of symbols, list the major symbols and their meanings;
2. It will be helpful to give some figure to illustrate the concepts, such as persistent diagram;
3. Some math symbols and operators can be further explained, such as:
a. The two symbols in line 145 are hard to differentiate, especially on a laptop screen, maybe the authors can emphasize the shuttle differences, or use different symbols;
b. The formula in line 165, \|x-y\|_2^q needs more explanation
c. The formula in line 247 in the supplementary, the operator Proj_{\partial\Omega} needs more explanation






**Questions:**

1. The interpretation of the concept of random persistent diagram. For example, if we consider the MNIST data sets. Do we treat each image as a point cloud and build a persistent diagram ? Do we only consider the images of one digit or different digits together? Where does the randomness come from? Different writing styles of different people ? random noises in the imaging process ?

2. Theorem 3.1, the result is very general and can be applied in much broader fields. Does the inequality hold on general compact domains ? How tight is the bound ?

3. In the proof of theorem 3.1, please explain the current admissible transport plan . Is there other admissible transport scheme, which can lead to tighter estimation?

4. Please give some direct applications of the estimated persistent intensity/density functions, can we use it for generating persistent diagrams？ recognize， authenticate， classify persistent diagrams in TDA？

**Limitations:**

This work is theoretical, it mainly focuses on theoretical deductions. The limitations are not adequately addressed. It will be helpful if the limitations in practical applications are further discussed.
1. In reality, how difficult to satisfy all the assumptions listed in the paper ?
2. If the point cloud include several homology generators with similar birth and death times, may the current approach mix them and cause confusion ?
3. In theorem 3.6, from samples close to the diagonal, can we get more precise estimation ?

---

> ### Author Rebuttal · Authors · 2023-08-09
>
> We thank the reviewer for the constructive comments.
>
> 1. **Experiments.** In the MNIST data set, we treat each image as a point cloud and construct a persistence diagram. Images representing the same number can be regarded as generated from the same distribution, while those representing different numbers are generated from different distributions. The noise comes from various factors like, as the reviewer mentions, writing styles or imaging process.
>
> 2. We appreciate that the reviewer recognizes the value of Theorem 3.1. In our version, the total mass of the measures $\mu$ and $\nu$ may be different, so the optimal transport distance is defined such that the mismatching mass may be transported to the diagonal $\partial \Omega$. In general, for any compact set $\mathcal{C}$ and any measures $\mu,\nu$ such that $\mu(\mathcal{C}) = \nu(\mathcal{C})$, it can be guaranteed that
>
> $$
> \mathsf{OT}\_{p}^{p}(\mu,\nu) \leq [\mathsf{diam}(\mathcal{C})]^p \mathsf{Vol}(\mathcal{C}) \left\Vert p_{\mu}-p_{\nu}\right\Vert_{\infty}.
> $$
>
> Here, $\mathsf{diam}(\mathcal{C})$ and $\mathsf{Vol}(\mathcal{C})$ represent the diameter and volume of the compact set $\mathcal{C}$ respectively.
>
> Besides, the upper bound we show in Theorem 3.1 is also tight: it can be verified that the equation is reached when we pick $\mu$ as uniform on $\Omega$ and $\nu$ as the null measure $\nu(\Omega) = 0$ -- as we will show in the next paragraph.
>
> 3. Again, we thank the reviewer for the recognition of the theoretical value of Theorem 3.1 and going through the details of its proof. As we explained in lines 268-270 in the supplementary material: "*Intuitively,* $\hat{\pi}$ *represents such a transport: at each point* $\mathbf{x} \in \Omega$, *if* $p_{\mu}(\mathbf{x}) > p_{\nu}(\mathbf{x})$, *then we transport the mass of* $p_{\nu}$ *from* $\mathbf{x}$ *to* $\mathbf{x}$, *and the remaining mass from* $\mathbf{x}$ *to its projection onto* $\partial \Omega$; *if* $p_{\nu}(\mathbf{x}) > p_{\mu}(\mathbf{x})$, *then the opposite is done.*"
>
> We believe it is not possible to find another admissible transport that leads to a tighter estimate. Consider the following example: let $p_{\mu}(\mathbf{\omega}) = 1$ and $p_{\nu}(\mathbf{\omega}) = 0$ for all $\mathbf{\omega} \in \Omega$. Essentially, $\mu$ is uniform on $\Omega$ and $\nu$ has zero mass. In this case, all the mass in $\mu$ would need to be transported to the diagonal $\partial \Omega$, and the optimal transport plan is to transport the mass on every point $\mathbf{\omega} \in \Omega$ to its projection onto $\partial \Omega$. It is therefore easy to verify that
> $$
> \left\Vert p_{\mu} - p_{\nu}\right\Vert_{\infty} =1,
> $$
> and
> $$
> \mathsf{OT}\_{p}^{p}(\mu,\nu) = \int\_{\Omega} \left\Vert\mathbf{\omega} - \partial \Omega\right\Vert\_{2}^{q} \mathrm{d} \mathbf{\omega} = \frac{2}{(q+1)(q+2)} \left(\frac{L}{\sqrt{2}}\right)^{q+2}.
> $$
>
> 4. The framework we proposed can be applied to classify two populations of persistent diagrams by comparing their estimated intensity functions. Representing distributions of persistence diagrams with intensity functions offer a very concrete way to compare them and quantify the magnitude of their differences. This is indeed a main motivation for our work. We will make this point explicit in our revision.

---

### Official Review · Reviewer_7uZh · 2023-07-09

**Soundness:** 3 good
**Presentation:** 2 fair
**Contribution:** 3 good
**Rating:** 5
**Confidence:** 2

**Summary:**

The paper tackles the problem of estimating the persistence intensity function, describing the distribution of rando persistence diagrams, and proposes a variant called persistence probability function that integrates to one.  The paper starts with a theoretical analysis of the estimation error bound of the intensity function using the OT measure and the L-infinite norm, showing that the latter allows the definition of stricter bounds. The paper also proposes a method to estimate the persistence intensity function and the persistence probability function under the assumption of i.i.d. samples using a kernel density estimation approach.

**Strengths:**

Persistent diagrams are an important tool for characterizing topological structures (e.g. surfaces and graphs). Being able to estimate with high accuracy the distribution of such structures could indeed be beneficial to their analysis, with applications also to the learning domain.

The paper, at least from a not-so-expert reader like I am, seems very rigorous in the theoretical analysis and provides in the sup. mat. all the proofs of the introduced theorems.



**Weaknesses:**

The main weakness (if we want to call it so) of the paper is that it is not easy to read by nonexperts of the specific topic. It is quite dense and mostly mathematical and does not provide many intuitive explanations of why some properties could be important from a practical perspective.
In general, I would have appreciated a more gentle introduction to the problem and a wider introduction/literature review on the practical application/advantages of persistent intensity functions.


**Questions:**

I think that it would help to add some details in the introduction about the importance and applicability of the proposed tool, which is still not clear to me.
In the conclusions, you state that statistical inference is not yet possible with the proposed method. Does this mean that it cannot be used in practice?

Row 101: of problems
Row 139: of the expected


**Limitations:**

I don’t foresee any particular negative societal impact. A discussion about the practical applicability of the method would be interesting.

---

> ### Author Rebuttal · Authors · 2023-08-09
>
> Our methodology can certainly be used in practice. However, our statistical guarantees only show the consistencies of our estimators. In order to carry out more sophisticated inferential tasks, e.g. hypothesis testing and confidence sets, a more refined analysis is in order. For example, in order to compute an asymptotically valid confidence band for the persistence intensity/density function, it appears necessary to study the validity of the bootstrap or other resampling methods. This is a non-trivial task that we will leave for future work.
>
> We will improve the language in the introduction to provide better motivations for our contributions.

---

### Author Rebuttal · Authors · 2023-08-09

We would like to clarify a few important points about our paper that perhaps we did not express as clearly as we intended to. We will include additional text in the introduction and throughout the manuscript to make sure there will not be confusion.

Our work is not intended to suggest an alternative framework to the current and prevailing TDA practices.  Rather, we explore a statistically grounded approach whose main objective is to describe the  *distribution* of a random persistence diagram and not any particular realization or target persistence diagram. As we are interested in capturing the overall randomness of persistence diagrams,  we seek to represent both the topological signal and the topological noise, the latter being our primary target. Concretely, and using the example suggested by the reviewer, suppose that we are interested in the distributions of persistence diagrams originating from a uniform distribution and from a non-uniform distribution on the unit sphere whose density is, say, inversely proportion to the arc length distance from an arbitrary reference point on the sphere. The topological signature is the same in both cases (that of the unit sphere), but the topological noise is different, having different distributions. See **Figure 1 in the attached pdf file** for an illustration. In this paper we study a methodology, rooted in the literature on non-parametric density estimation, that is in principle able to identify and quantify such a difference using the highly interpretable quantity of persistence intensity function. We believe this contribution is neither trivial nor lacking rationale and in fact may hold the potential to yield new statistical methods for TDA.

Secondly,  the notion of persistence intensity function is neither due to us nor new, and has been considered and used before: it has been suggested by [CWRW15], used in practical applications, e.g. by [WNv+ 21], and recently formalized and studied in great detail by [CD19] in the TDA literature. Indeed our results complement the contributions of [CD19], who posed the problem of statistical estimation of persistence intensity functions. More generally, if one is inclined to treat a persistence diagram as a (complex!) point process, the intensity function is a natural object to investigate.

Thirdly, in addition to focusing on distributional properties of persistence diagrams (as a means to express topological noise), our analysis differs from standard TDA approaches in that we assume different asymptotic behavior, by requiring the availability of a growing number of i.i.d. persistence diagrams and *not* of a growing number of i.i.d. data points to construct each observed persistence diagram.

Finally, the methodology we consider is computationally inexpensive to apply, a feature that does not always apply to TDA methods.

In conclusion, we believe that persistence intensity functions provide an alternative set of tools for statistical inference for TDA that is rather distinct and does not clash with existing TDA paradigms and, as such, is worth being investigated. In this paper we take the first step towards studying the validity and limitation of such an approach.

We will expand the introduction and the main body of the paper to clarify our perspective and provide better context.

---

### Decision · Program_Chairs · 2023-09-21

**Decision:**

Reject

**Comment:**

The paper considers the problem of estimating persistence intensity & density functions and specifically derives convergence rates for the considered kernel-based estimators. Throughout the full review period, this manuscript remained with quite diverse scores and reviews. Although the authors provided a thorough rebuttal, the overall impression of the paper is still mixed, with some questions/concerns answered and addressed while others remain up for debate. After reading the paper myself and considering the reviews, my impression is that this manuscript would have to be rewritten in many parts (1) to make it more accessible to the non-expert reader and (2) to clarify the concerns raised by LEMN and 1g88. In particular, while I partially agree with the author's rebuttal to the concerns of reviewer 1g88, I share the opinion that discounting some concerns as "feature, not a bug" is somewhat of an easy way out. In my opinion, it is worth taking the comments to heart; addressing them in the paper in a thorough way will make the manuscript stronger. Taking all points into consideration, I am recommending a "Reject" at this point, as the changes required seem too substantial to be covered within the NeurIPS review-rebuttal-cycle (in particular, another full review cycle to fully address the reviewer concerns).